# PLANNER AWARE PATH LEARNING IN DIFFUSION LANGUAGE MODELS TRAINING

**Fred Zhangzhi Peng**[1,‡,*] **Zachary Bezemek**[1,‡,*]**, Jarrid Rector-Brooks**[2,3,4]**, Shuibai Zhang**[5]**,
Anru R. Zhang**[1]**, Michael Bronstein**[6,7]**, Alexander Tong**[7†]**, Avishek Joey Bose**[2,6,8†]

[1] Duke University    [2] Mila    [3] Université de Montréal [4] California Institute of Technology
[5] University of Wisconsin–Madison [6] University of Oxford [7] AITHYRA
[8] Imperial College London    [‡] Equal contribution    [†] Equal advising

## ABSTRACT

Diffusion language models have emerged as a powerful alternative to autoregressive models, enabling fast inference through more flexible and parallel generation paths. This flexibility of sampling is unlocked by new engineered sampling strategies, or *planners*, that select more favorable generation paths by iteratively planning—versus uniformly at random—where to denoise along the sequence. However, by modifying the reverse paths via planning, planners create an irrevocable mismatch between the uniformly random denoising paths assumed during training and planning-based inference. In this paper, we systematically investigate the mismatch of discrete diffusion training and inference under planning and theoretically prove that the standard discrete diffusion training evidence lower bound (ELBO) does not accurately describe a denoiser that uses a non-uniform planner. To address this gap, we derive a new planned evidence lower bound (P-ELBO) that incorporates planner-based reverse dynamics directly into the training objective. Using the P-ELBO, we introduce *Planner Aware Path Learning* (PAPL), a novel training scheme that aligns training and inference under a planned denoiser. PAPL is implemented as a simple yet effective modification to the standard masked discrete diffusion loss, making it widely applicable and easy to adopt. Empirically, we show PAPL delivers consistent gains across domains, including a 40% relative improvement in protein sequences, improved text generation with up to a $4\times$ relative MAUVE gain, and 23% relative improvement in code generation HUMANEVAL pass@10. Code is available at github.com/pengzhangzhi/PAPL.

## 1 INTRODUCTION

The landscape of generative modeling over discrete data has led to foundational breakthroughs in deep learning, with Large Language Models (LLMs) being an exemplary technology that has transcended beyond natural language processing (Achiam et al., 2023). Until recently, the de facto gold standard for building LLMs has been Autoregressive models (ARMs), which are highly scalable for pre-training LLMs—allowing them to capture complex dependencies in data—but incur rigid inference schemes due to the autoregression mechanism that generates samples in a causal order—e.g., left to right for natural language (Guo et al., 2025). In contrast to ARMs, recent advances in Diffusion Language Models (DLMs) have the potential to disrupt the current status quo for generative modeling of discrete data, as they natively support flexible generation orders and allow for fully parallel sampling of tokens at inference time (Austin et al., 2021a; Lou et al., 2024; Sahoo et al., 2024; Shi et al., 2024). The increased flexibility of modeling discrete data in any order makes DLMs arguably a more natural tool than ARMs for tackling high-impact problem domains that lack a natural causal ordering, such as biological sequence design and code completion (Nie et al., 2025a; Wang et al., 2024; 2025b), spurring their rapid recent development and application (Song et al., 2025).

Indeed, the most performant variant of DLMs, i.e., Masked Diffusion Models (MDMs) (Shi et al., 2024; Sahoo et al., 2024), approach generative modeling as a denoising task, wherein partially masked

---

*Correspondence to `zp70@duke.edu` and `zwb@duke.edu`

sequences are iteratively refined using a learned denoiser that time-reverses the Markov transition dynamics of the masked corruption process. However, a key assumption, and thus a central limitation, of DLMs is that following the reverse dynamics of uniformly denoising a position at inference implicitly assumes denoising with a *perfect denoiser* (Peng et al., 2025a). As a result, in practice, to fully take advantage of flexible generation paths and generate higher-quality samples beyond uniform decoding, the reverse process must be modified by a *planner*: a rule that selects which tokens to reveal next, such as greedy decoding (Chang et al., 2022), ancestral sampling (Shi et al., 2024; Schiff et al., 2025), or Path planning (P2) (Peng et al., 2025a). In fact, using a planning strategy is more than a humble artefact of optimizing for inference; it can be seen as avoiding denoising over exponential infilling problems at inference—unlike training, in which the task is provably computationally intractable (Kim et al., 2025). More than just theory, employing a planner when denoising has been shown to substantially improve sample quality across various application domains, including text, code, protein sequences, and discrete image modeling (Peng et al., 2025a; Nie et al., 2025a; Shi et al., 2024).

A fundamental aspect of DLM inference under planning is the fact that denoising is not necessarily conducted by uniformly picking a position to unmask. Consequently, this new reverse process creates an irrevocable mismatch between the forward masking process, used during training, which corrupts sequences by masking positions uniformly at random. This mismatch of forward and reverse processes also suggests that, in effect, training denoisers in DLMs attempts to solve a harder problem than the one they are ultimately used for. This raises the following central question:

> **Q**. *How should we adapt the training of denoisers in diffusion language models when inference inevitably proceeds under a planner?*

**Present work**. In this paper, we seek to answer the question by introducing a new theoretical framework that aims to align the training of DLMs with *pre-assumed knowledge* of planner-based inference. Our framework is built using basic facts of Markov chains, which allows us to set up the training of DLMs as minimizing a path-wise KL divergence. More precisely, the path-wise KL is between the reverse dynamics of a DLM using a planner and supervised ideal reverse dynamics, also under planning, that hit the data distribution.

Armed with this path-wise KL, we first theoretically prove in §3.1 how greedy ancestral sampling at inference of DLMs violates the standard DLM ELBO (Sahoo et al., 2024; Shi et al., 2024)—validating the thesis of a mismatch of forward and reverse dynamics under planning. We next derive a new planned evidence lower bound (P-ELBO) in §3.2, of which the standard planning-agnostic—i.e., uniformly at random denoising—DLM ELBO is a special case. Furthermore, we demonstrate recent heuristics that act as planners, such as MaskGIT (Chang et al., 2022), and P2 (Peng et al., 2025a), emerge as principled instances of our new P-ELBO, which unifies existing strategies under one umbrella. Given the insights in P-ELBO, we propose a new loss function for training the denoiser that directly incorporates any choice of planner, and thereby allows training to match inference properly. We summarize our main contributions in this paper as follows:

- **Unifying framework**. We derive a novel generalized planner-aware generalized lower bound (P-ELBO) that takes into account the use of planning in the reverse dynamics of a DLM.
- **Efficient implementation**. Starting from the P-ELBO, we design a new simplified loss termed *Planner Aware Path Learning* (PAPL) that amounts to a one-line code change and uses self-planning. Specifically, PAPL leverages the denoiser itself—i.e., places where the denoiser is most confident—to compute a weighted loss on more likely generation paths compared to standard DLMs.
- **Improved performance**. Empirically, PAPL consistently improves the quality of diffusion language models under identical configurations. We observe PAPL in protein sequence generation yields a $40\%$ relative increase in foldability, surpassing larger diffusion and autoregressive baselines while preserving diversity. On code generation, PAPL improves HUMANEVAL pass@1 from 18.5 to 20.8, pass@10 from 31.1 to 38.4, and HUMANEVAL-INFILL pass@1 from 30.0 to 32.5. On text generation, PAPL achieves up to a $4\times$ improvement in MAUVE and reduces generative perplexity by over $40\%$ compared to prior diffusion models.

## 2 BACKGROUND AND PRELIMINARIES

**Notation**. Let $\mathcal{V} = \{1, \dots, d\}$ denote a finite vocabulary. We reserve the final symbol, $d = \mathbf{m}$, as a special *mask token*, while the remaining $d - 1$ symbols correspond to ordinary vocabulary items. We

consider sequences of length $L$, so that a data point is represented as $\boldsymbol{x} = (x^1, \ldots, x^L) \in \mathcal{V}^L$. The empirical data distribution $p_{\text{data}}$ is supported on a training set $\mathcal{D} \subset \mathcal{V}^L$. We denote by $\Delta^d = \{u \in \mathbb{R}^d : u^i \geq 0, \sum_{i=1}^d u^i = 1\}$ the probability simplex. Each $u \in \Delta^d$ specifies a categorical distribution $\text{Cat}(j; u) = u^j$ over $j \in \mathcal{V}$. For a particular token $x \in \mathcal{V}$, we write $\delta(x) \in \Delta^d$ for the one-hot distribution that places all its mass on $x$. To avoid ambiguity, we use superscripts (e.g., $x^i$) to index sequence positions, and subscripts (e.g., $x_t$) to index time steps of a stochastic process. For $\mathbf{x}, \mathbf{y} \in \mathcal{V}^L$, we use $d_{\text{HAM}}(\mathbf{x}, \mathbf{y})$ to denote the Hamming distance between $\mathbf{x}$ and $\mathbf{y}$. We also use $N_M(\mathbf{x})$ to denote the number of coordinates in $\mathbf{x}$ which are equal to $\mathbf{m}$. For a finite set $S$, we use $\text{Unif}(S)$ to denote the uniform distribution on that set. For $l < n \in \mathbb{N}$, we denote by $[l : n] = \{l, l+1, \ldots, n\}$.

## 2.1 Masked Diffusion Language Models

A masked diffusion language model (MDLM) generates samples from a data distribution $p_{\text{data}} \in \Delta^{d^L}$ through an iterative sampling process which gradually denoises a full mask sequence $[\mathbf{m}]^L$ to a sequence which does not contain any masks. This iterative sampling procedure makes use of a denoiser $D_\theta : \mathcal{V}^L \to (\Delta^d)^L$, which outputs a distribution over the clean tokens at each position. In particular, for $\mathbf{x} \in \mathcal{V}^L$ and $y \in \mathcal{V}, y \neq \mathbf{m}$, $\text{Cat}(y; D_\theta^i(\mathbf{x}))$ approximates the probability that the $i$'th token in a sequence is $y$ given the unmasked positions in the sequence match those of $\mathbf{x}$ (Hoogeboom et al., 2022; Sahoo et al., 2024; Shi et al., 2024; Zheng et al., 2025; Ou et al., 2025).

Moreover, the Gillespie sampling scheme (Gillespie, 1977; 1976) allows us to establish an exact equivalence between DLM and any-order autoregressive model (AOARM) (Uria et al., 2014; Hoogeboom et al., 2022). More precisely, the Gillespie sampling scheme allows for a single coordinate to be denoised at each step. Concretely, starting from the fully masked sequence, the procedure iteratively (1) selects a position uniformly at random among the currently masked tokens, (2) samples a replacement token from the denoiser's predictive distribution at that position, and (3) updates the sequence by filling in the chosen token while leaving all other positions unchanged.

Let $p_\theta^{\text{unif}}$ be the distribution on $\mathcal{V}^L$ of $\mathbf{x}_L$ resulting from applying the above iterative sampling procedure in which the masked coordinate to denoise is chosen uniformly at random. This uniform unmasking process is the main sampling strategy employed by MDLMs (Sahoo et al., 2024; Shi et al., 2024). The connection between MDLMs and AOARMs (Ou et al., 2025) can be seen more explicitly through the MDLM evidence lower bound (ELBO), which lower bounds the log marginal $\log(p_\theta^{\text{unif}}(\mathbf{x}_0))$,

$$\log(p_\theta^{\text{unif}}(\mathbf{x}_0)) \geq \mathcal{E}^{\theta,\text{unif}}(\mathbf{x}_0) = \mathbb{E}_{\sigma \sim \text{Unif}(\Sigma^L)} \left[ \sum_{i=1}^L \log \left( \text{Cat}(x_0^{\sigma(i)}; D_\theta^i \left( \mathbf{x}_0^{\sigma(<i)} \right)) \right) \right] \quad (1)$$

$$= L \mathbb{E}_{k \sim \text{Unif}([0:L-1])} \left[ \mathbb{E}_{\mathbf{x}_k \sim \text{Unif}(\mathcal{X}_{L-k}(\mathbf{x}_0))} \left[ \sum_{i=1, x_k^i = \mathbf{m}}^L \frac{1}{L-k} \log \left( \text{Cat} \left( x_0^i; D_\theta^i(\mathbf{x}_k) \right) \right) \right] \right],$$

where $\Sigma^L$ is the set of all permutations of length $L$. Additionally, for $\sigma \in \Sigma^L$, we denote $\sigma(< i)$ as the first $i-1$ elements of $\sigma$. Thus, for $\mathbf{x} \in \mathcal{V}^L$, $\mathbf{x}^{\sigma(<i)} \in \mathcal{V}^L$ has coordinates in $\sigma(< i)$ set to those of $\mathbf{x}$ and the rest set to $\mathbf{m}$, and for $\mathbf{x} \in \mathcal{V}^L$ and $k \in [0 : L]$, $\mathcal{X}_k(\mathbf{x}) \subset \mathcal{V}^L$ is the set of sequences which are the same as $\mathbf{x}$ but with exactly $k$ coordinates masked.

**Existing DLM Sampling Strategies**. While the vanilla masked diffusion sampler proceeds by unmasking one position chosen uniformly at random, a variety of alternative planners have been proposed to improve generation quality by biasing the unmasking order. A straightforward modification is greedy decoding as in MaskGIT (Chang et al., 2022), where at each step the denoiser selects the position with the highest confidence to unmask next (Gong et al., 2025). Another line of work introduces remasking, where previously generated tokens may be reverted to the mask state and resampled. Resampling diffusion models (RDM) (Zheng et al., 2023) extend greedy planning with resampling. More recently, path planning (P2) (Peng et al., 2025a) has been proposed as a unifying framework that generalizes all of the above strategies. P2 decomposes each step into a planning stage, where a planner chooses positions to update—including both masked and already unmasked tokens—and a denoising stage, where the selected positions are resampled with the denoiser.

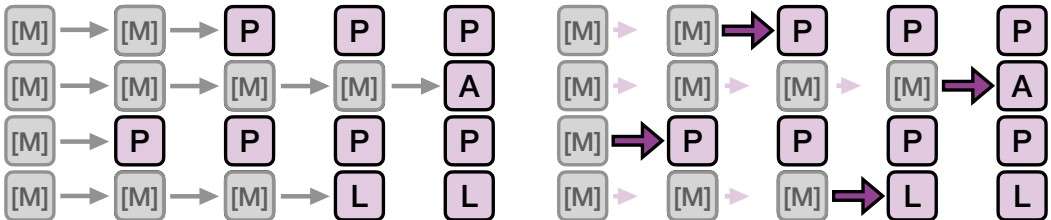

Figure 1: **Planner-Aware Path Learning (PAPL) resolves training–inference mismatch in DLMs.** Standard uniform training for DLMs (left) applies a uniform loss across all masked positions, distributing capacity over regions that inference-time planners never traverse. PAPL (right) introduces planner-aware weights into the loss, aligning training with the planner's preferred trajectories (outlined arrows) and eliminating training-inference mismatch.

## 3 METHOD: PATH LEARNING

We now present *Planner-aware Path Learning* (PAPL), aiming to align training with planner-based inference. In §3.1, we introduce a general formulation of sampling with a planner and derive the corresponding transition probabilities. In §3.2, we show that greedy sampling modifies these dynamics in a way that departs from the MDLM ELBO, motivating the need for a new objective. We then derive a planner-aware ELBO in(P-ELBO) §3.3, from which different sampling schemes—including uniform and greedy ancestral—emerge as special cases (§A.2). Finally, in §3.4, we simplify the P-ELBO into an efficient training objective, leading to our final efficient (PAPL) algorithm (Alg. 1).

**Discrete-time Markov Chains**. The starting point for our analysis is to re-examine the ELBO in Equation (1) through the lens of a continuous time Markov chain (CTMC) (Campbell et al., 2022; 2024; Lou et al., 2024; Sun et al., 2023). From the CTMC perspective, the sampling path of a DLM can be compared against a family of reference chains parameterized by a sample $\mathbf{x}_0$, which generates that specific datum. For ease of presentation, we forego lifting to this continuous perspective and instead compute the ELBO via a *discrete-time* Markov chain of the DLM $X^\theta$ to that of a family of reference discrete Markov chains $Y$, before taking appropriate limits to recover the continuous perspective. Specifically, for a datum $\mathbf{x}_0$ we have discrete Markov chains of length $L$ which satisfy $X_0^\theta \overset{d}{=} Y_0^{\mathbf{x}_0}$ and $Y_L^{\mathbf{x}_0} = \mathbf{x}_0$.

We now recall that the transition matrix $Q$ for a discrete time Markov chain $X$ encodes $Q(y, x) = \mathbb{P}(X_{k+1} = y | X_k = x)$. From this perspective, for vanilla DLMs we have $p_\theta^{\text{unif}}(\mathbf{x}) = \mathbb{P}(X_L^\theta = \mathbf{x})$ where $X_0^\theta = [\mathbf{m}]^L$ and $X^\theta$'s dynamics are described the transition matrix, given by:

$$Q^\theta(\mathbf{y}, \mathbf{x}) = \begin{cases} \text{Cat}(y^i; D_\theta^i(\mathbf{x}))/N_M(\mathbf{x}) & , d_{\text{HAM}}(\mathbf{x}, \mathbf{y}) = 1, x^i = \mathbf{m}, y^i \neq \mathbf{m} \\ 0 & , \text{ otherwise} \end{cases},$$

where $d_{\text{HAM}}$ is the hamming distance and $\mathbf{x}, \mathbf{y} \in \mathcal{V}^L$. We compare to $Y^{\mathbf{x}_0}$ with $Y_0^{\mathbf{x}_0} = [\mathbf{m}]^L$ and define the transition matrices as follows:

$$R(\mathbf{y}, \mathbf{x}; \mathbf{x}_0) = \begin{cases} \text{Cat}(y^i; \delta(x_0^i))/N_M(\mathbf{x}) & , d_{\text{HAM}}(\mathbf{x}, \mathbf{y}) = 1, x^i = \mathbf{m}, y^i \neq \mathbf{m} \\ 0 & , \text{ otherwise} \end{cases}. \quad (2)$$

In this view, the second expression of equation 1 is can be rewritten as:

$$\mathcal{E}^{\theta, \text{unif}}(\mathbf{x}_0) = L \mathbb{E}_{k \sim \text{Unif}([0:L-1])} \left[ \mathbb{E}_{\mathbf{x}_k \sim r_k(\cdot; \mathbf{x}_0)} \left[ \sum_{\mathbf{y} \in \mathcal{V}^L} R(\mathbf{y}, \mathbf{x}_k; \mathbf{x}_0)) \log \left( \frac{Q^\theta(\mathbf{y}, \mathbf{x}_k)}{R(\mathbf{y}, \mathbf{x}_k; \mathbf{x}_0)} \right) \right] \right], \quad (3)$$

where we set $r_k(\mathbf{x}; \mathbf{x}_0) = \mathbb{P}(Y_k^{\mathbf{x}_0} = \mathbf{x})$. We rederive the ELBO from this perspective by stating the (discrete) path wise KL divergence in our Proposition §A.7. Importantly, this ELBO enjoys a simple interpretation in how the denoised coordinates are chosen—a fact which will later develop to incorporate coordinates chosen under a planner in Proposition 3.2. Moreover, this perspective is more easily amenable to modifying the sampling dynamics for $X^\theta$ and deducing the corresponding ELBO.

### 3.1 REVERSE DYNAMICS WITH A PLANNER

We begin by introducing a planner function $G_\phi : \mathcal{V}^L \times \mathcal{V}^L \to \Delta^L$ that modifies the sampling process by selecting in reverse transition which coordinate in the sequence to be denoised next. For simplicity, we assume the planner only selects masked positions, i.e. $\text{Cat}(i; G_\phi(\mathbf{z}, \mathbf{x})) = 0$ whenever $x^i \neq \mathbf{m}$—matching greedy ancestral sampling (Nie et al., 2025a). Under these dynamics, each backwards step can be decomposed into two substeps as follows: 1.) starting from the current sequence $\mathbf{x}_k$, the denoiser produces candidate predictions $\mathbf{z} \sim D_\theta(\mathbf{x}_k)$ for all positions. 2.) The planner then samples an index $i \sim G_\phi(\mathbf{z}, \mathbf{x}_k)$, and the token at this index is updated by setting $x_{k+1}^i = z^i$ while all other positions remain unchanged. Thus, each transition from $\mathbf{x}_k$ to $\mathbf{x}_{k+1}$ differs at exactly one coordinate.

The probability of unmasking the $i$-th coordinate of $\mathbf{x}_k$ to token $y$, after marginalizing over $\mathbf{z}$ can be explicitly written as a transition kernel $q_{\theta,\phi}^i(y|\mathbf{x}_k)$ as follows:

$$q_{\theta,\phi}^i(y|\mathbf{x}_k) = \text{Cat}\left(y; D_\theta^i(\mathbf{x}_k)\right) F_{\theta,\phi}(\mathbf{x}_k, y, i) \tag{4}$$

$$F_{\theta,\phi}(\mathbf{x}, y, i) := \mathbb{E}_{\mathbf{z}\sim D_\theta(\mathbf{x})}\left[\text{Cat}\left(i; G_\phi(\mathbf{z}^{-i,y}, \mathbf{x}_k)\right)\right], \tag{5}$$

where $\mathbf{z}^{-i,y}$ denotes the same sample $\mathbf{z}$ except that the $i$-th component has been replaced with $y$.

### 3.2 GREEDY ANCESTRAL VIOLATES THE VANILLA DLM ELBO

Greedy ancestral sampling (Nie et al., 2025a), as widely used in the literature (Gong et al., 2025; Besnier et al., 2025) such as MaskGIT (Chang et al., 2022), employs a specific choice of planner which selects the most confident position according to the denoiser itself,

$$G_\phi(\mathbf{z}, \mathbf{x}_k) = \delta\left(\underset{j:x_k^j=\mathbf{m}}{\arg\max}\, \text{Cat}(z^j; D_\theta(\mathbf{x}_k))\right). \tag{6}$$

Unfortunately, when using greedy sampling, the standard DLM ELBO may not, in fact, satisfy the ELBO inequality. Using the transition probabilities equation 4, we prove the following:

**Proposition 3.1.** *For $p_\theta^{greedy}(\mathbf{x}_0)$ defined with $G_\phi$ in equation 6 and $D_\theta$ an imperfect denoiser, we may have*

$$\log(p_\theta^{greedy}(\mathbf{x}_0)) < \mathcal{E}^{\theta,unif}(\mathbf{x}_0),$$

*where $\mathcal{E}^{\theta,unif}(\mathbf{x}_0)$ is as in equation 1.*

We prove Proposition 3.1 in §A.1.7. The key takeaway is that the ELBO in equation 1 is only valid for the uniform unmasking process $p_\theta^{\text{unif}}$. As a result, the reverse dynamics of greedy sampling no longer match those assumed by the standard training loss. In a nutshell, the model is being trained for uniform unmasking, but inference follows a different process. More broadly, this insight applies to any planner: whenever the sampling procedure deviates from uniform, the training objective no longer strictly reflects the quality of the generated samples.

We remark that the proof of Proposition 3.1 hinges on constructing an explicit counter-example, in particular assuming the denoiser is inconsistent along different paths: That is, we assume in our construction that there are two different permutations $\sigma, \bar{\sigma}$ of $[1:L]$ and $\mathbf{x}_0 \in \mathcal{V}^L$ a clean sequence in the data distribution such that $\prod_{i=1}^L \text{Cat}\left(x_0^{\sigma(i)}; D_\theta^i\left(\mathbf{x}_0^{\sigma(<i)}\right)\right) \neq \prod_{i=1}^L \text{Cat}\left(x_0^{\bar{\sigma}(i)}; D_\theta^i\left(\mathbf{x}_0^{\bar{\sigma}(<i)}\right)\right)$. Indeed, for a perfect denoiser, in the above we would have equality for any $\mathbf{x}_0, \sigma$, and $\bar{\sigma}$, and sampling along any path $\sigma$ would always result in a sample from the data distribution. In this situation, there would be no point of planning a generation path. However, in practice there is no relationship being enforced between these quantities, and it has been observed repeatedly in the literature that the "path"="denoising order" taken greatly influences sample quality - see, e.g. Ou et al. (2025) Appendix J.4, Shih et al. (2022) Section 4, and Li et al. (2021) Section 6.

### 3.3 PLANNER-AWARE EVIDENCE LOWER BOUND

The key mismatch is that vanilla DLM training assumes uniform unmasking, while inference instead follows a planner. To correct this, we next introduce a planner-aware ELBO (P-ELBO) that explicitly accounts for the planner's role in the reverse dynamics.

**Proposition 3.2.** *For any planner $G_\phi$, let $p_\theta^{G_\phi}$ denote the distribution of $\mathbf{x}_L$ obtained via the planner-guided sampling scheme. Then we have the following ELBO:*

$$\log(p_\theta^{G_\phi}(\mathbf{x}_0)) \geq \mathcal{E}^{\theta,\phi}(\mathbf{x}_0) = \mathcal{E}_1^{\theta,\phi}(\mathbf{x}_0) + \mathcal{E}_2^{\theta,\phi}(\mathbf{x}_0),$$

$$\mathcal{E}_1^{\theta,\phi}(\mathbf{x}_0) = L \mathop{\mathbb{E}}_{k \sim Unif([0:L-1])} \left[ \mathop{\mathbb{E}}_{\mathbf{x}_k \sim r_k^{G_\phi}(\cdot;\mathbf{x}_0)} \left[ \sum_{i=1,x_k^i=\mathbf{m}}^{L} Cat(i; G_\phi(\mathbf{x}_0, \mathbf{x}_k)) \log \left( Cat(x_0^i; D_\theta^i(\mathbf{x}_k)) \right) \right] \right]$$

$$\mathcal{E}_2^{\theta,\phi}(\mathbf{x}_0) = -L \mathop{\mathbb{E}}_{k \sim Unif([0:L-1])} \left[ \mathop{\mathbb{E}}_{\mathbf{x}_k \sim r_k^{G_\phi}(\cdot;\mathbf{x}_0)} \left[ \sum_{i=1,x_k^i=\mathbf{m}}^{L} Cat(i; G_\phi(\mathbf{x}_0, \mathbf{x}_k)) \log \left( \frac{Cat(i; G_\phi(\mathbf{x}_0, \mathbf{x}_k))}{F_{\theta,\phi}(\mathbf{x}_k, x_0^i, i)} \right) \right] \right],$$

*where $r_k^{G_\phi}$ is the distribution at time $k$ of a Markov chain with initial data $(\mathbf{m}, \ldots, \mathbf{m})$ and transition rates as in equation 2 but with $1/N_M(\mathbf{x})$ replaced by $G_\phi^i(\mathbf{x}_0, \mathbf{x})$.*

A proof of this ELBO from a self-contained, discrete-time Markov chain perspective can be found in §A.1.4 and from a continuous time Markov chain perspective in §A.3.3. The first term $\mathcal{E}_1^{\theta,\phi}$ resembles the standard DLM ELBO: it is the log-probability of predicting the correct token in each masked position, but now *weighted by the probability that the planner would choose that position next*. In other words, it is a planner-weighted cross-entropy.

The second term $\mathcal{E}_2^{\theta,\phi}$ is new. It appears only when the planner's decision can depend on the full clean target sequence $\mathbf{x}_0$. Intuitively, it measures the gap between (a) the "ideal" planner that has access to the ground truth, and (b) the "effective" planner that only relies on the denoiser's predictions, and mathematically is the negative KL divergence between these two distributions. We highlight that for the uniform planner setting, this term vanishes, recovering the standard DLM ELBO. Putting this together, the planner-aware loss used for training is simply the negative ELBO from Proposition 3.2 $\mathcal{L}(\theta, \phi) = -\mathbb{E}_{\mathbf{x}_0 \sim p_{\text{data}}}\left[\mathcal{E}^{\theta,\phi}(\mathbf{x}_0)\right]$, which minimizes the KL divergence between the planner-guided model distribution $p_\theta^{G_\phi}$ and the data. We curate results on how several common existing instantiations of planned denoising fall into this framework and their corresponding ELBOs in §A.2. We also prove a result identifying the form of the optimal planner for a fixed denoiser under the trainling loss associated to the ELBO from Proposition 3.2 in §A.1.5.

### 3.4 EFFICIENT IMPLEMENTATION OF PLANNER AWARE PATH LEARNING (PAPL)

Greedy decoding is widely used at inference and often improves sample quality. A natural idea is therefore to train the denoiser under the same greedy planner but with corrections such that we optimize the P-ELBO. Unfortunately, the exact greedy ELBO from Cor. A.9 is computationally infeasible: simulating the greedy path for each data point requires $k$ denoiser evaluations at step $k$, whereas vanilla DLM training needs only a single forward pass. Consequently, we aim to design an

---

**Algorithm 1** PAPL Training

1: **while** not converged **do**
2:     $\mathbf{x}_0 \sim p_{\text{data}}(x)$
3:     $k \sim \text{Unif}([0 : L - 1])$
4:     $\mathbf{x}_k \sim \text{Unif}(\mathcal{X}_{L-k}(\mathbf{x}_0))$
5:     $w^i \leftarrow \text{Cat}(i; G_\phi^\tau(\mathbf{x}_0, \mathbf{x}_k))$ for all masked $i$
6:     *// Setting $\alpha = 0$ recovers standard DLM training*
7:     $\mathcal{L}_{\text{PAPL}} \leftarrow -\sum \frac{1}{L-k}(1 + \alpha w^i) \log \text{Cat}(x_0^i; D_\theta^i(\mathbf{x}_k))$
8:     Update parameters $\theta$ using $\nabla_\theta \mathcal{L}_{\text{PAPL}}$
9: **end while**
10: **return** Trained denoiser $D_\theta$

---

efficient algorithm that still falls within our theoretical framework of training under planning.

**Soft greedy planner**. We relax the deterministic argmax in Equation (6) into a softmax:

$$\text{Cat}(j; G_\phi^\tau(\mathbf{z}, \mathbf{x})) \propto \exp\left( \frac{1}{\tau} \log \text{Cat}\left( z^j; D_\theta^j(\mathbf{x}) \right) \right),$$

where $\tau$ is the softmax temperature. This assigns higher weight to positions where the denoiser is confident, and reduces to uniform sampling as $\tau \uparrow \infty$ and greedy sampling as $\tau \downarrow 0$. Specializing Prop. 3.2 to $G_\phi^\tau$ yields a planner-weighted cross-entropy plus a complex correction term as outlined in Corollary A.10. Detaching gradients through the planner removes the correction term, leaving only the simple weighted cross-entropy. This makes the objective stable and efficient.

**Stabilization**. Instead of simulating planner-driven paths to sample $\mathbf{x}_k$, we reuse the vanilla DLM masking scheme: mask $L - k$ positions uniformly at random. This keeps training as cheap as standard DLM. In practice, the pure weighted loss can have high variance, as shown in the training curves Fig. 5. We stabilize this by interpolating with the vanilla DLM loss. The resultant effect is that the uniform weights $1/(L - k)$ are replaced with planner-adjusted weights: $\frac{1}{L-k}(1 + \alpha w^i)$, where the weights are $w^i \propto \text{Cat}(i; G_\phi^\tau(\mathbf{x}_0, \mathbf{x}_k))$ and $\alpha$ controls the strength of planner weighting.

**Final objective**. The resulting *Planner-Aware Path Learning (PAPL)* loss is therefore just the standard masked diffusion cross-entropy, augmented with planner weights:

$$\mathcal{L}_{\text{PAPL}}(\theta) = - \mathbb{E}_{\mathbf{x}_0, k, \mathbf{x}_k} \left[ \sum_{i\,:\,x_k^i = \mathbf{m}} \tfrac{1}{L-k}(1 + \alpha w^i) \, \log\left(\text{Cat}(x_0^i; D_\theta^i(\mathbf{x}_k))\right) \right]. \tag{7}$$

This amounts to a one-line modification of the vanilla DLM loss, making PAPL easy to adopt. We detail the connection between the softmax regularized training objective from Corollary A.10 and the PAPL training objective equation 7 in §A.2.3.

## 4  EXPERIMENTS

We evaluate PAPL on protein sequence generation, text and code generation, domains that demand non-trivial structure. Across domains, we compare against (i) the masked diffusion language model (DLM) baseline (same architecture, size, and training setup as PAPL), (ii) prior autoregressive and diffusion-based models (cf. §B for details).

### 4.1  PROTEIN SEQUENCE GENERATION

**Setup**. We evaluate PAPL on the task of protein sequence generation. We train a 150M DLM baseline and a PAPL-augmented variant under identical configurations. During inference, both models use P2 sampling. For evaluation, each model generates 100 sequences at lengths $200, 300, \ldots, 800$, which are folded into 3D structures using ESMFold (Lin et al., 2023). Structural quality is measured by pLDDT, pTM, and pAE; diversity is measured by token-level entropy and sequence uniqueness. To obtain a single robust metric of functionality, we define a sequence as foldable if it simultaneously satisfies pLDDT $> 80$, pTM $> 0.7$, and pAE $< 10$. Comparisons include diffusion-based

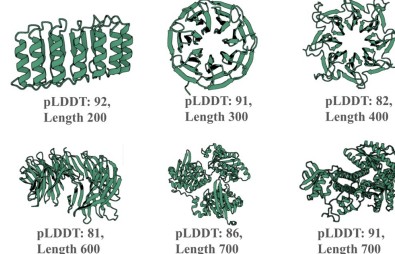

pLDDT: 92, Length 200  pLDDT: 91, Length 300  pLDDT: 82, Length 400

pLDDT: 81, Length 600  pLDDT: 86, Length 700  pLDDT: 91, Length 700

Figure 2: Visualization of PAPL generated proteins folded with ESMFold.

baselines (EvoDiff (Alamdari et al., 2023), DPLM (Wang et al., 2024)) and autoregressive baselines (ESM3 (Hayes et al., 2025), ProGen2 (Nijkamp et al., 2023)). See §B.1 for further details.

Table 1: Protein sequence generation benchmark. We evaluate structure quality via pLDDT, pTM, and pAE, and diversity via token entropy and sequence uniqueness. Foldability is the percentage of sequences satisfying pLDDT $> 80$, pTM $> 0.7$, and pAE $< 10$.

| | Model | pLDDT↑ | pTM↑ | pAE↓ | Foldability (%)↑ | Entropy↑ | Diversity (%)↑ |
|---|---|---|---|---|---|---|---|
| Large | ESM3 | 34.13 | 0.23 | 24.65 | 1.50 | 3.99 | **93.44** |
| | ProGen2-medium | 57.94 | 0.38 | 20.81 | 12.75 | 2.91 | 91.45 |
| | ProGen2-large | 55.07 | 0.35 | 22.00 | 11.87 | 2.73 | 91.48 |
| | DPLM-650M | 79.53 | 0.66 | 11.85 | 49.14 | 3.18 | 92.22 |
| 150M-scale | EvoDiff | 31.84 | 0.21 | 24.76 | 0.43 | **4.05** | 93.19 |
| | ProGen2-small | 49.38 | 0.28 | 23.38 | 4.48 | 2.55 | 89.31 |
| | DPLM-150M | 80.23 | 0.65 | 12.07 | 48.14 | 3.14 | 92.80 |
| | DLM-150M | 81.32 | 0.65 | 12.00 | 42.43 | 3.21 | 92.45 |
| | **DLM-150M + PAPL (ours)** | **81.48** | **0.72** | **8.97** | **59.40** | 3.12 | 91.73 |

**Results**. Table 1 reports quantitative results. Compared to the DLM-150M baseline, PAPL achieves higher pLDDT (81.48 vs. 81.32), stronger pTM (0.72 vs. 0.65), lower pAE (8.97 vs. 12.00), and yields a 40% relative increase in foldability (59.40% vs. 42.43%). Importantly, entropy (3.12) and diversity (91.73%) remain on par with the baseline, confirming that improved folding does not induce collapse. PAPL-trained models outperform EvoDiff and ESM3 in all structural metrics, even when compared to DPLM-650M and ProGen2-2.7B. Figure 2 visualizes 3D folds of sequences generated by PAPL, which exhibit well-formed helices and sheets with coherent tertiary organization.

**Takeaway**. By aligning the training objective with the inference-time planner, PAPL significantly improves the structural plausibility of generated proteins. This is achieved under identical model configurations and without sacrificing diversity, outperforming all baselines.

## 4.2 TEXT GENERATION

**Setup**. We evaluate PAPL on unconditional text generation using the OPENWEBTEXT (Gokaslan & Cohen, 2019) (OWT) corpus, a large-scale collection of web pages designed to replicate the distribution of OpenAI's WebText. Text is tokenized with the GPT-2 byte-pair tokenizer, and sequences are wrapped to a maximum length of $L = 1024$ tokens. We compare against both autoregressive and diffusion-based baselines, including an autoregressive GPT-style language model, standard MDLMs, and MDLMs equipped with forward–backward (FB) and discrete flow matching (DFM) correctors. All checkpoints are reused from prior work to ensure comparability. In inference, we generate 5,000 sequences using planner-based decoding with P2 sampling (Peng et al., 2025a). We evaluate generation quality and diversity primarily with MAUVE (Pillutla et al., 2021), which measures the divergence between generated and reference distributions. As secondary metrics, we also report generative perplexity (Gen PPL) and entropy.

Table 2: Unconditional text generation. For each sampling step $T$, we report MAUVE (higher is better), generative perplexity (Gen PPL; lower is better), and entropy (higher is better). † indicates nucleus sampling. For each $T$, the best diffusion scores are **bolded**.

| | $T = 32$ | | | $T = 64$ | | | $T = 128$ | | |
|---|---|---|---|---|---|---|---|---|---|
| Method | MAUVE | PPL | Ent. | MAUVE | PPL | Ent. | MAUVE | PPL | Ent. |
| Data (ref.) | 1.00 | 14.8 | 5.44 | 1.00 | 14.8 | 5.44 | 1.00 | 14.8 | 5.44 |
| AR† | 0.760 | 1.21 | 5.22 | 0.760 | 1.21 | 5.22 | 0.760 | 1.21 | 5.22 |
| MDLM† | 0.006 | 100.45 | **5.60** | 0.011 | 72.08 | **5.55** | 0.015 | 61.5 | **5.52** |
| MDLM+FB† | 0.007 | 95.76 | 5.56 | 0.016 | 59.05 | 5.49 | 0.064 | 42.8 | 5.44 |
| MDLM+DFM† | 0.004 | 303.8 | 5.31 | 0.007 | 108.8 | 5.33 | 0.041 | 37.9 | 5.31 |
| ReMDM† | 0.007 | 93.53 | 5.58 | 0.016 | 60.38 | 5.51 | 0.057 | 42.5 | 5.43 |
| PAPL† (**ours**) | **0.013** | **40.19** | 5.32 | **0.046** | **29.98** | 5.24 | **0.067** | **24.33** | 5.16 |

**Results**. Table 2 reports unconditional text generation performance across sampling steps $T \in \{32, 64, 128\}$. Our method, PAPL, consistently and substantially improves diffusion-based generation. At $T = 128$, PAPL attains the strongest diffusion MAUVE (0.067) and lowest Gen PPL (24.3), outperforming ReMDM (0.057 MAUVE, 42.5 PPL) and MDLM+DFM (0.041 MAUVE, 37.9 PPL). entropy remains comparable across all methods (5.1–5.6), indicating that PAPL's gains in quality and perplexity are not driven by mode collapse.

**Takeaways**. PAPL delivers consistent and significant improvements over prior discrete diffusion models across all sampling budgets. These gains hold under fast sampling regimes ($T < L$). While diffusion approaches still trail autoregressive models in absolute quality, PAPL markedly reduces this gap without sacrificing diversity.

## 4.3 CODE GENERATION

**Setup.** We evaluate PAPL on code generation, a domain requiring both syntactic validity and semantic correctness. Following the Open-dLLM framework (Peng et al., 2025b), we initialize from Qwen2.5-Coder and adapt it to the diffusion setting with bidirectional attention. Models are trained on the FineCode corpus, which combines algorithmic and QA-style data, and are optimized with a masked cross-entropy loss as in the Open-dLLM recipe. Evaluation covers HUMANEVAL (Chen et al., 2021), MBPP (Austin et al., 2021b), and their augmented variants, as well as HUMANEVAL-INFILL and the Python

Table 3: Code infilling performance on HUMANEVAL-INFILL PASS@1 and SANTACODER-FIM EXACT MATCH. Large-scale models ($\geq$7B) are shown as reference, while the main comparison is among compact sub-billion models.

| Model | HumanEval | SantaCoder |
|---|---|---|
| **Reference Models ($\geq$7B)** | | |
| LLaDA-8B | 48.3 | 35.1 |
| Dream-7B | 39.4 | 40.7 |
| DiffuCoder-7B | 54.8 | 38.8 |
| Dream-Coder-7B | 55.3 | 40.0 |
| **Compact Models ($\leq$1B)** | | |
| DLM (0.5B) | 30.0 | 30.7 |
| DLM (0.5B) + PAPL (**Ours**) | **32.5** | **32.3** |
| DLM (0.5B) + PAPL (Oracle length) | 77.4 | 56.4 |

subset of SANTACODER-FIM. We report pass@1 and pass@10 for completion tasks, and exact match for infilling, using official protocols. See Section B.3.1 for more details.

Table 4: Code generation performance on HUMANEVAL, HUMANEVAL+, MBPP, and MBPP+. Large-scale models ($\geq$7B) are shown as reference, while the main comparison is among compact sub-billion models. Results marked with $\dagger$ are adopted from prior work (Havasi et al., 2025).

| Model | HUMANEVAL | | HUMANEVAL+ | | MBPP | | MBPP+ | |
|---|---|---|---|---|---|---|---|---|
| | Pass@1 | Pass@10 | Pass@1 | Pass@10 | Pass@1 | Pass@10 | Pass@1 | Pass@10 |
| **Reference Models ($\geq$7B)** | | | | | | | | |
| LLaDA (8B) | 35.4 | 50.0 | 30.5 | 43.3 | 38.8 | 53.4 | 52.6 | 69.1 |
| Dream (7B) | 56.7 | 59.2 | 50.0 | 53.7 | 55.4 | 56.2 | 71.5 | 72.5 |
| **Compact Models ($\leq$1B)** | | | | | | | | |
| Autoregressive$^\dagger$ (1.3B) | 17.0 | 34.7 | 14.0 | 28.6 | **25.6** | **45.4** | – | – |
| Mask DFM (1.3B)$^\dagger$ | 9.1 | 17.6 | 7.9 | 13.4 | 6.2 | 25.0 | – | – |
| Edit Flow (1.3B)$^\dagger$ | 12.8 | 24.3 | 10.4 | 20.7 | 10.0 | 36.4 | – | – |
| Uniform $X_0$ + Edit Flow (1.3B)$^\dagger$ | 9.7 | 24.3 | 9.7 | 19.5 | 9.4 | 33.4 | – | – |
| DLM (0.5B) | 18.5 | 31.1 | 17.5 | 28.0 | 17.6 | 32.6 | 17.6 | 32.6 |
| DLM (0.5B) + PAPL (**Ours**) | **20.8** | **38.4** | **17.6** | **35.2** | 16.7 | 38.4 | **23.9** | **53.6** |

**Results**. Table 4 shows that PAPL consistently outperforms the baseline diffusion model across all completion benchmarks. On HUMANEVAL, pass@1 improves from 18.5 to 20.8 and pass@10 from 31.1 to 38.4. Similar gains are observed on HUMANEVAL+ and MBPP+, where PAPL improves pass@10 by more than +10 points. This disproportionate improvement at pass@10 suggests that PAPL is not just improving the single best prediction, but is learning a more robust generative distribution over the entire solution space, making it highly effective at generating a diverse set of high-quality candidates. Table 3 reports results on infilling. Here too, PAPL improves over the baseline: pass@1 increases from 30.0 to 32.5 on HUMANEVAL-INFILL, and exact-match accuracy rises from 30.7 to 32.3 on SANTACODER-FIM. These improvements hold under identical configurations and inference settings, confirming that planner-aware training better aligns the denoiser with the planner-based reverse process.

**Takeaway**. Across both completion and infilling tasks, PAPL consistently improves the correctness of code generation. These results highlight the generality of our approach: learning better generative paths benefits tasks requiring strict logical structure just as it does those requiring biological fidelity.

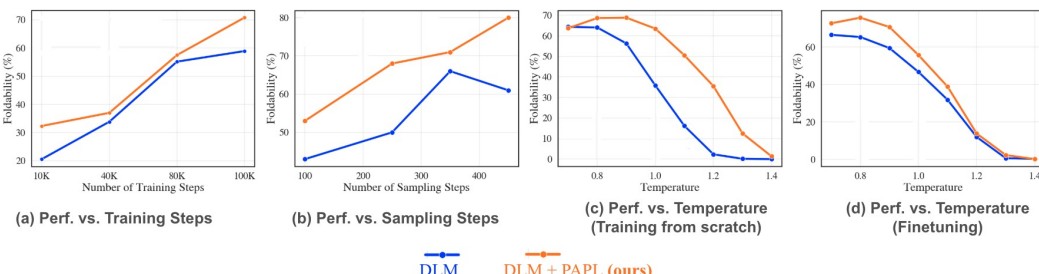

(a) Perf. vs. Training Steps    (b) Perf. vs. Sampling Steps    (c) Perf. vs. Temperature (Training from scratch)    (d) Perf. vs. Temperature (Finetuning)

DLM    DLM + PAPL (**ours**)

Figure 3: PAPL consistently improves over DLM across training, sampling steps, and temperature. (a) Faster convergence in training steps. (b) Higher performance across sampling steps. (c) More robust to temperature when training from scratch. (d) More robust to temperature when fine-tuning.

## 4.4 ABLATION

**Head-to-head comparison.** We compare PAPL with the vanilla DLM baseline across training and inference conditions. Figure 3 shows that PAPL converges faster during training (panel a), maintains stronger performance across different sampling steps (panel b), and is substantially more robust to temperature variation, both when trained from scratch and when fine-tuned (panels c–d). These results confirm that PAPL improves final quality while also stabilizing optimization and inference dynamics.

**Hyperparameter tuning.** We study the two key components introduced by PAPL: the softmax temperature $\tau$ and the path learning weight $\alpha$. As shown in Fig. 4, lowering $\tau$ below the default ($\tau = 1$) consistently improves foldability, while larger values hurt performance, suggesting that sharper planner distributions provide more effective supervision. Increasing $\alpha$ strengthens performance up to $\alpha = 5$, demonstrating that emphasizing planner-weighted paths enhances stability and final quality. Increasing $\alpha$ beyond 5 on this task lowers performance indicating the usefulness of interpolating

between MDLM and PAPL losses. Runs for these weights were stopped early to save compute and so do not appear in Fig. 4.

## 5 RELATED WORK

Masked diffusion language models (DLMs) have recently emerged as promising alternatives to autoregressive models for discrete sequence generation (Sahoo et al., 2024; Shi et al., 2024; Nie et al., 2025a; Gong et al., 2025). A large body of work has focused on improving sampling efficiency and quality through heuristic strategies such as greedy unmasking (Gong et al., 2025), iterative remasking (Zheng et al., 2023; Wang et al., 2024), informed correctors (Zhao et al., 2025), and planner-guided

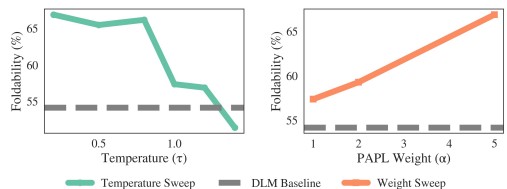

Figure 4: **Effect of $\tau$ and $\alpha$ on foldability.** Lower $\tau$ ($< 1$) improves performance. Increasing $\alpha$ steadily boosts foldability up to $\alpha = 5$. The dashed line denotes the vanilla DLM baseline.

approaches like P2 (Peng et al., 2025a) and confidence-planning (). While effective, these methods assume models trained under a uniform denoising order and modify the sampling path only at inference time, creating a mismatch between training and generation.

A related line of work investigates generation order in the setting of *any-order autoregressive models* (AOARMs). Shih et al. (2022) propose restricting the model to a fixed family of orders to reduce redundancy, whereas LO-ARM (Wang et al., 2025c) and (Li et al., 2021) treat generation order as a latent variable and learn it with REINFORCE (Williams, 1992). However, the reliance on high-variance policy gradient estimators in LO-ARM limits its scalability to large models and datasets. DDPD (Liu et al., 2025) trains for a planner-selected denoising order, where the planner is effectively a uniform discrete diffusion model. This methodology requires a planner of similar or greater size to the denoiser, and hence suffers from a similar pitfall to its AOARM counterparts.

## 6 CONCLUSION

In this work, we investigate the role of planning at inference time with respect to denoiser training for DLMs. Through this, we identify a mismatch in popular planner-guided inference paths and standard denoiser training that uses uniformly at random masking. We propose a new Planner-Aware Evidence Lower Bound (P-ELBO) and developed a practical algorithm, Planner-Aware Path Learning (PAPL), to align the denoiser training with its intended use at inference. We demonstrate that the P-ELBO recovers all current planning strategies, and in particular enjoys straightforward implementation with PAPL being a single line code change with no additional computational overhead from standard DLM training. Empirically, PAPL achieves a 40% relative increase in protein foldability, a $4\times$ MAUVE gain in text generation, and a 23% relative improvement in code generation on HumanEval. demonstrating the benefits of reconciling the training and sampling processes. While PAPL uses the denoiser's own confidence to plan, there are many other possible planning functions presenting ripe directions to extend and generalize the PAPL algorithm. We discuss how other unmasking schemes such as "top probability margin Kim et al. (2025), RDM Zheng et al. (2023), Top-k "block-denoising" Nie et al. (2025b), and "Confidence Thresholding" Wu et al. (2025) can fit into a properly adapted version of our mathematical framework in §A.2 and §A.5. Expanding this analysis to find a computationally viable loss analogous to what the PAPL loss of equation 7 provides for greedy ancestral is an interesting avenue for future work. Finally, we remark that our papers strength is also its limitation: while we show that one should modify the ELBO in order to account for alternative decoding strategies in MDMs, this means that some amount of post-training is necessary in order to test whether the decoding strategies performance can be improved via using a planner-aware loss. This training overhead makes our methodology expensive to implement with large planning models compared to just testing performance at inference time.

## ACKNOWLEDGMENTS

Fred extends sincere gratitude to Kaiwen Zheng for his invaluable insights. Zack extends his gratitude to Jim Nolen for his support and insightful discussions. The authors acknowledge funding from UNIQUE, CIFAR, NSERC, Intel, Samsung, as well as the Hartwell Foundation and CHDI Foundation. The research was enabled in part by computational resources provided by the Digital Research Alliance of Canada (https://alliancecan.ca), Mila (https://mila.quebec), and NVIDIA. This research is partially supported by the EPSRC Turing AI World-Leading Research Fellowship No. EP/X040062/1 and EPSRC AI Hub No. EP/Y028872/1. Z.B. is partially supported by NSF-DMS award 2038056. F.Z.P. and A.R.Z. are partially supported by NIH R01HL169347.

## ETHICS STATEMENT

This work introduces Planner Aware Path Learning (PAPL), a training framework for discrete diffusion models. Our experiments are limited to publicly available datasets in code and biological sequence generation. All results are computational; no biological synthesis or wet-lab experiments were performed. While improved generative models may be misused (e.g., generating harmful biological sequences or insecure code), we explicitly discourage such applications and release models only with documentation of intended use and limitations.

## REPRODUCIBILITY STATEMENT

We provide detailed descriptions of model architectures, training objectives, datasets, and evaluation protocols in the main text and appendix. Hyperparameters, training schedules, and implementation details are included to enable replication. Code and pretrained models will be released upon publication to support full reproducibility.

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

APPENDICES

# A  THEORETICAL DERIVATIONS AND PROOFS

## A.1  FOUNDATIONAL DERIVATIONS

### A.1.1  PROPERTIES OF KL DIVERGENCE

Recall for $p, q \in \Delta^{|\mathcal{X}|}$ for $\mathcal{X}$ some finite set, we define

$$D_{KL}(p||q) := \sum_{x \in \mathcal{X}} p(x) \log \left( \frac{p(x)}{q(x)} \right)$$

when $p(x) = 0$ for every $x \in \mathcal{X}$ such that $q(x) = 0$ and $+\infty$ otherwise. Also recall that for $x$ such that $p(x) = 0$, we interpret $0 \log(0) = 0$. Here we will recall some basic properties of $D_{KL}$ that will aid in our proof.

**Lemma A.1.** *Non-negativity of KL-Divergence: For any $p, q$ distributions on a finite set $\mathcal{X}$,*

$$D_{KL}(p||q) \geq 0,$$

*with $D_{KL}(p||q) = 0$ if and only if $p = q$.*

*Proof.* First we observe that that $g : [0, \infty) \to \mathbb{R}$ given by $g(t) = t \log(t) - t + 1$ has $g'(t) = \log(t)$, $g'(t) < 0$ for $t \in (0, 1)$ and $g'(t) > 0$ for $t > 1$, so $g$ has a global minimum of 0 at $t = 1$. Then:

$$
\begin{aligned}
D_{KL}(p||q) &= \sum_{x \in \mathcal{X}} p(x) \log \left( \frac{p(x)}{q(x)} \right) \\
&= \sum_{x \in \mathcal{X}} q(x) \frac{p(x)}{q(x)} \log \left( \frac{p(x)}{q(x)} \right) + 1 - 1 \\
&= \sum_{x \in \mathcal{X}} q(x) \frac{p(x)}{q(x)} \log \left( \frac{p(x)}{q(x)} \right) + \sum_{x \in \mathcal{X}} q(x) - \sum_{x \in \mathcal{X}} p(x) \\
&= \sum_{x \in \mathcal{X}} q(x) \left( \frac{p(x)}{q(x)} \log \left( \frac{p(x)}{q(x)} \right) - \frac{p(x)}{q(x)} + 1 \right) \\
&= \sum_{x \in \mathcal{X}} q(x) g \left( \frac{p(x)}{q(x)} \right) \\
&\geq \sum_{x \in \mathcal{X}} q(x) * 0,
\end{aligned}
$$

with equality holding if and only if $q(x) = p(x), \forall x \in \mathcal{X}$. $\qquad \square$

**Lemma A.2.** *Chain Rule for KL Divergence between Joint Law of 2 Discrete Random Variables:*

*For $p, q$ distributions on $\mathcal{X} \times \mathcal{Y}$, where $\mathcal{X}$ and $\mathcal{Y}$ are finite sets, denoting by $p_{\mathcal{X}}, q_{\mathcal{X}}$ the $\mathcal{X}$ marginals of $p$ and $q$ respectively and by $p_{\mathcal{Y}|\mathcal{X}}(y|x) = \frac{p(x,y)}{p_{\mathcal{X}}(x)}$ and similarly for $q_{\mathcal{Y}|\mathcal{X}}(y|x)$:*

$$D_{KL}(p||q) = D_{KL}(p_{\mathcal{X}}||q_{\mathcal{X}}) + \mathbb{E}_{x \sim p_{\mathcal{X}}} \left[ D_{KL}(p_{\mathcal{Y}|\mathcal{X}}(\cdot|x)||q_{\mathcal{Y}|\mathcal{X}}(\cdot|x)) \right].$$

*Proof.* By definition:

$$
\begin{aligned}
D_{KL}(p||q) &= \sum_{(x,y) \in \mathcal{X} \times \mathcal{Y}} p(x,y) \log \left( \frac{p(x,y)}{q(x,y)} \right) \\
&= \sum_{(x,y) \in \mathcal{X} \times \mathcal{Y}} p(x,y) \log \left( \frac{p_{\mathcal{Y}|\mathcal{X}}(y|x) p_{\mathcal{X}}(x)}{q_{\mathcal{Y}|\mathcal{X}}(y|x) q_{\mathcal{X}}(x)} \right) \\
&= \sum_{(x,y) \in \mathcal{X} \times \mathcal{Y}} p(x,y) \log \left( \frac{p_{\mathcal{X}}(x)}{q_{\mathcal{X}}(x)} \right) + \sum_{(x,y) \in \mathcal{X} \times \mathcal{Y}} p(x,y) \log \left( \frac{p_{\mathcal{Y}|\mathcal{X}}(y|x)}{q_{\mathcal{Y}|\mathcal{X}}(y|x)} \right)
\end{aligned}
$$

$$= \sum_{x \in \mathcal{X}} p_{\mathcal{X}}(x) \log \left( \frac{p_{\mathcal{X}}(x)}{q_{\mathcal{X}}(x)} \right) + \sum_{x \in \mathcal{X}} p_{\mathcal{X}}(x) \sum_{y \in \mathcal{Y}} p_{\mathcal{Y}|\mathcal{X}}(y|x) \log \left( \frac{p_{\mathcal{Y}|\mathcal{X}}(y|x)}{q_{\mathcal{Y}|\mathcal{X}}(y|x)} \right)$$

$$= D_{KL}(p_{\mathcal{X}}||q_{\mathcal{X}}) + \sum_{x \in \mathcal{X}} p_{\mathcal{X}}(x) D_{KL}(p_{\mathcal{Y}|\mathcal{X}}(\cdot|x)||q_{\mathcal{Y}|\mathcal{X}}(\cdot|x))$$

$$= D_{KL}(p_{\mathcal{X}}||q_{\mathcal{X}}) + \mathbb{E}_{x \sim p_{\mathcal{X}}} \left[ D_{KL}(p_{\mathcal{Y}|\mathcal{X}}(\cdot|x)||q_{\mathcal{Y}|\mathcal{X}}(\cdot|x)) \right].$$

$\square$

**Corollary A.3.** *Chain Rule for KL Divergence between Joint Law of N Discrete Random Variables:*

*For $N \in \mathbb{N}$ and $p, q$ distributions on $\mathcal{X}_0 \times \mathcal{X}_1 \times \ldots \mathcal{X}_N$ where $\mathcal{X}_0, \ldots, \mathcal{X}_N$ are finite sets, denoting for $k \in \{0, 1, \ldots, N\}$ $p_{0:k}$ the $\mathcal{X}_0 \times \cdots \times \mathcal{X}_k$ marginal of $p$ and similarly for $q_{0:k}$, and by $p_{k+1|0:k}(x_{k+1}|x_0, \ldots, x_k) = \frac{p_{k+1}(x_0, \ldots, x_k, x_{k+1})}{p_k(x_0, \ldots, x_k)}$ for $x_i \in \mathcal{X}_i, i = 0, 1, \ldots, k$ and similarly for $q_{k+1|0:k}$, we have:*

$$D_{KL}(p||q) = D_{KL}(p_0||q_0)$$
$$+ \sum_{k=0}^{N-1} \mathbb{E}_{(x_0, x_1, \ldots, x_k) \sim p_{0:k}} \left[ D_{KL}(p_{k+1|0:k}(\cdot|x_0, \ldots, x_k)||q_{k+1|0:k}(\cdot|x_0, \ldots, x_k)) \right]$$

*Proof.* This follows from iteratively applying Lemma A.2. $\square$

**Corollary A.4.** *Inequality for Marginalization of Discrete Distributions:*

*For $p, q$ distributions on $\mathcal{X} \times \mathcal{Y}$, where $\mathcal{X}$ and $\mathcal{Y}$ are finite sets, denoting by $p_{\mathcal{X}}, q_{\mathcal{X}}$ the $\mathcal{X}$ marginals of $p$ and $q$ respectively:*

$$D_{KL}(p||q) \geq D_{KL}(p_{\mathcal{X}}||q_{\mathcal{X}})$$

*Proof.* This follows from Lemma A.2 via noting that the term inside the expectation is non-negative for each $y$ via Lemma A.1. $\square$

### A.1.2 DISCRETE TIME MARKOV CHAINS

**Definition A.5.** Discrete Time Markov Chains: A discrete time Markov chain on finite state space $\mathcal{S}$ is a sequence of random variables $\{X_k\}_{k \in \mathbb{N}}$ such that for any $k \in \mathbb{N}$ and $x_0, x_1, \cdots, x_{k-2}, y, x \in \mathcal{S}$:

$$\mathbb{P}(X_k = x | X_{k-1} = y, X_{k-2} = x_{k-2}, \ldots, X_1 = x_1, X_0 = x_0) = \mathbb{P}(X_k = x | X_{k-1} = y). \quad (8)$$

The distribution of a path of length $k$ $(X_0, X_1, \ldots, X_k) \in \mathcal{S}^{k+1}$ of a Markov chain $\{X_k\}_{k \in \mathbb{N}}$ at any time is entirely determined by its one step transition probabilities, which we encode into its transition matrix:

$$Q_k(x, y) = \mathbb{P}(X_{k+1} = x | X_k = y), \quad x, y \in \mathcal{V}, k \in \mathbb{N}. \quad (9)$$

In the case where $\mathbb{P}(X_k = x | X_{k-1} = y) = \mathbb{P}(X_1 = x | X_0 = y)$ for all $k \in \mathbb{N}$, i.e. when the transition matrix does not depend on the time $k$, we say the chain is time-homogeneous.

**Proposition A.6.** *KL Divergence Between Paths of Discrete Time Markov Chains:*

*Let $\mathcal{R}, \mathcal{P}$ be probability distributions on $\mathcal{S}^{N+1}$ corresponding to the distribution of paths of length $N$ of two discrete time Markov chains $Y$ and $X$ on $\mathcal{S}$ with transition matrices $R$ and $Q$ respectively. Also suppose that $Y_0 \sim \mu$ and $X_0 \sim \nu$ for some $\mu, \nu \in \Delta^{|\mathcal{S}|}$. Then:*

$$D_{KL}(\mathcal{R}||\mathcal{P}) = D_{KL}(\mu||\nu) + \sum_{k=0}^{N-1} \mathbb{E}_{x_k \sim r_k} \left[ \sum_{y \in S} R_k(y, x_k) \log \left( \frac{R_k(y, x_k)}{Q_k(y, x_k)} \right) \right],$$

*where $r_k \in \Delta^{|\mathcal{S}|}$ is given by:*

$$r_k(x) = \mathbb{P}(Y_k = x).$$

*Proof.* By Corollary A.3 (using the same notation as in the statement thereof):

$$D_{KL}(\mathcal{R}||\mathcal{P}) = D_{KL}(\mathcal{R}_0||\mathcal{P}_0)$$
$$+ \sum_{k=0}^{N-1} \mathbb{E}_{(x_0,x_1,\ldots,x_k)\sim\mathcal{R}_{0:k}} \left[ D_{KL}(\mathcal{R}_{k+1|0:k}(\cdot|x_0,x_1,\ldots,x_k)||\mathcal{P}_{k+1|0:k}(\cdot|x_0,x_1,\ldots,x_k)) \right]$$
$$= D_{KL}(\mu||\nu)$$
$$+ \sum_{k=0}^{N-1} \mathbb{E}_{(x_0,x_1,\ldots,x_k)\sim\mathcal{R}_{0:k}} \left[ D_{KL}(\mathbb{P}(Y_{k+1} = \cdot|Y_k = x_k)||\mathbb{P}(X_{k+1} = \cdot|X_k = x_k)) \right]$$

by definition of $\mu, \nu$ and the Markov property equation 8

$$= D_{KL}(\mu||\nu) + \sum_{k=0}^{N-1} \mathbb{E}_{x_k \sim r_k} \left[ D_{KL}(\mathbb{P}(Y_{k+1} = \cdot|Y_k = x_k)||\mathbb{P}(X_{k+1} = \cdot|X_k = x_k)) \right]$$

by definition of $r_k$ and $\mathcal{R}_{0:k}$

$$= D_{KL}(\mu||\nu) + \sum_{k=0}^{N-1} \mathbb{E}_{x_k \sim r_k} \left[ \sum_{y \in S} R_k(y, x_k) \log \left( \frac{R_k(y, x_k)}{Q_k(y, x_k)} \right) \right]$$

by definition of the transition matrix equation 18.

$\square$

### A.1.3 APPLICATION TO THE ELBO

The loss corresponding to an ELBO $\mathcal{E}^\theta(\mathbf{x}_0)$ (i.e. a quantity satisfying $\log p_\theta(\mathbf{x}_0) \geq \mathcal{E}^\theta(\mathbf{x}_0), \forall \mathbf{x}_0 \in \mathcal{V}^L$ and $p_\theta$ the generated data distribution) is always given by

$$\mathcal{L}(\theta) = -\mathbb{E}_{\mathbf{x}_0 \sim p_{data}} \left[ \mathcal{E}^\theta(\mathbf{x}_0) \right],$$

so that

$$D_{KL}(p_{data}||p_\theta) = \sum_{\mathbf{x} \in S^L} p_{data}(\mathbf{x}) \log \left( \frac{p_{data}(\mathbf{x})}{p_\theta(\mathbf{x})} \right)$$
$$= \sum_{\mathbf{x} \in S^L} p_{data}(\mathbf{x}) \log p_{data}(\mathbf{x}) - \sum_{\mathbf{x} \in S^L} p_{data}(\mathbf{x}) \log(p_\theta(\mathbf{x}))$$
$$= -H(p_{data}) - \mathbb{E}_{\mathbf{x}_0 \sim p_{data}} \left[ \log(p_\theta(\mathbf{x}_0)) \right]$$
$$\leq -H(p_{data}) - \mathbb{E}_{\mathbf{x}_0 \sim p_{data}} \left[ \mathcal{E}^\theta(\mathbf{x}_0) \right]$$
$$= -H(p_{data}) + \mathcal{L}(\theta).$$

Here $H(p)$ denotes the Shannon entropy of $p$. Note that, crucially, $p_\theta$ must be the distribution on $\mathcal{V}^L$ which one samples from at inference time, since this is what one wishes to make equal to $p_{data}$ via minimizing $\mathcal{L}(\theta)$ to $H(p_{data})$.

In the following proposition, we show Proposition A.6 and the basic properties of KL divergence from Subsection A.1.1 can be used to derive an ELBO, and hence training loss, for any sampling procedure which can be described via a discrete time Markov chain.

**Proposition A.7.** *Application to ELBO:*

*Suppose $p$ is a distribution on $S$ the given by $p(x) = \mathbb{P}(X_N = x)$ for some $N \in \mathbb{N}$ and $X$ a Markov chain on $S$ with transition matrix $Q$. Then for $x_0 \in S$ and $Y^{x_0}$ any Markov chain with rate matrix $R(\cdot, \cdot; x_0)$ satisfying both*

1. *$Y_0^{x_0}$ is equal in distribution to $X_0$*

2. *$\mathbb{P}(Y_N^{x_0} = x_0) = 1$,*

*we have:*

$$\log(p(x_0)) \geq -\sum_{k=0}^{N-1} \mathbb{E}_{x_k \sim r_k(\cdot;x_0)} \left[ \sum_{y \in S} R_k(y, x_k; x_0) \log \left( \frac{R_k(y, x_k; x_0)}{Q_k(y, x_k)} \right) \right],$$

*where $r_k(\cdot; x_0) \in \Delta^{|S|}$ is given by:*

$$r_k(x; x_0) = \mathbb{P}(Y_k^{x_0} = x).$$

*Proof.* First we observe that:

$$\log(p(x_0)) = -D_{KL}(\delta(x_0)||p) = -D_{KL}(\mathbb{P}(Y_N^{x_0} = \cdot)||\mathbb{P}(X_N = \cdot))$$

by definition. Then, applying Corollary A.4 to $\mathcal{R}(\cdot; x_0), \mathcal{P}$ the distributions on $\mathcal{S}^{N+1}$ corresponding to paths of length $N$ of $Y^{x_0}$ and $X$ respectively, using $\mathcal{X} = \mathcal{S}$ and $\mathcal{Y} = \mathcal{S}^N$:

$$-D_{KL}(\mathbb{P}(Y_N^{x_0} = \cdot)||\mathbb{P}(X_N = \cdot)) \geq -D_{KL}(\mathcal{R}(\cdot; x_0)||\mathcal{P}).$$

Finally, by Proposition A.6:

$$-D_{KL}(\mathcal{R}(\cdot; x_0)||\mathcal{P}) = -D_{KL}(\mathbb{P}(Y_0^{x_0} = \cdot)||\mathbb{P}(X_0 = \cdot))$$
$$-\sum_{k=0}^{N-1} \mathbb{E}_{x_k \sim r_k(\cdot;x_0)} \left[ \sum_{y \in S} R_k(y, x_k; x_0) \log \left( \frac{R_k(y, x_k; x_0)}{Q_k(y, x_k)} \right) \right]$$
$$= -\sum_{k=0}^{N-1} \mathbb{E}_{x_k \sim r_k(\cdot;x_0)} \left[ \sum_{y \in S} R_k(y, x_k; x_0) \log \left( \frac{R_k(y, x_k; x_0)}{Q_k(y, x_k)} \right) \right],$$

where in the last step we used Lemma A.1 and that $\mathbb{P}(Y_0^{x_0} = \cdot) = \mathbb{P}(X_0 = \cdot)$ by assumption. $\square$

### A.1.4 PROOF OF PROPOSITION 3.2: A SIMPLE, DISCRETE TIME PERSPECTIVE

Here we provide a novel, self-contained proof of Proposition 3.2. In particular, this encapsulates the proof of the standard DLM ELBO equation 3 by setting $\text{Cat}(i; G_\phi(\mathbf{z}, \mathbf{x})) = \frac{1}{N_M(\mathbf{x})}$ for all $\mathbf{z}$ and $i$ such that $x^i = \mathbf{m}$. This proof methodology helps elucidate the freedom of choice of reference dynamics, and does not require any of the prerequisite knowledge on continuous time Markov chains as other existing proofs in the discrete diffusion literature.

**Proposition 3.2.** *For any planner $G_\phi$, let $p_\theta^{G_\phi}$ denote the distribution of $\mathbf{x}_L$ obtained via the planner-guided sampling scheme. Then we have the following ELBO:*

$$\log(p_\theta^{G_\phi}(\mathbf{x}_0)) \geq \mathcal{E}^{\theta,\phi}(\mathbf{x}_0) = \mathcal{E}_1^{\theta,\phi}(\mathbf{x}_0) + \mathcal{E}_2^{\theta,\phi}(\mathbf{x}_0),$$

$$\mathcal{E}_1^{\theta,\phi}(\mathbf{x}_0) = L \mathop{\mathbb{E}}_{k \sim \text{Unif}([0:L-1])} \left[ \mathop{\mathbb{E}}_{\mathbf{x}_k \sim r_k^{G_\phi}(\cdot;\mathbf{x}_0)} \left[ \sum_{i=1, x_k^i=\mathbf{m}}^{L} \text{Cat}(i; G_\phi(\mathbf{x}_0, \mathbf{x}_k)) \log \left( \text{Cat}(x_0^i; D_\theta^i(\mathbf{x}_k)) \right) \right] \right]$$

$$\mathcal{E}_2^{\theta,\phi}(\mathbf{x}_0) = -L \mathop{\mathbb{E}}_{k \sim \text{Unif}([0:L-1])} \left[ \mathop{\mathbb{E}}_{\mathbf{x}_k \sim r_k^{G_\phi}(\cdot;\mathbf{x}_0)} \left[ \sum_{i=1, x_k^i=\mathbf{m}}^{L} \text{Cat}(i; G_\phi(\mathbf{x}_0, \mathbf{x}_k)) \log \left( \frac{\text{Cat}(i; G_\phi(\mathbf{x}_0, \mathbf{x}_k))}{F_{\theta,\phi}(\mathbf{x}_k, x_0^i, i)} \right) \right] \right],$$

*where $r_k^{G_\phi}$ is the distribution at time $k$ of a Markov chain with initial data $(\mathbf{m}, \ldots, \mathbf{m})$ and transition rates as in equation 2 but with $1/N_M(\mathbf{x})$ replaced by $G_\phi^i(\mathbf{x}_0, \mathbf{x})$.*

*Proof.* By equation 4, for $\mathbf{x} \in \mathcal{V}^L$, $p_\theta^{G_\phi}(\mathbf{x}) = \mathbb{P}(X_L^{G_\phi,\theta} = \mathbf{x})$ where $X^{G_\phi,\theta}$ is the time homogeneous discrete time Markov chain on $\mathcal{V}^L$ with transition matrix given for $\mathbf{x}, \mathbf{y} \in \mathcal{V}^L$ by:

$$Q^{\theta,\phi}(\mathbf{y}, \mathbf{x}) = \begin{cases} \text{Cat}\left(y^i; D_\theta^i(\mathbf{x})\right) F_{\theta,\phi}(\mathbf{x}, y^i, i), & d_{\text{HAM}}(\mathbf{x}, \mathbf{y}) = 1, x^i \neq y^i, x^i = \mathbf{m} \\ 0, & \text{otherwise} \end{cases}$$

and $\mathbb{P}(X_0^{G_\phi,\theta} = \mathbf{x}) = \text{Cat}(\mathbf{x}; \delta((\mathbf{m}, \ldots, \mathbf{m}))).$

So to obtain an ELBO for $p_\theta^{G_\phi}$ using Proposition A.7, we select any family of transition matrices $R(\cdot, \cdot; \mathbf{x}_0)$ parameterized by $\mathbf{x}_0 \in \mathcal{V}^L$ determining a family Markov chains $Y^{\mathbf{x}_0}$ such that

$$\mathbb{P}(Y_L^{\mathbf{x}_0} = \mathbf{x}_0 | Y_0^{\mathbf{x}_0} = (\mathbf{m}, \dots, \mathbf{m})) = 1. \tag{10}$$

There are infinitely many such choices for the reference dynamics $Y^{\mathbf{x}_0}$, but in order to make the paths of the reference dynamics apriori as close to those of $X^{G_\phi,\theta}$ as possible, we opt to keep the planner $G_\phi$ in the transition probabilities and simply replace $D_\theta(\mathbf{x})$ by $\delta(\mathbf{x}_0)$ in the rate matrix $Q^{\theta,\phi}$. Recalling the definition of $F_{\theta,\phi}$ from equation 5, this yields $R(\cdot, \cdot; \mathbf{x}_0)$ to be the time homogeneous rate matrix $R^{G_\phi}(\cdot, \cdot; \mathbf{x}_0)$ given by, for $\mathbf{x}, \mathbf{y} \in \mathcal{V}^L$:

$$R^{G_\phi}(\mathbf{y}, \mathbf{x}; \mathbf{x}_0) = \begin{cases} \text{Cat}(i; G_\phi(\mathbf{x}_0, \mathbf{x}))\text{Cat}(y^i; \delta(x_0^i)), & d_{\text{HAM}}(\mathbf{x}, \mathbf{y}) = 1, x^i \neq y^i, x^i = \mathbf{m} \\ 0, & \text{otherwise} \end{cases}.$$

Observe that this is the same transition matrix from equation 2 but with $1/N_M(\mathbf{x})$ replaced by $G_\phi^i(\mathbf{x}_0, \mathbf{x})$. Also observe that indeed equation 10 is satisfied, since at each step we simply choose a coordinate $i$ among masked positions of $Y^{\mathbf{x}_0}$ with probability $\text{Cat}(i; G_\phi(\mathbf{x}_0, \mathbf{x}))$ and flip it from $\mathbf{m}$ to $x_0^i$.

Then, inserting these choices into Proposition A.7 and using

$$\log\left(\frac{\text{Cat}(i; G_\phi(\mathbf{x}_0, \mathbf{x}))}{\text{Cat}\left(y^i; D_\theta^i(\mathbf{x})\right) F_{\theta,\phi}(\mathbf{x}, y^i, i)}\right) = -\log\left(\text{Cat}\left(y^i; D_\theta^i(\mathbf{x})\right)\right) - \log\left(\frac{F_{\theta,\phi}(\mathbf{x}, y^i, i)}{\text{Cat}(i; G_\phi(\mathbf{x}_0, \mathbf{x}))}\right)$$

the proof of Proposition 3.2 is complete. $\qquad\square$

### A.1.5 ORM OF THE OPTIMAL PLANNER FOR A FIXED DENOISER: PROOF OF PROPOSITION A.8

Recall that the loss associated to the ELBO from Proposition A.7 is:

$$\mathcal{L}(\theta, \phi) = -\mathbb{E}_{\mathbf{x}_0 \sim p_{\text{data}}}\left[\mathcal{E}^{\theta,\phi}(\mathbf{x}_0)\right]. \tag{11}$$

A natural question is: for a fixed (imperfect) denoiser, what is the optimal form of $G_\phi$ which minimizes this objective? We will show here:

**Proposition A.8.** *Consider the loss $\mathcal{L}(\phi) = -\mathbb{E}_{\mathbf{x}_0 \sim p_{\text{data}}}\left[\mathcal{E}^{\theta,\phi}(\mathbf{x}_0)\right]$ where $\mathcal{E}^{\theta,\phi}(\mathbf{x}_0)$ is as in Proposition A.7 and $D_\theta$ is fixed. Then $\mathcal{L}(\phi)$ is uniquely minimized over $G_\phi$ when, for $\mathbf{x}_0, \mathbf{x}_k \in \mathcal{V}^L$ with $\mathbf{x}_0$ containing no masked tokens and $\mathbf{x}_k$ equal to $\mathbf{x}_0$ in all unmasked positions:*

$$G_\phi^i(\mathbf{x}_0, \mathbf{x}_k) \propto q_{\theta,\phi}^i(x_0^i | \mathbf{x}_k), \tag{12}$$

*for $q_{\theta,\phi}^i$ the transition probabilities of the data generating discrete time Markov chain's dynamics from equation 4.*

That is, our loss finds a planner which is mutually consistent with the denoiser in that it picks at each step a coordinate $i$ to unmask with probability proportional to the probability of denoising coordinate $i$ to $x_0^i$ at the current generation step under the planned path. In short: the optimal planner tends to assign high mass to trajectories whose sequence of single-coordinate updates yields high likelihood of producing the observed $x_0$, which during training is supervised by the data. Observe that equation 12 is a fixed-point equation relating values of $G_\phi^i(\mathbf{x}_0, \mathbf{x}_k)$ to itself and the imperfect denoiser through $F_{\theta,\phi}(\mathbf{x}_k, x_0^i, i)$ of equation 5, so one can not simply *choose* to use this optimal planner and indeed needs to train towards optimality in practice.

*Proof.* To see this, we first observe that for fixed $x_k$, no constraint need be enforced in the relationship between the distributions $G_\phi(\mathbf{z}, \mathbf{x}_k), G_\phi(\bar{\mathbf{z}}, \bar{\mathbf{x}}_k) \in \Delta^L$ when $(\bar{\mathbf{z}}, \bar{\mathbf{x}}_k) \neq (\mathbf{z}, \mathbf{x}) \in \mathcal{V}^{2L}$. This means that minimizing equation 11 is equivalent to minimizing the integrand:

$$\sum_{i=1, x_k^i = m}^{L} \text{Cat}(i; G_\phi(\mathbf{x}_0, \mathbf{x}_k)) \log\left(\frac{\text{Cat}(i; G_\phi(\mathbf{x}_0, \mathbf{x}_k))}{\text{Cat}(x_0^i; D_\theta^i(\mathbf{x}_k))F_{\theta,\phi}(\mathbf{x}_k, x_0^i, i)}\right)$$

$$= D_{\mathrm{KL}}(G_\phi(\mathbf{x}_0, \mathbf{x}_k) || r_{\theta,\phi}(\mathbf{x}_0, \mathbf{x}_k)) + \log(C_{\theta,\phi}(\mathbf{x}_0, \mathbf{x}_k)) \tag{13}$$

for fixed $x_0, x_k$, and $k$, where $r_{\theta,\phi}(\mathbf{x}_0, \mathbf{x}_k) \in \Delta^L$ is given by:

$$\mathrm{Cat}(i; r_{\theta,\phi}(\mathbf{x}_0, \mathbf{x}_k)) \propto \mathrm{Cat}(x_0^i; D_\theta^i(\mathbf{x}_k)) F_{\theta,\phi}(\mathbf{x}_k, x_0^i, i) = q_{\theta,\phi}^i(x_0^i | \mathbf{x}_k),$$

and where $C_{\theta,\phi}$ is its normalizing constant.

Next we observe that, although $C_{\theta,\phi}$ appears to depend implicitly on $G_{\theta,\phi}(\mathbf{x}_0, \mathbf{x}_k)$ through $F_{\theta,\phi}(\mathbf{x}_k, x_0^i, i)$ (recall equation 5) this does not affect the minimization problem. To see this, we observe that

$$F_{\theta,\phi}(\mathbf{x}_k, x_0^i, i) = \prod_{j \neq i}^L \mathrm{Cat}(x_0^j; D_\theta^j(\mathbf{x}_k)) \mathrm{Cat}(i; G_\phi(\mathbf{x}_0, \mathbf{x}_k))$$
$$+ \mathbb{E}_{\mathbf{z} \sim D_\theta(\mathbf{x})} \left[ \mathbb{1}_{\mathbf{z}^{-\mathbf{i}} \neq \mathbf{x}_0^{-\mathbf{i}}} \mathrm{Cat}\left( i; G_\phi(\mathbf{z}^{-i,x_0^i}, \mathbf{x}_k) \right) \right]$$

where for $\mathbf{x} \in \mathcal{V}^L$, $\mathbf{x}^{-i} \in \mathcal{V}^{L-1}$ is $\mathbf{x}$ but with its $i$'th component removed.

Then

$$C_{\theta,\phi}(\mathbf{x}_0, \mathbf{x}_k) = \sum_{i=1}^L \mathrm{Cat}(x_0^i; D_\theta^i(\mathbf{x}_k)) F_{\theta,\phi}(\mathbf{x}_k, x_0^i, i)$$
$$= \prod_{j=1}^L \mathrm{Cat}(x_0^j; D_\theta^j(\mathbf{x}_k)) \left( \sum_{i=1}^L \mathrm{Cat}(i; G_\phi(\mathbf{x}_0, \mathbf{x}_k)) \right)$$
$$+ \sum_{i=1}^L \mathrm{Cat}(x_0^i; D_\theta^i(\mathbf{x}_k)) \mathbb{E}_{\mathbf{z} \sim D_\theta(\mathbf{x})} \left[ \mathbb{1}_{\mathbf{z}^{-\mathbf{i}} \neq \mathbf{x}_0^{-\mathbf{i}}} \mathrm{Cat}\left( i; G_\phi(\mathbf{z}^{-i,x_0^i}, \mathbf{x}_k) \right) \right]$$
$$= \prod_{j=1}^L \mathrm{Cat}(x_0^j; D_\theta^j(\mathbf{x}_k))$$
$$+ \sum_{i=1}^L \mathrm{Cat}(x_0^i; D_\theta^i(\mathbf{x}_k)) \mathbb{E}_{\mathbf{z} \sim D_\theta(\mathbf{x})} \left[ \mathbb{1}_{\mathbf{z}^{-\mathbf{i}} \neq \mathbf{x}_0^{-\mathbf{i}}} \mathrm{Cat}\left( i; G_\phi(\mathbf{z}^{-i,x_0^i}, \mathbf{x}_k) \right) \right].$$

That is, $C_{\theta,\phi}(\mathbf{x}_0, \mathbf{x}_k)$ only depends on $G_\phi(\mathbf{z}, \mathbf{x}_k)$ for $\mathbf{z} \neq \mathbf{x}_0$. Hence, minimizing equation 13 over $G_\phi(\mathbf{x}_0, \mathbf{x}_k)$ is equivalent to minimizing the KL divergence between $G_\phi(\mathbf{x}_0, \mathbf{x}_k)$ and $r_{\theta,\phi}(\mathbf{x}_0, \mathbf{x}_k)$. By Lemma A.1, this occurs precisely when equation 12 holds. □

### A.1.6 DERIVATION OF GENERAL PLANNED TRANSITION PROBABILITIES (EQ. 4)

Recall that the sampling methodology discussed in §3 is as per Alg. 2.

---

**Algorithm 2** Gillespie Sampler with a Planner

1: **Initialize:** $t \leftarrow 0$, $\mathbf{x}_0 \leftarrow (m, \ldots, m)$, planner $G_\phi$, denoiser $D_\theta$
2: **for** $k = 0 : L - 1$ **do**
3:     **Plan** Sample $\mathbf{z} \sim D_\theta(\mathbf{x}_k)$
4:     Sample dimension $i \sim G_\phi(\mathbf{z}, \mathbf{x}_k)$
5:     **Denoise**
6:     $\mathbf{x}_{k+1} \leftarrow \mathbf{x}_k$
7:     $x_{k+1}^i \leftarrow z^i$
8: **end for**
9: **return** $\mathbf{x}_L$
10:

---

Let $p_\theta^{G_\phi} \in \Delta^{d^L}$ denote the distribution on $\mathcal{V}^L$ of a sample obtained via running Alg. 2 , and let $p_{\theta,k+1}^{G_\phi}(\cdot | \mathbf{x}_k) \in \Delta^{d^L}$ denote the distribution of $\mathbf{x}_{k+1}$ given $\mathbf{x}_k$. With abuse of notation, we will also let $p_{\theta,k+1}^{G_\phi}(\cdot, \cdot | \mathbf{x}_k) \in \Delta^{d^{L^2}}$ denote the joint distribution of $\mathbf{x}_{k+1}$ and the $k + 1$'st sample $\mathbf{z}$. Note that

$\mathbf{x}_{k+1}$ may only differ from $\mathbf{x}_k$ in a single coordinate $i$ such that $x_k^i = \mathbf{m}$. So, letting for $\mathbf{x} \in \mathcal{V}^L$, $i \in [1:L]$, $y \in \mathcal{V}$, $\mathbf{x}^{-i,y} \in \mathcal{V}^L$ be equal to $\left[x^0, x^1, \ldots, x^{i-1}, y, x^{i+1}, \ldots x^L\right]$:

$$
\begin{aligned}
p_{\theta,k+1}^{G_\phi}(\mathbf{x}_k^{-i,y}|\mathbf{x}_k) &= \sum_{\mathbf{z} \in \mathcal{V}^L} p_{\theta,k+1}^{G_\phi}(\mathbf{x}_k^{-i,y}, \mathbf{z}|\mathbf{x}_k) \\
&= \sum_{\mathbf{z} \in \mathcal{V}^L : z^i = y} \prod_{j=1}^L \mathrm{Cat}\left(z^j; D_\theta^j(\mathbf{x}_k)\right) \mathrm{Cat}\left(i; G_\phi(\mathbf{z}, \mathbf{x}_k)\right) \\
&= \mathrm{Cat}\left(y; D_\theta^i(\mathbf{x}_k)\right) \sum_{\mathbf{z} \in \mathcal{V}^L} \prod_{j=1}^L \mathrm{Cat}\left(z^j; D_\theta^j(\mathbf{x}_k)\right) \mathrm{Cat}\left(i; G_\phi(\mathbf{z}^{-i,y}, \mathbf{x}_k)\right) \\
&= \mathrm{Cat}\left(y; D_\theta^i(\mathbf{x}_k)\right) F_{\theta,\phi}(\mathbf{x}_k, y, i)
\end{aligned}
$$

where $F_{\theta,\phi}$ is as in equation 5. $p_{\theta,k+1}^{G_\phi}(\mathbf{x}_k^{-i,y}|\mathbf{x}_k)$ is precisely what we denote as $q_{\theta,\phi}^i(y; \mathbf{x}_k)$ in equation 4.

### A.1.7 PROOF OF PROPOSITION 3.1 (GREEDY ANCESTRAL VIOLATES THE VANILLA ELBO)

Continuing with the notation from the previous subsection, letting $p_{\theta,\Sigma}^{G_\phi}(\mathbf{x}, \sigma)$ denote the probability of generating the sample $\mathbf{x}$ along the path $\sigma \in \Sigma^L$, we have:

$$
\begin{aligned}
p_{\theta,\Sigma}^{G_\phi}(\mathbf{x}, \sigma) &= \prod_{k=1}^L p_{\theta,k}^{G_\phi}(\mathbf{x}^{\sigma(<k+1)}|\mathbf{x}^{\sigma(<k)}) \\
&= \prod_{k=1}^L \mathrm{Cat}\left(x^{\sigma(k)}; D_\theta^{\sigma(k)}(\mathbf{x}^{\sigma(<k)})\right) F_{\theta,\phi}(\mathbf{x}^{\sigma(<k)}, x^{\sigma(k)}, \sigma(k)),
\end{aligned}
$$

where here we use the same notation as in equation 1. Thus, we arrive at:

$$
\begin{aligned}
p_\theta^{G_\phi}(\mathbf{x}) &= \sum_{\sigma \in \Sigma^L} p_{\theta,\Sigma}^{G_\phi}(\mathbf{x}, \sigma) \\
&= \sum_{\sigma \in \Sigma^L} \prod_{k=1}^L \mathrm{Cat}\left(x^{\sigma(k)}; D_\theta^{\sigma(k)}(\mathbf{x}^{\sigma(<k)})\right) F_{\theta,\phi}(\mathbf{x}^{\sigma(<k)}, x^{\sigma(k)}, \sigma(k)). \quad (14)
\end{aligned}
$$

Using equation 14 and specializing $G_\phi$ to the case of greedy ancestral sampling, we readily obtain a proof of Proposition 3.1.

**Proposition 3.1.** *For $p_\theta^{greedy}(\mathbf{x}_0)$ defined with $G_\phi$ in equation 6 and $D_\theta$ an imperfect denoiser, we may have*

$$
\log(p_\theta^{greedy}(\mathbf{x}_0)) < \mathcal{E}^{\theta,unif}(\mathbf{x}_0),
$$

*where $\mathcal{E}^{\theta,unif}(\mathbf{x}_0)$ is as in equation 1.*

*Proof.* Since we just need a counterexample to the ELBO property, we may restrict to the case of $L = 2$ and $\mathcal{V} = \{1, 2, \mathbf{m}\}$. To construct an example denoiser in this setting, we only need to define 6 terms:

$$
\begin{aligned}
c_1 &= \mathrm{Cat}\left(1, D_\theta^1(\mathbf{m}, \mathbf{m})\right), & c_2 &= \mathrm{Cat}\left(1, D_\theta^2(\mathbf{m}, \mathbf{m})\right), & c_3 &= \mathrm{Cat}\left(1, D_\theta^1(\mathbf{m}, 1)\right), \\
c_4 &= \mathrm{Cat}\left(1, D_\theta^1(\mathbf{m}, 2)\right), & c_5 &= \mathrm{Cat}\left(1, D_\theta^2(1, \mathbf{m})\right), & c_6 &= \mathrm{Cat}\left(1, D_\theta^2(2, \mathbf{m})\right).
\end{aligned}
$$

Then:

$$
\begin{aligned}
\mathrm{Cat}\left(2, D_\theta^1(\mathbf{m}, \mathbf{m})\right) &= 1 - c_1, & \mathrm{Cat}\left(2, D_\theta^2(\mathbf{m}, \mathbf{m})\right) &= 1 - c_2, & \mathrm{Cat}\left(2, D_\theta^1(\mathbf{m}, 1)\right) &= 1 - c_3, \\
\mathrm{Cat}\left(2, D_\theta^1(\mathbf{m}, 2)\right) &= 1 - c_4, & \mathrm{Cat}\left(2, D_\theta^2(1, \mathbf{m})\right) &= 1 - c_5, & \mathrm{Cat}\left(2, D_\theta^2(2, \mathbf{m})\right) &= 1 - c_6.
\end{aligned}
$$

Note that imperfect denoisers need not be inconsistent, meaning that there is no reason to enforce any relationship between $c_1, \ldots, c_6 \in (0, 1)$.

Let's take for our example $\mathbf{x} = (1, 1)$.

Then, from equation 1:

$$\mathcal{E}^{\theta,\text{unif}}(\mathbf{x}) = \frac{1}{L!} \sum_{\sigma \in \Sigma^L} \sum_{i=1}^{L} \log \left( \text{Cat}(x^{\sigma(i)}; D_\theta^i(\mathbf{x}^{\sigma(<i)}) \right) = \frac{1}{2} \log(c_1 c_2 c_3 c_5).$$

To find $p_\theta^{\text{greedy}}(\mathbf{x})$, we use equation 14:

$$p_\theta^{\text{greedy}}(\mathbf{x}) = \sum_{\sigma \in \Sigma^L} \prod_{i=1}^{L} \text{Cat}\left(x^{\sigma(i)}; D_\theta^{\sigma(i)}(\mathbf{x}^{\sigma(<i)})\right) F_{\theta,\phi}(\mathbf{x}^{\sigma(<i)}, x^{\sigma(i)}, \sigma(i))$$

$$= c_1 c_5 F_{\theta,\phi}((\mathbf{m}, \mathbf{m}), 1, 1) F_{\theta,\phi}((1, \mathbf{m}), 1, 2) + c_2 c_3 F_{\theta,\phi}((\mathbf{m}, \mathbf{m}), 1, 2) F_{\theta,\phi}((\mathbf{m}, 1), 1, 1)$$

$$=: c_1 c_5 d_1 + c_2 c_3 d_2.$$

$d_1, d_2$ will be found as functions of the $c$'s, and we will find $c$'s such that

$$(c_1 c_5 d_1 + c_2 c_3 d_2)^2 < c_1 c_2 c_3 c_5$$

Taking $\log$ of both sides and dividing by 2, the inequality will be shown.

Inserting the definition of $F_{\theta,\phi}$ from equation 5 and the specific choice of $G_\phi$ from equation 6, we have

$$F_{\theta,\phi}((\mathbf{m}, \mathbf{m}), 1, 1) = \mathbb{E}_{z \sim D_\theta^2(\mathbf{m},\mathbf{m})} \left[ \text{Cat}\left(1, G_\phi((1, z), (\mathbf{m}, \mathbf{m}))\right) \right]$$

$$= c_2 \text{Cat}\left(1, G_\phi((1, 1), (\mathbf{m}, \mathbf{m}))\right) + (1 - c_2) \text{Cat}\left(1, G_\phi((1, 2), (\mathbf{m}, \mathbf{m}))\right)$$

$$= c_2 \mathbb{1}_{c_1 > c_2} + (1 - c_2) \mathbb{1}_{c_1 > 1 - c_2}$$

and

$$F_{\theta,\phi}((1, \mathbf{m}), 1, 2) = \mathbb{E}_{z \sim D_\theta^1(1,\mathbf{m})} \left[ \text{Cat}\left(2, G_\phi((z, 1), (1, \mathbf{m}))\right) \right]$$

$$= c_5 \text{Cat}\left(2, G_\phi((1, 1), (1, \mathbf{m}))\right) + (1 - c_5) \text{Cat}\left(2, G_\phi((2, 1), (1, \mathbf{m}))\right)$$

$$= c_5 + (1 - c_5) = 1,$$

so $d_1 = c_2 \mathbb{1}_{c_1 > c_2} + (1 - c_2) \mathbb{1}_{c_1 > 1 - c_2}$. Here $\mathbb{1}$ denotes the indicator function.

Similarly,

$$F_{\theta,\phi}((\mathbf{m}, \mathbf{m}), 1, 2) = \mathbb{E}_{z \sim D_\theta^1(\mathbf{m},\mathbf{m})} \left[ \text{Cat}\left(2, G_\phi((z, 1), (\mathbf{m}, \mathbf{m}))\right) \right]$$

$$= c_1 \text{Cat}\left(2, G_\phi((1, 1), (\mathbf{m}, \mathbf{m}))\right) + (1 - c_1) \text{Cat}\left(2, G_\phi((2, 1), (\mathbf{m}, \mathbf{m}))\right)$$

$$= c_1 \mathbb{1}_{c_1 < c_2} + (1 - c_1) \mathbb{1}_{1 - c_1 < c_2}$$

and $F_{\theta,\phi}((\mathbf{m}, 1), 1, 1) = 1$, so $d_2 = c_1 \mathbb{1}_{c_1 > c_2} + (1 - c_1) \mathbb{1}_{1 - c_1 > c_2}$.

Taking any $c_1, c_2$ such that $c_2 > c_1$ and $1 > c_1 + c_2$, we get $d_1 = 0$ and $d_2 = c_1$. Then

$$(c_1 c_5 d_1 + c_2 c_3 d_2)^2 = c_1^2 c_2^2 c_3^2 < c_1 c_2 c_3 c_5$$

$$\Leftrightarrow c_1 c_2 c_3 < c_5$$

There are many choices here that work. For instance, $c_1 = c_3 = 1/4, c_2 = c_5 = 1/2$, as

$$c_1 c_2 c_4 = 1/32 < 1/2 = c_5.$$

$\square$

Note that this means there are data distributions and denoisers for which

$$\mathcal{L}^{\text{unif}}(\theta) = -\mathbb{E}_{\mathbf{x}_0 \sim p_{\text{data}}} \left[ \mathcal{E}^{\theta,\text{unif}}(\mathbf{x}_0) \right] < -\mathbb{E}_{\mathbf{x}_0 \sim p_{\text{data}}} \left[ \log p_\theta^{\text{greedy}}(\mathbf{x}_0) \right],$$

(recall here the discussion in Subsection A.1.3), so

$$D_{KL}(p_{data} || p_\theta^{\text{greedy}}) = -H(p_{data}) - \mathbf{E}_{\mathbf{x}_0 \sim p_{\text{data}}} \left[ \log p_\theta^{\text{greedy}}(\mathbf{x}_0) \right]$$

$$> -H(p_{data}) + \mathcal{L}^{\text{mask}}(\theta).$$

This means that training to make $\mathcal{L}^{\text{mask}}(\theta)$ small cannot provide any guarantee that $p_\theta^{\text{greedy}}$ is close to $p_{data}$.

A.2 INSTANTIATIONS

Our general P-ELBO recovers familiar training objectives when we plug in specific planners.

**Uniform planner**. If $G_\phi$ selects uniformly among masked tokens—that is, $\text{Cat}(i; G_\phi(\mathbf{z}, \mathbf{x})) = 1/N_M(\mathbf{x})$ for masked $i$ and $0$ otherwise–then planner-based sampling reduces to vanilla ancestral sampling. In this case $G_\phi$ does not depend on $\mathbf{z}$, which makes $\mathcal{E}_2^{\theta,\phi}(\mathbf{x}_0) = 0$. Substituting into Prop. 3.2, we exactly recover the standard DLM ELBO in equation 1.

**Greedy planner**. If $G_\phi$ always selects the most confident position according to the denoiser, as in equation 6, then the sampling path becomes deterministic. The associated ELBO is then as follows:

**Corollary A.9.** *Let $Y_0 = (\mathbf{m}, \dots, \mathbf{m})$, and define recursively for $k = 1, \dots, L$:*

$$j_k = \underset{i: Y_{k-1}^i = \mathbf{m}}{\arg\max} \, Cat(x_0^i; D_\theta^i(Y_{k-1})), \quad Y_k^i = \begin{cases} x_0^{j_k}, & i = j_k, \\ Y_{k-1}^i, & otherwise. \end{cases}$$

*For $p_\theta^{greedy}$ the distribution of $\mathbf{x}_L$ under greedy ancestral sampling,*

$$\log(p_\theta^{greedy}(\mathbf{x}_0)) \geq \mathcal{E}^{\theta,greedy}(\mathbf{x}_0) = L \, \mathbb{E}_{k \sim Unif([0:L-1])} \left[ \sum_{i: Y_k^i = \mathbf{m}} \log Cat(x_0^i; D_\theta^i(Y_k)) \right].$$

Compared to the uniform case in equation 1, the greedy ELBO only accumulates logits along the *greedy path* defined by the denoiser. This highlights the mismatch: the standard DLM objective trains on uniformly random paths, but greedy inference relies on a single deterministic path.

**Soft greedy planner**. We use the soft greedy planner:

$$\text{Cat}\left(j; G_\phi^\tau(\mathbf{z}, \mathbf{x})\right) := \exp\left(\frac{1}{\tau} \log\left(\text{Cat}(z^j; D_\theta^j(\mathbf{x}))\right)\right) / C_\tau(\mathbf{z}, \mathbf{x}) \tag{15}$$

$$C^\tau(\mathbf{z}, \mathbf{x}) := \sum_{i=1, x^i = \mathbf{m}}^{L} \exp\left(\frac{1}{\tau} \log\left(\text{Cat}(z^i; D_\theta^i(\mathbf{x}))\right)\right).$$

as a regularized approximation to the greedy planner 6 in order to motivate, after performing the series of modifications discussed in §3.4, the PAPL training algorithm 1. The ELBO associated to this choice of planner is:

**Corollary A.10.** *For $p_\theta^\tau$ the distribution of $\mathbf{x}_L$ resulting from the planned sampling Algorithm of §3 with $G_\phi = G_\phi^\tau$ as in equation 15, we have:*

$$\log(p_\theta^\tau(\mathbf{x}_0)) \geq \mathcal{E}_1^{\theta,\phi,\tau}(\mathbf{x}_0) + \mathcal{E}_2^{\theta,\phi,\tau}(\mathbf{x}_0),$$

$$\mathcal{E}_1^{\theta,\phi,\tau}(\mathbf{x}_0) = L \underset{k \sim Unif([0:L-1])}{\mathbb{E}} \left[ \underset{\mathbf{x}_k \sim r_k^\tau(\cdot; \mathbf{x}_0)}{\mathbb{E}} \left[ \sum_{i=1, x_k^i = m}^{L} Cat(i; G_\phi^\tau(\mathbf{x}_0, \mathbf{x}_k)) \log\left(Cat(x_0^i; D_\theta^i(\mathbf{x}_k))\right) \right] \right]$$

$$\mathcal{E}_2^{\theta,\phi,\tau}(\mathbf{x}_0) = L \underset{k \sim Unif([0:L-1])}{\mathbb{E}} \left[ \underset{\mathbf{x}_k \sim r_k^\tau(\cdot; \mathbf{x}_0)}{\mathbb{E}} \left[ \underset{\mathbf{z} \sim D_\theta(\mathbf{x}_k)}{\mathbb{E}} \left[ \sum_{i=1, x_k^i = m}^{L} Cat(i; G_\phi^\tau(\mathbf{x}_0, \mathbf{x}_k)) \times \right. \right. \right.$$

$$\left. \left. \left. \times \log\left(\frac{C^\tau(\mathbf{x}_0, \mathbf{x}_k)}{C^\tau(\mathbf{z}^{-i, x_0^i}, \mathbf{x}_k)}\right) \right] \right] \right],$$

*where here we recall the notation $\mathbf{z}^{-i, x_0^i}$ means the $i$'th coordinate of $\mathbf{z}$ is replaced by the $i$'th coordinate of $\mathbf{x}_0$, and $r_k^\tau(\mathbf{x}; \mathbf{x}_0) = \mathbb{P}(Y_k^{\mathbf{x}_0} = \mathbf{x})$ for $Y^{\mathbf{x}_0}$ the discrete time Markov chain with rate matrix equation 16*

$$R^\tau(\mathbf{y}, \mathbf{x}; \mathbf{x}_0) = \begin{cases} Cat(i; G_\phi^\tau(\mathbf{x}_0, \mathbf{x})) Cat(y^i; \delta(x_0^i)), & d_{HAM}(\mathbf{x}, \mathbf{y}) = 1, x^i \neq y^i, x^i = \mathbf{m} \\ 0, & otherwise \end{cases} \tag{16}$$

*and $Y_0^{\mathbf{x}_0} = (\mathbf{m}, \dots, \mathbf{m})$.*

We remark that while this is simply used as an approximation to greedy ancestral sampling for the purposes of this manuscript, soft greedy sampling is also used in practice in, e.g. Wang et al. (2025a)'s "Confidence Based Schedule," so this result is of independent interest as a corrected ELBO to these sampling schemes.

**Other unmasking schemes**. We remark there are other unmasking schemes in the literature for which one obtains an ELBO via our Proposition 3.2. For example, to obtain an ELBO for the "top probability margin" method of Kim et al. (2025), one inserts the choice

$$G_\phi(\mathbf{z}, \mathbf{x}) = \delta\left(\underset{i:x^i=\mathbf{m}}{\arg\max} |\text{Cat}(y; D_\theta^i(\mathbf{x})) - \text{Cat}(\bar{y}; D_\theta^i(\mathbf{x}))|\right),$$

where $y = \arg\max_{j\in\mathcal{V}} \text{Cat}(j; D_\theta^i(\mathbf{x}))$ and $\bar{y} = \arg\max_{j\neq y\in\mathcal{V}} \text{Cat}(j; D_\theta^i(\mathbf{x}))$. As the focus of this work is obtaining a viable objective for use with greedy ancestral sampling, we do not provide expanded details on how to train for this planner user our ELBO.

**Extensions to remasking and denoising multiple positions simultaneously**. So far we assumed that once unmasked, a token remains fixed. In practice, planners such as RDM (Zheng et al., 2023) P2 (Peng et al., 2025a) allow remasking and resampling, in addition to denoising multiple tokens simultaneously. There are also methods which attempt to denoise multiple tokens simultaneously, but do not allow remasking, such as top-k block denoising Nie et al. (2025b) and confidence thresholding Wu et al. (2025). Our proof technique extends naturally to these cases, yielding planner-aware ELBOs of the same form as Prop. 3.2. For completeness, in §A.4 we provide a generalization to P2-style planners and show its specialization to P2-TopK, in addition to discussion how the generalized version of the ELBO could be used for finding training stragies for these other sampling methods.

### A.2.1 PROOF OF COROLLARY A.9 (ELBO FOR GREEDY PLANNER)

Specializing the ELBO from Proposition 3.2 to the case of greedy-ancestral sampling, we set $G_\phi$ to be as in equation 6.

*Proof.* We first observe that inserting the choice of $G_\phi$ from equation 6 into equation 2 in the place of $1/N_M(\mathbf{x})$, $Y^{\mathbf{x}_0}$ becomes deterministic, with dynamics $Y_0 = (\mathbf{m}, \ldots, \mathbf{m})$, and

$$Y_k^i = \begin{cases} x_0^{j_{k-1}}, & i = j_{k-1} \\ Y_{k-1}^i, & \text{otherwise} \end{cases}, \quad k = 1, \ldots, L,$$

$$j_k = \text{argmax}_{i\in[1:L], Y_k^i=\mathbf{m}}\text{Cat}(x_0^i; D_\theta^i(Y_k)), \quad k = 0, \ldots, L.$$

For $\mathcal{E}_1^{\theta,\phi}(\mathbf{x}_0)$, we have by definition $\text{Cat}(i; G_\phi(\mathbf{x}_0, \mathbf{x}_k)) = \text{Cat}(i; \delta(j_k))$, so:

$$\mathcal{E}_1^{\theta,\phi}(\mathbf{x}_0) = \sum_{k=0}^{L-1}\left[\log\left(\text{Cat}(x_0^{j_k}; D_\theta^{j_k}(Y_k))\right)\right]$$

Similarly, for the term $\mathcal{E}_2^{\theta,\phi}(\mathbf{x}_0)$, we have, recalling the definition of $F_{\theta,\phi}$ from equation 5:

$$\mathcal{E}_2^{\theta,\phi}(\mathbf{x}_0) = \sum_{k=0}^{L-1}\log\left(F_{\theta,\phi}(Y_k, x_0^{j_k}, j_k)\right)$$

$$= \sum_{k=0}^{L-1}\log\left(\sum_{\mathbf{z}\in\mathcal{V}^L:\text{Cat}(z^i; D_\theta^i(Y_k))<\text{Cat}(x_0^{j_k}; D_\theta^{j_k}(Y_k)), \forall i\in[1,L], Y_k^i=\mathbf{m}, i\neq j_k}\prod_{i=1}^{L}\text{Cat}(z^i; D_\theta^i(Y_k))\right)$$

$$\geq \sum_{k=0}^{L-1}\log\left(\prod_{i\in[1,L], Y_k^i=\mathbf{m}, i\neq j_k}\text{Cat}(x_0^i; D_\theta^i(Y_k))\right) \text{ by definition of } j_k$$

$$= \sum_{k=0}^{L-1}\sum_{i=1, Y_k^i=\mathbf{m}, i\neq j_k}^{L}\log\left(\text{Cat}(x_0^i; D_\theta^i(Y_k))\right).$$

Summing this expression of $\mathcal{E}_1^{\theta,\phi}$ with this lower bound on $\mathcal{E}_2^{\theta,\phi}$ we have the result of Corollary A.9. $\square$

### A.2.2 PROOF OF COROLLARY A.10 (ELBO FOR SOFTMAX PLANNER)

We now specialize the ELBO found in Proposition 3.2 to a smooth approximation of the greedy ancestral planner from equation 6 - namely, we take $G_\phi = G_\phi^\tau$ as in equation 15.

*Proof.* $\mathcal{E}_1^{\theta,\phi,\tau}$ is simply inserting $G_\phi = G_\phi^\tau$ into $\mathcal{E}^{\theta,\phi}$ from Proposition 3.2.

Now we make a lower bound on $\mathcal{E}_2^{\theta,\phi}(\mathbf{x}_0)$. With this choice of $G_\phi$:

$$\mathcal{E}_2^{\theta,\phi}(\mathbf{x}_0) = -\sum_{k=0}^{L-1}\left[\mathbb{E}_{\mathbf{x}_k \sim r_k^\tau(\cdot;\mathbf{x}_0)}\left[\sum_{i=1,x_k^i=m}^{L} \mathrm{Cat}(i;G_\phi^\tau(\mathbf{x}_0,\mathbf{x}_k))\log\left(\frac{\mathrm{Cat}(i;G_\phi^\tau(\mathbf{x}_0,\mathbf{x}_k))}{F_{\theta,\phi}^\tau(\mathbf{x}_k,x_0^i,i)}\right)\right]\right]$$

where $F_{\theta,\phi}^\tau$ is as in equation 5 with $G_\phi = G_\phi^\tau$. So, by Jensen's inequality:

$$\mathcal{E}_2^{\theta,\phi}(\mathbf{x}_0)$$
$$= -\sum_{k=0}^{L-1}\left[\mathbb{E}_{\mathbf{x}_k \sim r_k^\tau(\cdot;\mathbf{x}_0)}\left[\sum_{i=1,x_k^i=m}^{L} \mathrm{Cat}(i;G_\phi^\tau(\mathbf{x}_0,\mathbf{x}_k))\log\left(\mathrm{Cat}(i;G_\phi^\tau(\mathbf{x}_0,\mathbf{x}_k))\right)\right]\right]$$
$$+\sum_{k=0}^{L-1}\left[\mathbb{E}_{\mathbf{x}_k \sim r_k^\tau(\cdot;\mathbf{x}_0)}\left[\sum_{i=1,x_k^i=m}^{L} \mathrm{Cat}(i;G_\phi^\tau(\mathbf{x}_0,\mathbf{x}_k))\log\left(F_{\theta,\phi}^\tau(\mathbf{x}_k,x_0^i,i)\right)\right]\right]$$
$$\geq -\sum_{k=0}^{L-1}\left[\mathbb{E}_{\mathbf{x}_k \sim r_k^\tau(\cdot;\mathbf{x}_0)}\left[\sum_{i=1,x_k^i=m}^{L} \mathrm{Cat}(i;G_\phi^\tau(\mathbf{x}_0,\mathbf{x}_k))\log\left(\mathrm{Cat}(i;G_\phi^\tau(\mathbf{x}_0,\mathbf{x}_k))\right)\right]\right]$$
$$+\sum_{k=0}^{L-1}\left[\mathbb{E}_{\mathbf{x}_k \sim r_k^\tau(\cdot;\mathbf{x}_0)}\left[\sum_{i=1,x_k^i=m}^{L} \mathrm{Cat}(i;G_\phi^\tau(\mathbf{x}_0,\mathbf{x}_k))\mathbb{E}_{\mathbf{z}\sim D_\theta(\mathbf{x}_k)}\left[\log\left(\mathrm{Cat}\left(i;G_\phi^\tau(\mathbf{z}^{-i,x_0^i},\mathbf{x}_k)\right)\right)\right]\right]\right]$$
$$=\sum_{k=0}^{L-1}\left[\mathbb{E}_{\mathbf{x}_k \sim r_k^\tau(\cdot;\mathbf{x}_0)}\left[\mathbb{E}_{\mathbf{z}\sim D_\theta(\mathbf{x}_k)}\left[\sum_{i=1,x_k^i=m}^{L} \mathrm{Cat}(i;G_\phi^\tau(\mathbf{x}_0,\mathbf{x}_k))\times\right.\right.\right.$$
$$\left.\left.\left.\times\log\left(\frac{\mathrm{Cat}\left(i;G_\phi^\tau(\mathbf{z}^{-i,x_0^i},\mathbf{x}_k)\right)}{\mathrm{Cat}(i;G_\phi^\tau(\mathbf{x}_0,\mathbf{x}_k))}\right)\right]\right]\right]$$
$$=\sum_{k=0}^{L-1}\left[\mathbb{E}_{\mathbf{x}_k \sim r_k^\tau(\cdot;\mathbf{x}_0)}\left[\mathbb{E}_{\mathbf{z}\sim D_\theta(\mathbf{x}_k)}\left[\sum_{i=1,x_k^i=m}^{L} \mathrm{Cat}(i;G_\phi^\tau(\mathbf{x}_0,\mathbf{x}_k))\log\left(\frac{C^\tau(\mathbf{x}_0,\mathbf{x}_k)}{C^\tau(\mathbf{z}^{-i,x_0^i},\mathbf{x}_k)}\right)\right]\right]\right],$$

where in the last step we use that for any $\mathbf{z}$, $\mathrm{Cat}\left(i;G_\phi^\tau(\mathbf{z}^{-i,x_0^i},\mathbf{x}_k)\right)$ and $\mathrm{Cat}(i;G_\phi^\tau(\mathbf{x}_0,\mathbf{x}_k))$ have the same numerator in equation 15, just different normalizing constants. This lower bound is denoted as $\mathcal{E}_2^{\theta,\phi,\tau}$ in Corollary A.10. $\qquad\square$

### A.2.3 CONNECTION BETWEEN COROLLARY A.10 AND THE PAPL LOSS EQUATION 7

Here we show how one formally arrives at the PAPL loss from the detach gradient and stabilization steps taken in §3.4. We begin with the loss corresponding to the ELBO from Corollary A.10. This is given by:

$$\mathcal{L}(\theta,\phi) = -\mathbb{E}_{\mathbf{x}_0 \sim \mathbf{p}_{\text{data}}}\left[\mathcal{E}_1^{\theta,\phi,\tau}(\mathbf{x}_0) + \mathcal{E}_2^{\theta,\phi,\tau}(\mathbf{x}_0)\right],$$

where $\mathcal{E}_1^{\theta,\phi,\tau}, \mathcal{E}_2^{\theta,\phi,\tau}$ are as in Corollary A.10.

Next, we detach logits from the softmax weights $G_\phi^\tau$ given by equation 15. Observing that $\mathcal{E}_2^{\theta,\phi,\tau}$ depends only on these weights (through $C^\tau$) and not on the logits from the denoiser, we have

minimizing $\mathcal{L}(\theta, \phi)$ is equivalent to minimizing:

$$\mathcal{L}(\theta) = -\mathbb{E}_{\mathbf{x}_0 \sim \mathbf{p}_{\text{data}}} \left[ \mathcal{E}_1^{\theta, \phi, \tau}(\mathbf{x}_0) \right]$$

$$= -\sum_{k=0}^{L-1} \mathbb{E}_{\mathbf{x}_0 \sim \mathbf{p}_{\text{data}}} \left[ \mathbb{E}_{\mathbf{x}_k \sim r_k^\tau(\cdot; \mathbf{x}_0)} \left[ \sum_{i=1, x_k^i = m}^{L} \text{Cat}(i; G_\phi^\tau(\mathbf{x}_0, \mathbf{x}_k)) \log \left( \text{Cat}(x_0^i; D_\theta^i(\mathbf{x}_k)) \right) \right] \right].$$

After this, we replace sampling $\mathbf{x}_k \sim r_k^\tau(\cdot; \mathbf{x}_0)$ with sampling $\mathbf{x}_k \sim r_k(\cdot; \mathbf{x}_0)$ with $r_k$ as in equation 3. Indeed, this was the reason for using the softmax approximation of Corollary A.10 rather than the greedy ELBO of Corollary A.9 in the first place- which the deterministic paths from $E^{\theta, \text{greedy}}$ may be very far from the uniformly random paths of $r_k$, at least we have $r_k^\tau \to r_k$ as $\tau \to \infty$. The loss becomes:

$$\mathcal{L}(\theta) = -\sum_{k=0}^{L-1} \mathbb{E}_{\mathbf{x}_0 \sim \mathbf{p}_{\text{data}}} \left[ \mathbb{E}_{\mathbf{x}_k \sim r_k(\cdot; \mathbf{x}_0)} \left[ \sum_{i=1, x_k^i = m}^{L} w^{i, \tau} \log \left( \text{Cat}(x_0^i; D_\theta^i(\mathbf{x}_k)) \right) \right] \right],$$

where $w^{i, \tau} = \text{Cat}(i; G_\phi^\tau(\mathbf{x}_0, \mathbf{x}_k)) \propto \exp \left( \frac{1}{\tau} \log \left( \text{Cat}(z^j; D_\theta^j(\mathbf{x})) \right) \right)$. Finally, we observe that this is identical to the vanilla loss

$$\mathcal{L}^{\text{unif}}(\theta) = -\mathbb{E}_{\mathbf{x}_0 \sim \mathbf{p}_{\text{data}}} \left[ \mathcal{E}^{\theta, \text{unif}}(\mathbf{x}_0) \right]$$

associated to the vanilla ELBO equation 1, except that $\frac{1}{L-k}$ has been replaced by $w^{i, \tau}$ as the weight in the sum. Thus, interpolating with a constant which decreases linearly with the number of samples yields:

$$\mathcal{L}_{\text{PAPL}}(\theta) = -\mathbb{E}_{\mathbf{x}_0 \sim \mathbf{p}_{\text{data}}} \left[ \mathcal{E}^{\theta, \text{unif}}(\mathbf{x}_0) \right]$$

$$- \sum_{k=0}^{L-1} \mathbb{E}_{\mathbf{x}_0 \sim \mathbf{p}_{\text{data}}} \left[ \mathbb{E}_{\mathbf{x}_k \sim r_k(\cdot; \mathbf{x}_0)} \left[ \sum_{i=1, x_k^i = m}^{L} \frac{\alpha}{L-k} w^{i, \tau} \log \left( \text{Cat}(x_0^i; D_\theta^i(\mathbf{x}_k)) \right) \right] \right]$$

$$= - \sum_{k=0}^{L-1} \mathbb{E}_{\mathbf{x}_0 \sim \mathbf{p}_{\text{data}}} \left[ \mathbb{E}_{\mathbf{x}_k \sim r_k(\cdot; \mathbf{x}_0)} \left[ \sum_{i=1, x_k^i = m}^{L} \frac{1}{L-k} (1 + \alpha w^{i, \tau}) \log \left( \text{Cat}(x_0^i; D_\theta^i(\mathbf{x}_k)) \right) \right] \right].$$

Suppressing the distributions of the random variables $\mathbf{x}_0, k, \mathbf{x}_k$ in the notation, this is precisely equation 7.

### A.2.4 COMPARING RELATIVE SIZE OF THE APPROXIMATE LOSSES

Here we will see how the vanilla DLM loss (recalling here the discussion in A.1.3 and equation 3):

$$\mathcal{L}^{\text{unif}}(\theta) = -\mathbb{E}_{\mathbf{x}_0 \sim p_{data}} \left[ \mathcal{E}^{\theta, \text{unif}}(\mathbf{x}_0) \right]$$

$$= -L \mathbb{E}_{\mathbf{x}_0 \sim p_{data}} \left[ \mathbb{E}_{k \sim \text{Unif}([0:L-1])} \left[ \mathbb{E}_{\mathbf{x}_k \sim r_k(\cdot; \mathbf{x}_0)} \left[ \sum_{i=1, x_k^i \neq \mathbf{m}}^{L} \frac{1}{L-k} \log \left( \text{Cat} \left( x_0^i; D_\theta^i(\mathbf{x}_k) \right) \right) \right] \right] \right]$$

compares with the surrogate corrected loss:

$$\mathcal{L}^\tau(\theta) = -L \mathbb{E}_{\mathbf{x}_0 \sim p_{data}} \left[ \mathbb{E}_{k \sim \text{Unif}([0:L-1])} \left[ \mathbb{E}_{\mathbf{x}_k \sim r_k(\cdot; \mathbf{x}_0)} \left[ \sum_{i=1, x_k^i = m}^{L} \text{Cat}(i; G_\phi^\tau(\mathbf{x}_0, \mathbf{x}_k)) \times \right. \right. \right.$$

$$\left. \left. \left. \times \log \left( \text{Cat}(x_0^i; D_\theta^i(\mathbf{x}_k)) \right) \right] \right] \right],$$

which is the PAPL loss before interpolation with the vanilla MDM loss as per the previous subsection.

Here recall $r_k(\cdot; \mathbf{x}_0)$ from equation 3 and $C^\tau, G^\tau$ from equation 15.

**Proposition A.11.** *For any* $\tau_1 > \tau_2 > 0$, $\mathcal{L}^{\text{unif}}(\theta) \geq \mathcal{L}^{\tau_1}(\theta) \geq \mathcal{L}^{\tau_2}(\theta)$.

*Proof.* As the expected values are over the same distributions, it suffices to prove the result for the integrands. Let $\mathbf{x}_0, \mathbf{x}_k \in \mathcal{V}^L$ and $\mathcal{M} = \{i \in \{1, \ldots, L\} : x^i = \mathbf{m}\}$. Note $|\mathcal{M}| = L - k$ by definition. Define:

$$\ell^i = \log\left(\text{Cat}\left(x_0^i; D_\theta^i(\mathbf{x_k})\right)\right), i \in \mathcal{M}$$

so that

$$\text{Cat}(i; G_\phi^\tau(\mathbf{x}_0, \mathbf{x}_k)) = \exp(\ell^i/\tau)/C^\tau(\ell) := w_\tau^i(\ell)$$

$$C^\tau(\ell) = \sum_{i \in \mathcal{M}} \exp(\ell^i/\tau).$$

Noting the minus sign in front of the losses, we simply need to establish that

$$\sum_{i \in \mathcal{M}} \frac{1}{L-k}\ell^i \leq \sum_{i \in \mathcal{M}} w_{\tau_1}^i(\ell)\ell^i \leq \sum_{i \in \mathcal{M}} w_{\tau_2}^i(\ell)\ell^i.$$

Observing that $\lim_{\tau \to \infty} w_\tau^i(\ell) = \frac{1}{L-k}, \forall i \in \mathcal{M}$ and $\ell$, we simply show that

$$\frac{d}{d\tau}\sum_{i \in \mathcal{M}} w_\tau^i(\ell)\ell^i < 0, \forall \tau > 0.$$

Letting $F(\tau) = \sum_{i \in \mathcal{M}} w_\tau^i(\ell)\ell^i$, We have

$$\frac{d}{d\tau}w_\tau^i(\ell) = \frac{w_\tau^i(\ell)}{\tau^2}\left[\sum_{j \in \mathcal{M}} w_\tau^j(\ell)\ell^j - \ell^i\right] = \frac{w_\tau^i(\ell)}{\tau^2}\left[F(\tau) - \ell^i\right],$$

so

$$\frac{d}{d\tau}F(\tau) = \sum_{i \in \mathcal{M}} \frac{w_\tau^i(\ell)}{\tau^2}\left[F(\tau) - \ell^i\right]\ell^i = \frac{1}{\tau^2}\left[(F(\tau))^2 - \sum_{i \in \mathcal{M}} w_\tau^i(\ell)(\ell^i)^2\right].$$

By Jensen's inequality, $(F(\tau))^2 \leq \sum_{i \in \mathcal{M}} w_\tau^i(\ell)(\ell^i)^2$, so we are done. $\qquad\square$

### A.3 ALTERNATIVE PROOF OF PROPOSITION 3.2: CONTINUOUS TIME MARKOV CHAINS PERSPECTIVE

Here, for reference, we show how Proposition 3.2 can be derived from the continuous time Markov chains perspective taken in the discrete diffusion literature (Campbell et al., 2022; 2024; Lou et al., 2024; Sun et al., 2023).

### A.3.1 TIME-INHOMOGENEOUS CONTINUOUS TIME MARKOV CHAINS (CTMC)

A (time-inhomogeneous) continuous-time Markov chain $\{X_t\}_{t \geq 0}$ on a finite set $\mathcal{X}$ is a stochastic process satisfying the Markov property, which can be formally summarized as $\mathbb{P}(X_t = y | X_{s_1} = x_1, \ldots, X_{s_k} = x_k, X_s = x) = \mathbb{P}(X_t = y | X_s = x), \forall y, x_1, \ldots, x_k, x \in \mathcal{X}, 0 \leq s_1 < s_2 < \ldots < s_k < s < t \leq 1$. One can construct such a process by specifying a "rate matrix" $Q_t \in \mathbb{R}^{|\mathcal{X}| \times |\mathcal{X}|}$ with $Q_t(y, x) > 0$ and $Q_t(x, x) = -\sum_{y \neq x} Q_t(y, x)$ for all $x \neq y \in \mathcal{X}$ and $t \geq 0$. Along with an initial distribution $\mu \in \Delta^{|X|}$, $Q$ determines the 1-dimensional time marginals $\mathbb{P}(X_t = \cdot) \in \Delta^{|X|}$ via the Kolmogorov equation:

$$\frac{d}{dt}\mathbb{P}(X_t = \cdot) = Q_t\mathbb{P}(X_t = \cdot), \qquad t \geq 0 \tag{17}$$

$$\mathbb{P}(X_0 = x) = \mu(x), \qquad x \in \mathcal{X}.$$

When the above holds, we will say $Q$ "generates" $X$. Note that one can see necessarily that if $Q$ generates $X$,

$$Q_t(y, x) := \lim_{s \downarrow t} \frac{d}{ds}\mathbb{P}(X_s = y | X_t = x), \quad x \neq y \in \mathcal{X}. \tag{18}$$

Knowing the entries of $Q$ also provides a means of generating samples from $X_t$ at any given time, since paths of $\{X_t\}_{t \geq 0}$ can be realized via a sequence of jump times $\{\tau_n\}_{n \in \mathbb{N}}$, with $\tau_i = \inf\{t > \tau_{i-1} : X_t \neq X_{\tau_{i-1}}\}$ and the effective discrete-time jump process $\{X_{\tau_i}\}_{i \in \mathbb{N}}$. Then

$$\mathbb{P}(X_{\tau_{i+1}} = y | X_{\tau_{i+1}} = x, \tau_i = t) = -\frac{Q_t(y,x)}{Q_t(x,x)}, \tag{19}$$

and

$$\log(\mathbb{P}(\tau_{i+1} > t | X_{\tau_i} = x, \tau_i = s)) = \int_s^t Q_p(x,x)dp. \tag{20}$$

For more background on time-inhomogenous continuous-time Markov chains, see e.g. Chapter 2 of Yin & Zhang (2013) or the appendix of Ren et al. (2025).

### A.3.2    DLMS IN THE CTMC FRAMEWORK

In the original CTMC framework for DLMs (Lou et al., 2024; Shi et al., 2024; Sahoo et al., 2024), one begins with a coordinate-wise forward corruption process:

$$p_t(x_t^i | x_0^i) = \mathrm{Cat}(x_t^i; \alpha_t \delta(x_0^i) + (1 - \alpha_t)\delta(\mathbf{m})) \tag{21}$$

for $\alpha : [0,1] \to [0,1]$ a differentiable, monotone-decreasing function with $\alpha_0 = 1$ and $\alpha_1 = 1$. Using equation equation 17, one sees that noising each coordinate independently according to corresponds to a CTMC $\overrightarrow{X}_t$ with state space $\mathcal{V}^L$, intial data $\mathbf{x}_0$ and rate matrix given by, for $\mathbf{x}, \mathbf{y} \in \mathcal{V}^L$

$$\overrightarrow{Q}_t^{\mathbf{x}_0}(\mathbf{y}, \mathbf{x}) = \begin{cases} \sigma(t), & d_{\mathrm{HAM}}(\mathbf{x}, \mathbf{y}) = 1, x^i \neq y^i, x^i = \mathbf{m} \\ -\sigma(t) N_M(\mathbf{x}), & \mathbf{x} = \mathbf{y} \\ 0, & \text{otherwise} \end{cases}$$

where $\sigma(t) = -\frac{d}{dt}\log(\alpha_t)$.

One then uses a classic time-reversal formula (see, e.g. Sun et al. (2023) Proposition 3.2.) to obtain a rate matrix generating $\overleftarrow{X}_t^{\mathbf{x}_0}$ so that $\mathbb{P}\left(\overleftarrow{X}_t^{\mathbf{x}_0} = \mathbf{x}\right) = \mathbb{P}\left(\overrightarrow{X}_t = \mathbf{x} | \overrightarrow{X}_t = \mathbf{x}_0\right), \forall \mathbf{x} \in \mathcal{V}^L$. This rate matrix is given by, for $\mathbf{x}, \mathbf{y} \in \mathcal{V}^L$:

$$\overleftarrow{Q}_t^{\mathbf{x}_0}(\mathbf{y}, \mathbf{x}) = \begin{cases} \beta_t \mathrm{Cat}(y^i; \delta(x_0^i)), & d_{\mathrm{HAM}}(\mathbf{x}, \mathbf{y}) = 1, x^i \neq y^i, x^i = \mathbf{m} \\ -\beta_t N_M(\mathbf{x}), & \mathbf{x} = \mathbf{y} \\ 0, & \text{otherwise} \end{cases} \tag{22}$$

where

$$\beta_t := -\frac{\frac{d\alpha_{1-t}}{dt}}{1 - \alpha_{1-t}} \tag{23}$$

Letting $\tau_k$ for $k \in \mathbb{N}$ be the time of $\overleftarrow{X}_t^{\mathbf{x}_0}$'s $k$'th jump, we have by equation 20:

$$\log \mathbb{P}(\tau_{k+1} > t | \tau_k = s, \overleftarrow{X}_{\tau_k}^{\mathbf{x}_0} = \mathbf{x}) = -N_M(\mathbf{x}) \int_s^t \beta_\tau d\tau$$

and by equation 19, for $\mathbf{x} \neq \mathbf{y}$:

$$\mathbb{P}(\overleftarrow{X}_{\tau_{k+1}}^{\mathbf{x}_0} = y | \overleftarrow{X}_{\tau_k}^{\mathbf{x}_0} = x, \tau_{k+1} = t)$$
$$= \begin{cases} 1/N_M(\mathbf{x}), & d_{\mathrm{HAM}}(\mathbf{x}, \mathbf{y}) = 1, x^i \neq y^i, x^i = \mathbf{m}, y^i = x_0^i \\ 0, & \text{otherwise} \end{cases}. \tag{24}$$

That is, one waits for an exponential clock to ring with the given speed, then regardless of how long it took, chooses uniformly at random between masked positions of $\mathbf{x}$ to get some index $i$, and unmasks that token to $x_0^i$.

One then seeks to denoise from $(\mathbf{m}, \ldots, \mathbf{m})$ to $\mathbf{x}_0 \sim p_{data}$ using the CTMC $\overset{\leftarrow \theta, \text{unif}}{X_t}$ with state space $\mathcal{V}^L$ which one obtains via replacing $\delta(x_0^i)$ in equation 22 with a neural denoiser $D_\theta^i(\mathbf{x})$. A rate matrix generating $\overset{\leftarrow \theta, \text{unif}}{X_t}$ is given by, for $\mathbf{x}, \mathbf{y} \in \mathcal{V}^L$:

$$Q^{\theta, \text{mask}}(\mathbf{y}, \mathbf{x}) = \begin{cases} \beta_t \text{Cat}(y^i; D_\theta^i(\boldsymbol{x})), & d_{\text{HAM}}(\mathbf{x}, \mathbf{y}) = 1, x^i \neq y^i, x^i = \mathbf{m} \\ -\beta_t N_M(\mathbf{x}), & \mathbf{x} = \mathbf{y} \\ 0, & \text{otherwise} \end{cases}. \quad (25)$$

This means, letting $\tau_k^\theta$ for $k \in \mathbb{N}$ be the time of $\overset{\leftarrow \theta, \text{mask}}{X_t}$'s $k$'th jump, we have by equation 20:

$$\log \mathbb{P}(\tau_{k+1}^\theta > t | \tau_k^\theta = s, \overset{\leftarrow \theta, \text{mask}}{X_{\tau_k}} = \mathbf{x}) = -N_M(\mathbf{x}) \int_s^t \beta_\tau d\tau \quad (26)$$

and by equation 19, for $\mathbf{x} \neq \mathbf{y}$:

$$\mathbb{P}(\overset{\leftarrow \theta, \text{mask}}{X_{\tau_{k+1}}} = \mathbf{y} | \overset{\leftarrow \theta, \text{mask}}{X_{\tau_k}} = \mathbf{x}, \tau_{k+1} = t)$$
$$= \begin{cases} \text{Cat}(y^i; D_\theta^i(\mathbf{x}))/N_M(\mathbf{x}), & d_{\text{HAM}}(\mathbf{x}, \mathbf{y}) = 1, x^i \neq y^i, x^i = \mathbf{m}, y^i \neq \mathbf{m} \\ 0, & \text{otherwise} \end{cases}. \quad (27)$$

That is, one waits for an exponential clock with the same given speed as for $\overset{\leftarrow \mathbf{x}_0}{X}$ to ring , then regardless of how long it took, chooses uniformly at random between masked positions of $\mathbf{x}$ to get some index $i$, and unmasks that token to $y^i$ with probability $\text{Cat}(y^i; D_\theta^i(\boldsymbol{x}))$.

This is summarized succinctly via the corresponding Gillespie sampling scheme for a standard masked diffusion model, which, defining

$$\mathcal{M}(\mathbf{x}) \coloneqq \{j \in \{1, \ldots, L\} : x^j = \mathbf{m}\}, \quad \mathbf{x} \in \mathcal{V}^L \quad (28)$$

is given by Alg. 3.

---

**Algorithm 3** Gillespie Sampler for Masked Diffusion Models

1: **Initialize:** $\mathbf{x}_0 \leftarrow (\mathbf{m}, \mathbf{m}, \ldots, \mathbf{m})$, denoiser $D_\theta$
2: **for** $k = 0 : L - 1$ **do**
3:      **Choose Random Coordinate for Unmasking:**
4:      Sample dimension $i \sim \text{Unif}(\mathcal{M}(\mathbf{x}_k))$
5:      **Denoise:**
6:      Sample $z^i \sim D_\theta^i(\mathbf{x}_k)$
7:      $\mathbf{x}_{k+1} \leftarrow \mathbf{x}_k$
8:      $x_{k+1}^i \leftarrow z^i$
9: **end for**
10: **return** $x_L$
11:

---

### A.3.3 SETUP AND ELBO IN THE CTMC FRAMEWORK

Now we show how to derive a corrected ELBO for the dynamics described by Alg. 2 as one would using the CTMC framework. First, we observe that for any $\tilde{Y}^{\mathbf{x}_0}$ a CTMC on time interval $[0, 1]$ and state space $\mathcal{S}$ such that $\mathbb{P}(\tilde{Y}_1^{\mathbf{x}_0} = \mathbf{x}_0) = 1$ and $p$ a distribution on $\mathcal{S}$ given by $p(\mathbf{x}) = \mathbb{P}(X_1 = \mathbf{x})$ for another CTMC on time interval $[0, 1]$ and state space $\mathcal{S}$:

$$\log(p(\mathbf{x}_0)) = -D_{KL}(\delta(\mathbf{x}_0)||p) \geq -D_{KL}(\mathbb{R}^{\mathbf{x}_0}||\mathbb{Q}), \quad (29)$$

where we let $\mathbb{R}^{\mathbf{x}_0}$ denote the distribution of $\tilde{Y}^{\mathbf{x}_0}$ (on the Skorokhod space $D([0, 1]; \mathcal{S})$ of all cádlág paths from $[0, 1]$ to $\mathcal{V}^L$) and $\mathbb{Q}$ the same but for $X$, and to get the bound, we use the data-processing inequality (an infinite-dimensional generalization of Corollary A.4- see, e.g. Budhiraja & Dupuis (2019) Lemma 2.4 (f)).

That is, in order to make the terminal distribution $p$ of $X$ close to $\delta(\mathbf{x}_0)$, one can simply require that its entire path is close to that of $\tilde{Y}^{\mathbf{x}_0}$.

The benefit of using the $KL$ divergence between the paths is that via Girsanov's Theorem for Markov Jump processes (see e.g. Theorem III.5.34 in Jacod & Shiryaev (2013) for a general result or Ren et al. (2025) Theorems 3.3/3.5 for the specific Markov Chain setting) it yields a simple expression in terms of the rate matrices generating the dynamics of $\tilde{Y}^{\mathbf{x}_0}$ and $X$.

This result states that, for a CTMC $Y$ with rate matrix $R_t$ and $Y_0 \sim \mu$ and a CTMC $X$ with rate matrix $Q_t$ and $X_0 \sim \nu$ on the same state space $\mathcal{S}$, denoting by $\mathbb{R}$ the distribution of $Y$ on $D([0,1];\mathcal{S})$ and $\mathbb{Q}$ similarly but for $X$. the equality:

$$D_{KL}(\mathbb{R}||\mathbb{Q}) = D_{KL}(\mu||\nu) \tag{30}$$
$$+ \int_0^1 \mathbb{E}_{x_t \sim r_t} \left[ R_t(x_t, x_t) - Q_t(x_t, x_t) + \sum_{y \in \mathcal{S}, y \neq x_t} R_t(y, x_t) \log \left( \frac{R_t(y, x_t)}{Q_t(y, x_t)} \right) \right] dt$$
$$r_t(x) := \mathbb{P}(Y_t = x), x \in \mathcal{S}.$$

holds a under mild assumptions on $R$ and $Q$ (see remark 3.4 in Ren et al. (2025)). Note that equation 30 is simply a continuous time extension of Proposition A.6. Recalling equation 20, the term $R_t(x_t, x_t) - Q_t(x_t, x_t)$ measures the difference in jump times between $X$ and $Y$, while the second term is essentially a KL divergence between the transition rates.

We proceed by identifying a CTMC $X^{\theta,\phi}$ with state space $\mathcal{V}^L$ such that $\mathbb{P}\left(X_1^{\theta,\phi} = \mathbf{x}\right) = p_\theta^{G_\phi}(\mathbf{x})$, so that we may apply equation 29 and equation 30 to $p_\theta^{G_\phi}$ obtain an ELBO. Denoting by $Q_t^{\theta,\phi}$ the rate matrix for $X^{\theta,\phi}$, via equation 4 and equation 19, we must have for any $t \in [0,1]$ and $\mathbf{x} \neq \mathbf{y} \in \mathcal{V}^L$:

$$-\frac{Q_t^{\theta,\phi}(\mathbf{y}, \mathbf{x})}{Q_t^{\theta,\phi}(\mathbf{x}, \mathbf{x})} = \begin{cases} \text{Cat}\left(y^i; D_\theta^i(\mathbf{x})\right) F_{\theta,\phi}(\mathbf{x}, y^i, i), & d_{\text{HAM}}(\mathbf{x}, \mathbf{y}) = 1, x^i \neq y^i, x^i = \mathbf{m} \\ 0, & \text{otherwise} \end{cases}.$$

As the transition probabilities do not depend on the transition rates, we simply need that the transition rates are so that by time 1 all tokens will become unmasked. We thus simply maintain those from vanilla DLMs, found in equation 26, and so, recalling equation 20, we set for $\mathbf{x} \in \mathcal{V}^L$:

$$Q^{\theta,\phi}(\mathbf{x}, \mathbf{x}) = -\beta_t N_M(\mathbf{x}),$$

where we recall $\beta_t$ from equation 23.

We then have our full definition of $Q^{\theta,\phi}$. For $\mathbf{x}, \mathbf{y} \in \mathcal{V}^L$:

$$Q^{\theta,\phi}(\mathbf{y}, \mathbf{x}) = \begin{cases} \beta_t N_M(\mathbf{x}) \text{Cat}\left(y^i; D_\theta^i(\mathbf{x})\right) F_{\theta,\phi}(\mathbf{x}, y^i, i), & d_{\text{HAM}}(\mathbf{x}, \mathbf{y}) = 1, x^i \neq y^i, x^i = \mathbf{m} \\ -\beta_t N_M(\mathbf{x}), & \mathbf{x} = \mathbf{y} \\ 0, & \text{otherwise} \end{cases}.$$
$$\tag{31}$$

Letting $R_t(\cdot, \cdot; \mathbf{x}_0)$ be the rate matrix for our reference chain $Y^{\mathbf{x}_0}$ to be used for inserting into equation 29, since we don't want to worry about enforcing the jump times of $X^{\theta,\phi}$ and $Y^{\mathbf{x}_0}$ in the form of the ELBO (as these have no bearing on the sample generated by $\overset{\leftarrow\theta,\phi}{X}$ and hence should not be trained for), we also set for $\mathbf{x} \in \mathcal{V}^L$:

$$R_t(\mathbf{x}, \mathbf{x}; \mathbf{x}_0) = -\beta_t N_M(\mathbf{x}).$$

Now, to choose the off diagonal entries of $R_t(\cdot, \cdot; \mathbf{x}_0)$, we seek to modify jump locations to be different than in the vanilla setting, where the coordinate to flip is chosen uniformly at random (see equation 24).

Instead, we opt to choose $R(\cdot, \cdot; \mathbf{x}_0) = R^{G_\phi}(\cdot, \cdot; \mathbf{x}_0)$ to select coordinates to denoise according to the planner. In this sense, we will be learning both the forward and reverse process simultaneously when using this ELBO.

Recalling equation 19, we thus want for $\mathbf{x} \neq \mathbf{y} \in \mathcal{V}^L$:

$$-\frac{R^{G_\phi}(\mathbf{y}, \mathbf{x}; \mathbf{x}_0)}{R^{G_\phi}(\mathbf{x}, \mathbf{x}; \mathbf{x}_0)} = \begin{cases} \text{Cat}(i; G_\phi(\mathbf{x}_0, \mathbf{x}))\text{Cat}(y^i; \delta(x_0^i)), & d_{\text{HAM}}(\mathbf{x}, \mathbf{y}) = 1, x^i \neq y^i, x^i = \mathbf{m} \\ 0, & \text{otherwise} \end{cases}.$$

Now the dynamics of $Y^{\mathbf{x}_0} = Y^{G_\phi, \mathbf{x}_0}$ and its rate matrix $R(\cdot, \cdot; \mathbf{x}_0) = R^{G_\phi}(\cdot, \cdot; \mathbf{x}_0)$ have been determined.

We have for $\mathbf{x}, \mathbf{y} \in \mathcal{V}^L$:

$$R^{G_\phi}(\mathbf{y}, \mathbf{x}; \mathbf{x}_0) = \begin{cases} \beta_t N_M(\mathbf{x})\text{Cat}(i; G_\phi(\mathbf{x}_0, \mathbf{x})), & d_{\text{HAM}}(\mathbf{x}, \mathbf{y}) = 1, x^i \neq y^i, x^i = \mathbf{m}, y^i = x_0^i \\ -\beta_t N_M(\mathbf{x}) & , \quad \mathbf{x} = \mathbf{y} \\ 0, & \text{otherwise} \end{cases}.$$

(32)

Note that, taking $Y_0^{G_\phi, \mathbf{x}_0} = X_0^{\theta, \phi} = (\mathbf{m}, \dots, \mathbf{m})$, indeed $Y_1^{G_\phi, \mathbf{x}_0} = \mathbf{x}_0$, so that equation 29 applies.

We arrive at the following proposition:

**Proposition A.12.** *For any planner $G_\phi$, we have the following ELBO:*

$$\log(p_\theta^{G_\phi}(\mathbf{x}_0)) \geq E_1^{\theta, \phi}(\mathbf{x}_0) + E_2^{\theta, \phi}(\mathbf{x}_0),$$

*where, letting $r_t^{G_\phi}(\cdot; \mathbf{x}_0)$ be the distribution of $Y^{G_\phi, \mathbf{x}_0}$ with rate matrix equation 32 and $Y_0^{G_\phi, \mathbf{x}_0} = (\mathbf{m}, \dots, \mathbf{m})$:*

$$E_1^{\theta, \phi}(\mathbf{x}_0) = \int_0^1 \beta_t \underset{\mathbf{x}_t \sim r_t^{G_\phi}(\cdot; \mathbf{x}_0)}{\mathbb{E}} \left[ N_M(\mathbf{x}_t) \sum_{i=1, x_t^i = m}^L \text{Cat}(i; G_\phi(\mathbf{x}_0, \mathbf{x}_t)) \log \left( \text{Cat}(x_0^i; D_\theta^i(\mathbf{x}_t)) \right) \right] dt$$

$$E_2^{\theta, \phi}(\mathbf{x}_0) = -\int_0^1 \beta_t \underset{\mathbf{x}_t \sim r_t^{G_\phi}(\cdot; \mathbf{x}_0)}{\mathbb{E}} \left[ N_M(\mathbf{x}_t) \sum_{i=1, x_t^i = m}^L \text{Cat}(i; G_\phi(\mathbf{x}_0, \mathbf{x}_t)) \log \left( \frac{\text{Cat}(i; G_\phi(\mathbf{x}_0, \mathbf{x}_t))}{F_{\theta, \phi}(\mathbf{x}_t, x_0^i, i)} \right) \right] dt$$

*Proof.* Let $\mathbb{R}^{\mathbf{x}_0}$ denote the distribution of paths of $Y^{G_\phi, \mathbf{x}_0}$ and $\mathbb{Q}^{\theta, \phi}$ those of $X^{\theta, \phi}$ from the preceding discussion. Then, by equation 29, $\log(p_\theta^{G_\phi}(\mathbf{x}_0)) \geq -D_{KL}(\mathbb{R}^{\mathbf{x}_0}||\mathbb{Q}^{\theta, \phi})$. From equation 30, we have, using $Y^{G_\phi, \mathbf{x}_0}$ and $X^{\theta, \phi}$ have the same initial data so that the first KL term is 0:

$$-D_{KL}(\mathbb{R}^{\mathbf{x}_0}||\mathbb{Q}^{\theta, \phi})$$

$$= -\int_0^1 \underset{\mathbf{x}_t \sim r_t^{G_\phi}(\cdot; \mathbf{x}_0)}{\mathbb{E}} \left[ -Q_t^{\theta, \phi}(\mathbf{x}_t, \mathbf{x}_t) + R^{G_\phi}(\mathbf{x}_t, \mathbf{x}_t; \mathbf{x}_0) \right.$$

$$\left. + \sum_{\mathbf{y} \neq \mathbf{x}_t} R^{G_\phi}(\mathbf{y}, \mathbf{x}_t; \mathbf{x}_0) \log \left( \frac{R^{G_\phi}(\mathbf{y}, \mathbf{x}_t; \mathbf{x}_0)}{Q_t^{\theta, \phi}(\mathbf{y}, \mathbf{x}_t)} \right) \right] dt$$

$$= -\int_0^1 \beta_t \underset{\mathbf{x}_t \sim r_t^{G_\phi}(\cdot; \mathbf{x}_0)}{\mathbb{E}} \left[ N_M(\mathbf{x}_t) \sum_{i=1, x_t^i = m}^L \text{Cat}(i; G_\phi(\mathbf{x}_0, \mathbf{x}_t)) \times \right.$$

$$\left. \times \log \left( \frac{\text{Cat}(i; G_\phi(\mathbf{x}_0, \mathbf{x}_t))}{\text{Cat}(x_0^i; D_\theta^i(\mathbf{x}_t))F_{\theta, \phi}(\mathbf{x}_t, x_0^i, i)} \right) \right] dt$$

$$= \mathcal{E}_1^{\theta, \phi}(\mathbf{x}_0) + \mathcal{E}_2^{\theta, \phi}(\mathbf{x}_0).$$

$\square$

This form is seen to be equivalent to that in Proposition 3.2 in the next subsection.

### A.3.4   TIME INDEPENDENT FORMULATION OF THE ELBO

Now we observe that, as expected, the ELBO is independent of the time schedule $\alpha_t$ (and hence $\beta_t$). We start by observing that

$$\mathbb{P}(N_M(Y_t^{G_\phi, \mathbf{x}_0}) = k) = \binom{L}{k} \left( \exp \left( -\int_0^t \beta_s ds \right) \right)^k \left( 1 - \exp \left( -\int_0^t \beta_s ds \right) \right)^{L-k}.$$

One can see this via law of competing exponentials or solving equation 17 for the pure death chain representing the number of mask states.

Then, recalling that the transition probabilities for $Y_t^{G_\phi, \mathbf{x}_0}$ are independent of time and using equation 19, we have $Y_t^{G_\phi, \mathbf{x}_0} | N_M(Y_t^{G_\phi, \mathbf{x}_0}) = L - k$ is equal in distribution to $\bar{Y}_k^{G_\phi, \mathbf{x}_0}$, where $\bar{Y}^{G_\phi, \mathbf{x}_0}$ is the effective discrete time Markov chain with rate matrix equation 16.

Denoting by $\bar{p}_k^{G_\phi}(\cdot; \mathbf{x}_0)$ the distribution of $\bar{Y}_k^{G_\phi, \mathbf{x}_0}$, we have, for example:

$$E_1^{\theta, \phi}(\mathbf{x}_0) = \sum_{k=1}^{L} \binom{L}{k} \int_0^1 \beta_t \left( \exp\left( -\int_0^t \beta_s ds \right) \right)^k \left( 1 - \exp\left( -\int_0^t \beta_s ds \right) \right)^{L-k} dt$$

$$\times \quad k \mathbb{E}_{\mathbf{x}_k \sim \bar{p}_{L-k}^{G_\phi}(\cdot; \mathbf{x}_0)} \left[ \sum_{i=1, x_k^i = m}^{L} \mathrm{Cat}(i; G_\phi(\mathbf{x}_0, \mathbf{x}_k)) \log \left( \mathrm{Cat}(x_0^i; D_\theta^i(\mathbf{x}_k)) \right) \right].$$

Using $\exp(-\int_0^1 \beta_s ds) = \exp(-\int_0^1 \frac{d}{ds} \log(\alpha_{1-s}) ds) = \lim_{s \downarrow 0} \exp(\log(\alpha(1-s)) - \log(\alpha(0))) = \lim_{s \downarrow 0} \alpha(1-s) = 0$ since $\alpha(1) = 0, \alpha(0) = 1$, we have

$$\int_0^1 \beta_t \left( \exp\left( -\int_0^t \beta_s ds \right) \right)^k \left( 1 - \exp\left( -\int_0^t \beta_s ds \right) \right)^{L-k} dt = \int_0^1 u^{k-1}(1-u)^{L-k} du$$
$$= B(k, L - k + 1)$$
$$= \frac{1}{k \binom{L}{k}}$$

where $B$ is the beta function.

Thus, changing $k$ to $L - k$ in the sum, we have:

$$E_1^{\theta, \phi}(\mathbf{x}_0) = \sum_{k=0}^{L-1} \mathbb{E}_{\mathbf{x}_k \sim \bar{p}_k^{G_\phi}(\cdot; \mathbf{x}_0)} \left[ \sum_{i=1, x_t^i = m}^{L} \mathrm{Cat}(i; G_\phi(\mathbf{x}_0, \mathbf{x}_k)) \log \left( \mathrm{Cat}(x_0^i; D_\theta^i(\mathbf{x}_k)) \right) \right].$$

Recalling that $\bar{p}_k^{G_\phi}$ is what is denoted as $p_k^{G_\phi}$ (we only added the bar here to distinguish it from the distribution of the continuous time chain), we see $E_1^{\theta, \phi}$ from Proposition A.12 is equal to $\mathcal{E}_1^{\theta, \phi}$ from 3.2.

Applying the same manipulations to $\mathcal{E}_2^{\theta, \phi}(\mathbf{x}_0)$, we arrive at the following:

**Proposition A.13.** *For $\mathcal{E}_1^{\theta, \phi}$, $\mathcal{E}_2^{\theta, \phi}$ as in Proposition 3.2 and $E_1^{\theta, \phi}$, $E_2^{\theta, \phi}$ as in Proposition A.12, we have:*

$$\mathcal{E}_1^{\theta, \phi}(\mathbf{x}_0) = E_1^{\theta, \phi}(\mathbf{x}_0), \quad \mathcal{E}_2^{\theta, \phi}(\mathbf{x}_0) = E_2^{\theta, \phi}(\mathbf{x}_0), \quad \forall \mathbf{x}_0 \in \mathcal{V}^L.$$

That is, rather than simulate the CTMC up to some random time $t \sim \mathrm{Unif}(0, 1)$ to obtain a sample of $Y_t^{G_\phi, \mathbf{x}_0}$, one may instead either sample an entire trajectory of the discrete time chain $\bar{Y}^{G_\phi, \mathbf{x}_0}$ and accumulate losses, or sample a random number of jumps $k \sim \mathrm{Unif}([0 : L - 1])$ and a trajectory of $\bar{Y}_k^{G_\phi, \mathbf{x}_0}$ up to time $k$.

## A.4 GENERALIZATION TO PLANNERS WITH REMASKING (P2-STYLE)

P2 (Peng et al., 2025a) allows for the remasking of clean tokens while still requiring that the number of unmasked tokens in $\mathbf{x}_k$ is $k$. We replace the planner $G_\phi : \mathcal{V}^L \times \mathcal{V}^L \to \Delta^L$ with a sequence of planners $G_{\phi,2}^k : \mathcal{V}^L \times \mathcal{V}^L \to \Delta^{\binom{L}{k}}, k = 1, \ldots, L$, where $G_{\phi,2}^k$ outputs a distribution on subsets of size $k$ of $[1 : L]$. We then do:

In P2, we in practice use "P2-Topk", which corresponds to:

$$G_{\phi,2}^k(\mathbf{z}, \mathbf{x}) = \delta\left( \mathrm{Top\text{-}k}_{i \in [1:L]}\left( \mathrm{Cat}\left( i; \hat{G}_\phi^\eta(\mathbf{z}, \mathbf{x}) \right) \right) \right), \tag{33}$$

---

**Algorithm 4** Gillespie Sampler with a P2-Planner

---

1: **Initialize:** $t \leftarrow 0, \mathbf{x}_0 \leftarrow (m, \ldots, m)$, P2 planner $\{G_{\phi,2}^k\}_{k=1}^L$, denoiser $D_\theta$
2: **for** $k = 0 : L - 1$ **do**
3:     **Plan** Sample $\mathbf{z} \sim D_\theta(\mathbf{x}_k)$
4:     Sample dimensions $I^{k+1} = (i_1, \ldots, i_{k+1}) \sim G_{\phi,2}^{k+1}(\mathbf{z}, \mathbf{x}_k)$
5:     **Denoise**
6:     $x_{k+1}^i \leftarrow z^i$ for $i \in I^{k+1}$
7:     $x_{k+1}^i \leftarrow \mathbf{m}$ for $i \notin I^{k+1}$
8: **end for**
9: **return** $\mathbf{x}_L$
10:

---

where for $i \in \{1, \ldots, L\}$ and $\eta \geq 0$

$$\text{Cat}(i; \hat{G}_\phi^\eta(\mathbf{z}, \mathbf{x})) \propto \eta \text{Cat}(x^i; \delta(\mathbf{m})) \text{Cat}(z^i; D_\theta^i(\mathbf{x})) + \left(1 - \text{Cat}(x^i; \delta(\mathbf{m}))\right) \text{Cat}(z^i; B_\phi^i(\mathbf{z})) \quad (34)$$

in the case of P2-BERT, and

$$\text{Cat}(i; \hat{G}_\phi^\eta(\mathbf{z}, \mathbf{x})) \propto \eta \text{Cat}(x^i; \delta(\mathbf{m})) \text{Cat}(z^i; D_\theta^i(\mathbf{x})) + \left(1 - \text{Cat}(x^i; \delta(\mathbf{m}))\right) \text{Cat}(z^i; \hat{D}_\theta^i(\mathbf{x})) \quad (35)$$

in the case of P2-self. Here the output of the denoiser in unmasked positions $i$ of $\mathbf{x}$ is no longer assumed to be $\delta(x^i)$ when we write it as $\hat{D}$, an $B_\phi$ denotes an external BERT model. $\eta$ is a "stochasticity parameter" which controls the frequency of remasking - increasing $\eta$ boosts $\hat{G}^\eta$ in masked positions, so that unmasked positions are more likely to fall outside of the Top-k and be remasked.

### A.4.1 DERIVING THE TRANSITION PROBABILITIES

Here we derive the one step transition probabilities for Alg. 4. This is the analogue of equation 4 in the setting where we generalize to allow for remasking.

Let $p_\theta^{G_{\phi,2}} \in \Delta^{d^L}$ denote the distribution on $\mathcal{V}^L$ of a sample obtained via running Alg. 4, and let $p_{\theta,k+1}^{G_{\phi,2}}(\cdot | \mathbf{x}_k) \in \Delta^{d^L}$ denote the distribution of $\mathbf{x}_{k+1}$ given $\mathbf{x}_k$. With abuse of notation, we will also let $p_{\theta,k+1}^{G_{\phi,2}}(\cdot, \cdot | \mathbf{x}_k) \in \Delta^{d^{L^2}}$ denote the joint distribution $\mathbf{x}_{k+1}$ and the $k+1$'st sample $\mathbf{z}$. Note that $N_M(\mathbf{x}_{k+1}) = L - (k+1)$ with probability 1. So for $\mathbf{y} \in \mathcal{V}^L$ with $N_M(\mathbf{y}) = L - (k+1)$, we let $\mathcal{C}(\mathbf{y}) = \{i \in [1 : L] : y^i \neq \mathbf{m}\}$, and have:

$$p_{\theta,k+1}^{G_{\phi,2}}(\mathbf{y} | \mathbf{x}_k) = \sum_{\mathbf{z} \in \mathcal{V}} p_{\theta,k+1}^{G_{\phi,2}}(\mathbf{y}, \mathbf{z} | \mathbf{x}_k)$$

$$= \sum_{\mathbf{z} \in \mathcal{V} : z^i = y^i, \forall i \in \mathcal{C}(\mathbf{y})} \prod_{j=1}^L \text{Cat}\left(z^j; D_\theta^j(\mathbf{x}_k)\right) \text{Cat}\left(\mathcal{C}(\mathbf{y}); G_{\phi,2}^{k+1}(\mathbf{z}, \mathbf{x}_k)\right)$$

$$= \prod_{i \in \mathcal{C}(\mathbf{y})} \text{Cat}\left(y^i; D_\theta^i(\mathbf{x}_k)\right) \sum_{\mathbf{z} \in \mathcal{V}^L} \prod_{j=1}^L \text{Cat}\left(z^j; D_\theta^j(\mathbf{x}_k)\right) \text{Cat}\left(\mathcal{C}(\mathbf{y}); G_{\phi,2}^{k+1}(\mathbf{z}^{-\mathbf{y}}, \mathbf{x}_k)\right)$$

$$= \prod_{i \in \mathcal{C}(\mathbf{y})} \text{Cat}\left(y^i; D_\theta^i(\mathbf{x}_k)\right) F_{\theta,\phi,2}^{k+1}(\mathbf{x}_k, \mathbf{y}) \quad (36)$$

where

$$F_{\theta,\phi,2}^{k+1}(\mathbf{x}_k, \mathbf{y}) := \mathbb{E}_{\mathbf{z} \sim D_\theta(\mathbf{x})}\left[\text{Cat}\left(\mathcal{C}(\mathbf{y}); G_{\phi,2}^{k+1}(\mathbf{z}^{-\mathbf{y}}, \mathbf{x})\right)\right], \quad (37)$$

and for $\mathbf{z}, \mathbf{y} \in \mathcal{V}^L$, we denote by $\mathbf{z}^{-\mathbf{y}} \in \mathcal{V}^L$ the sequence which is the same as $\mathbf{z}$ except with $z^i$ replaced by $y^i$ for all $i \in \mathcal{C}(\mathbf{y})$. Note by our assumption on $D_\theta$ that this is 0 if $y^i \neq x_k^i$ in a position where $y^i, x_k^i \neq \mathbf{m}$. Also note that the proof was the exact same as in Subsection A.1.6.

### A.4.2 Markov Chain Setup and ELBO for P2 Planner

Now we observe how to apply Proposition A.7 to get an ELBO for $p_\theta^{G_{\phi,2}}$. We will then specialize this to the case of P2-TopK, and discuss the difficulties in obtaining a "regularized" approximation similar to Corollary A.10 which lends itself to a computationally viable approximation as in Subsection 3.4.

By equation 36, we have $p_\theta^{G_{\phi,2}}(\mathbf{x}) = \mathbb{P}(X_L^{G_{\phi,2},\theta} = \mathbf{x})$, where $X^{G_{\phi,2},\theta}$ is the time homogenous Markov chain on $\mathcal{V}^L$ with transition matrix given for $\mathbf{x}, \mathbf{y}$ by:

$$Q^{\theta,\phi,2}(\mathbf{y},\mathbf{x}) = \prod_{i \in \mathcal{C}(\mathbf{y})} \mathrm{Cat}\left(y^i; D_\theta^i(\mathbf{x})\right) F_{\theta,\phi,2}^{L-N_M(\mathbf{y})}(\mathbf{x},\mathbf{y}) \tag{38}$$

when $N_M(\mathbf{y}) = N_M(\mathbf{x}) - 1$ and $y^i = x^i, \forall i \in \mathcal{C}(\mathbf{y}) \cap \mathcal{C}(\mathbf{x})$ and 0 otherwise.

Once again, to obtain an ELBO for $p_\theta^{G_{\phi,2}}$ using Proposition A.7, we select any family of transition matrices $R(\cdot,\cdot;\mathbf{x}_0)$ parameterized by $\mathbf{x}_0 \in \mathcal{V}^L$ determining a family Markov chains $Y^{\mathbf{x}_0}$ such that equation 10 holds.

We choose, as in Subsection A.1.4, $Y^{\mathbf{x}_0} = Y^{G_{\phi,2},\mathbf{x}_0}$ to be the Markov chain with rate matrix obtained from replacing $D_\theta^i(\mathbf{x})$ with $\delta(x_0^i)$ in equation 38. Recalling the definition of $F_{\theta,\phi,2}$ from equation 37, this yields $R(\cdot,\cdot;\mathbf{x}_0)$ to be $R^{G_{\phi,2}}(\cdot,\cdot;\mathbf{x}_0)$ given by, for $\mathbf{x}, \mathbf{y} \in \mathcal{V}^L$:

$$R^{G_{\phi,2}}(\mathbf{y},\mathbf{x};\mathbf{x}_0) = \prod_{i \in \mathcal{C}(\mathbf{y})} \mathrm{Cat}\left(y^i; \delta(x_0^i)\right) \mathrm{Cat}\left(\mathcal{C}(\mathbf{y}); G_{\phi,2}^{L-N_M(\mathbf{y})}(\mathbf{x}_0,\mathbf{x})\right) \tag{39}$$

when $N_M(\mathbf{y}) = N_M(\mathbf{x}) - 1$ and 0 otherwise.

Note that indeed equation 10 holds, since at the $k$'th step $Y_k^{\mathbf{x}_0}$ always has $L - k$ masks, with the positions of the masks determined by sampling from the planner at each step.

Applying now Proposition A.7 with this choice of $R^{G_{\phi,2}}(\mathbf{y},\mathbf{x})$, we obtain:

**Proposition A.14.** *For any P2-style collection of planners $G_{\phi,2}^k, k \in [1, L]$, let $p_\theta^{G_{\phi,2}}$ denote the distribution of $\mathbf{x}_L$ obtained via the iterative sampling scheme Alg. 4. Then we have the following ELBO:*

$$\log(p_\theta^{G_{\phi,2}}(\mathbf{x}_0)) \geq \mathcal{E}_1^{\theta,\phi,2}(\mathbf{x}_0) + \mathcal{E}_2^{\theta,\phi,2}(\mathbf{x}_0),$$

$$\mathcal{E}_1^{\theta,\phi,2}(\mathbf{x}_0) = \sum_{k=0}^{L-1}\left[\mathbb{E}_{\mathbf{x}_k \sim r_k^{G_{\phi,2}}(\cdot;\mathbf{x}_0)}\left[\sum_{\mathbf{y} \in \mathcal{X}_{L-k-1}(\mathbf{x_0})} Cat(\mathcal{C}(\mathbf{y}); G_{\phi,2}^{k+1}(\mathbf{x}_0,\mathbf{x}_k))\times\right.\right.$$

$$\left.\left.\times \sum_{i \in \mathcal{C}(\mathbf{y})} \log\left(Cat(x_0^i; D_\theta^i(\mathbf{x}_k))\right)\right]\right]$$

$$\mathcal{E}_2^{\theta,\phi,2}(\mathbf{x}_0) = -\sum_{k=0}^{L-1}\left[\mathbb{E}_{\mathbf{x}_k \sim r_k^{G_{\phi,2}}(\cdot;\mathbf{x}_0)}\left[\sum_{\mathbf{y} \in \mathcal{X}_{L-k-1}(\mathbf{x_0})} Cat(\mathcal{C}(\mathbf{y}); G_{\phi,2}^{k+1}(\mathbf{x}_0,\mathbf{x}_k))\times\right.\right.$$

$$\left.\left.\times \log\left(\frac{Cat(\mathcal{C}(\mathbf{y}); G_{\phi,2}^{k+1}(\mathbf{x}_0,\mathbf{x}_k))}{F_{\theta,\phi,2}^{k+1}(\mathbf{x}_k,\mathbf{y})}\right)\right]\right]$$

*where $r_k^{G_{\phi,2}}(\mathbf{x};\mathbf{x}_0) = \mathbb{P}(Y_k^{\mathbf{x}_0} = \mathbf{x})$ for $Y^{\mathbf{x}_0}$ the discrete time Markov chain with rate matrix equation 39 and $Y_0^{\mathbf{x}_0} = (\mathbf{m}, \ldots, \mathbf{m})$, and here we recall the notation $\mathcal{X}_k(\mathbf{x}_0)$ from equation 1.*

Specializing to P2 Top-k, where $G_{\phi,2}$ is as in equation 33 so the jump positions of $Y^{\mathbf{x}_0}$ are deterministic and defined recursively via (suppressing the dependence on $\mathbf{x}_0$ in the notation):

$$I_k = \mathrm{Top\text{-}k}_{i \in [1:L]} \mathrm{Cat}(i; \hat{G}_\phi^\eta(\mathbf{x}_0, Y_{k-1})), k = 1, \ldots, L, I_0 = \emptyset \tag{40}$$

$$Y_k^i = \begin{cases} x_0^i, & i \in I_k \\ \mathbf{m}, & i \notin I_k, \end{cases}$$

We have $G_{\phi,2}^{k+1} = \delta(I_{k+1})$. So $\mathcal{E}_1^{\theta,\phi,2}$ becomes:

$$\mathcal{E}_1^{\theta,\phi,\text{top-k}}(\mathbf{x}_0) = \sum_{k=0}^{L-1} \sum_{i \in I_{k+1}} \log\left(\text{Cat}(x_0^i; D_\theta^i(Y_k))\right).$$

$\mathcal{E}^{\theta,\phi,2}(\mathbf{x}_0)$ similarly becomes:

$$\mathcal{E}^{\theta,\phi,\text{top-k}}(\mathbf{x}_0) = \sum_{k=0}^{L-1} \log\left(F_{\theta,\phi,2}^{k+1}(Y_k, Y_{k+1})\right).$$

Recalling the definition of $F_{\theta,\phi,2}^{k+1}$, we have here that:

$$F_{\theta,\phi,2}^{k+1}(Y_k, Y_{k+1})$$

$$= \sum_{\mathbf{z} \in \mathcal{V}^L : \text{Cat}(i; \hat{G}_\phi^\eta(\mathbf{z}^{-Y_{k+1}}, Y_k)) < \text{Cat}(j; \hat{G}_\phi^\eta(\mathbf{z}^{-Y_{k+1}}, Y_k))), \forall j \in I_{k+1}, i \notin I_{k+1}} \prod_{i=1}^L \text{Cat}(z^i; D_\theta^i(Y_k))$$

$$\geq \prod_{i \notin I_{k+1}} \text{Cat}(x_0^i; D_\theta^i(Y_k)),$$

by definition of $I_{k+1}$. Thus we have:

$$\mathcal{E}_1^{\theta,\phi,\text{top-k}}(\mathbf{x}_0) + \mathcal{E}^{\theta,\phi,\text{top-k}}(\mathbf{x}_0)$$

$$\geq \sum_{k=0}^{L-1} \left( \sum_{i \notin I_{k+1}} \log(\text{Cat}(x_0^i; D_\theta^i(Y_k))) + \sum_{i \in I_{k+1}} \log(\text{Cat}(x_0^i; D_\theta^i(Y_k))) \right)$$

$$= \sum_{k=0}^{L-1} \sum_{i=1}^{L} \log(\text{Cat}(x_0^i; D_\theta^i(Y_k))) = \sum_{k=0}^{L-1} \sum_{i=1, Y_k^i = \mathbf{m}}^{L} \log(\text{Cat}(x_0^i; D_\theta^i(Y_k))),$$

where in the last step we recall that we impose $D_\theta^i(Y_k) = \delta(Y_k^i)$ if $Y_k^i \neq \mathbf{m}$, and for any $i$ such that $Y_k^i \neq \mathbf{m}$, $Y_k^i = x_0^i$. The resulting ELBO specializing Proposition A.14 to P2-Topk is just the same as that for greedy ancestral in Corollary A.9, except that the paths are determined by the P2-style sequence of planners:

**Corollary A.15.** *For $p_\theta^{top\text{-}k}$ the distribution of $\mathbf{x}_L$ from Alg. 4 with $G_\phi$ as in equation 33, we have:*

$$\log(p_\theta^{top\text{-}k}(\mathbf{x}_0)) \geq \mathcal{E}^{\theta,top\text{-}k}(\mathbf{x}_0),$$

$$\mathcal{E}^{\theta,top\text{-}k}(\mathbf{x}_0) = L\mathbb{E}_{k \sim Unif([0:L-1])}\left[ \sum_{i=1, Y_k^i = \mathbf{m}}^{L} Cat(x_0^i; D_\theta^i(Y_k^{\mathbf{x_o}})) \right],$$

*where $Y^{\mathbf{x_o}}$ satisfies the recursion equation 40.*

However, regularizing the $\delta$ in the definition of the P2-Topk planner equation 33 to make it have full support, and hence be better approximated by a uniformly random style of sampling, quickly reveals that using such the ELBO becomes computationally prohibitive, even using e.g. the soft-max gumbel noise trick. Recall this is precisely what we did for the greedy ancestral planner using the softmax approximation done in Corollary A.10. This additional overhead arises because for each randomly sampled time step $k$, one would need not only to simulate $Y_k^{\mathbf{x_o}}$ to time $k$, which requires $k$ function evaluations of the planner, but also to compute and sum over all of the $\binom{L}{k-1}$ weights corresponding to the $\binom{L}{k-1}$ values of $\mathbf{y}$ which are possible locations for the next jump in the reference dynamics in lower bound Proposition A.14. For this reason, even though we use P2-Topk sampling in our experiments, we assume the role of remasking is relatively minimal compared to the unmasking selection process, so that the series of approximations inspired by Corollary A.10 outlined in §3.4 and detailed in §A.2.3 still results in a training objective more reflective of the sampling process than the vanilla loss equation 1.

Also note that, in P2 (Peng et al., 2025a), the ELBO result was proved assuming a continuous-time dependent, randomly sampled unmasking process, although the top-k sampling procedure Alg. 4 with $G_{\phi,2}$ as in equation 33 was used in practice. This creates a similar mismatch between training and sampling that is present in vanilla DLMs vs. greedy ancestral - see Proposition 3.1. The results shown here are for the practically used P2-Topk sampling procedure, hence there is little to no overlap in the analysis performed between the manuscripts.

### A.5 OTHER SAMPLING METHODS WITH MULTIPLE DENOISING POSITIONS AND/OR REMASKING

We show here how one can view other sampling strategies in the literature as special cases of Alg. 4 and hence train also for these strategies using the result of Proposition A.14.

**RDM sampling Zheng et al. (2023).** is captured by taking

$$G_{\phi,2}^k(\mathbf{z}, \mathbf{x}) = \delta \left( \text{Top-k}_{i \in [1:L]} \left( \text{Cat} \left( z^i; \hat{D}_\theta^i(\mathbf{x}) \right) \right) \right),$$

where $\hat{D}_\theta$ is as in equation 35. This strategy uses the denoisers logits on both masked and unmasked positions, and only keeps tokens which fall in the $\text{top}-k$ threshold.

**Top-j Block denoising Nie et al. (2025b).** is captured by taking

$$G_{\phi,2}^k(\mathbf{z}, \mathbf{x}) = \delta \left( \text{Top-j}_{i \in B(\mathbf{x})} \left( \text{Cat} \left( z^i; D_\theta^i(\mathbf{x}) \right) \right) \cup \{i : x^i \neq \mathbf{m}\} \right).$$

Here previously unmasked tokens are kept, $B(\mathbf{x})$ is a block of masked positions of $\mathbf{x}$ of a predetermined fixed size, and $j$ is a fixed desired number of positions to be denoised at each step. Note that time here should instead only run to $\lfloor L/j \rfloor$ rather than $L-1$, since at this point all tokens will be clean.

**Confidence Thresholding Wu et al. (2025).:** In this strategy, the number of tokens to be clean at a given time step is adaptive to how many tokens exceed a certain confidence threshold. This means that one would need to allow $k$ in Alg. 4 to change adaptively based on $\mathbf{z}$ and $\mathbf{x}$. While we don't provide details here for the sake of brevity, one can make such a modification to the sampling algorithm and follow the general principle of writing down the transitions of the discrete time Markov chain for the generation dynamics in terms of a planner and comparing with paths which use a supervised planner using Proposition A.7 to obtain a rigorous ELBO in such regimes.

## B EXPERIMENTAL DETAILS

### B.1 PROTEIN SEQUENCE GENERATION

**Setup.** We evaluate our approach against leading protein sequence generation models. The comparison includes three discrete diffusion models—DPLM (Wang et al., 2024), EvoDiff (Alamdari et al., 2023), and ESM3 (Hayes et al., 2025)—along with the autoregressive baseline ProGen2 (Nijkamp et al., 2023). Each model generates 100 sequences for target lengths in $200, 300, \ldots, 800$. DPLM follows its standard setting, using a sequence length tied to the number of sampling steps and temperature $0.9$, with rejection resampling disabled to ensure fairness. ESM3 is sampled with temperature $1.0$, cosine noise schedule, top-$p = 1$, and 500 denoising steps. Special tokens are stripped to guarantee valid amino acid outputs.

**Evaluation.** Generated sequences are assessed using structure prediction as a proxy for functional plausibility. Specifically, we fold each sequence with ESMFold (Lin et al., 2023) and extract three structural metrics:

- **pLDDT** (predicted Local Distance Difference Test): an estimate of local per-residue confidence, defined as the expected accuracy of predicted interatomic distances. Formally, for residue $i$,

$$\text{pLDDT}(i) = 100 \times \mathbb{E} \left[ 1 - \frac{|d_{ij}^{\text{pred}} - d_{ij}^{\text{true}}|}{d_{ij}^{\text{true}}} \right]_{j \in \mathcal{N}(i)},$$

where $d_{ij}$ denotes pairwise distances and $\mathcal{N}(i)$ indexes local neighbors. The reported score is the average over residues.

- **pTM** (predicted Template Modeling score): measures global structural similarity between predicted and true structures, adapted from the TM-score (Zhang & Skolnick, 2004). Given length $L$ and alignment $u(i)$ between residues,

$$\text{pTM} = \max_{\text{alignments } u} \frac{1}{L} \sum_{i=1}^{L} \frac{1}{1 + \left(d_{i,u(i)}/d_0(L)\right)^2},$$

where $d_{i,u(i)}$ is the distance deviation and $d_0(L)$ is a length-dependent scaling factor.

- **pAE** (predicted Alignment Error): estimates the expected positional error in aligning residue $i$ of the predicted structure to residue $j$ of the true structure. Formally,

$$\text{pAE}(i,j) = \mathbb{E}\left[\|x_i^{\text{pred}} - x_j^{\text{true}}\|_2\right],$$

averaged across all residue pairs. Lower values indicate better global alignment.

Since high local confidence can mask poor global geometry (e.g., high pLDDT but high pAE), we combine these into a binary *foldability* criterion: the fraction of sequences with pLDDT $> 80$, pTM $> 0.7$, and pAE $< 10$. This composite measure penalizes degenerate patterns (e.g., repetitive "ABABAB" sequences) that often achieve misleadingly high scores in isolation.

In addition to structural metrics, we compute two distributional statistics that capture whether the model avoids mode collapse:

- **Token entropy**: measures per-position variability of amino acid usage across generated sequences. Let $\mathcal{A}$ denote the set of amino acids observed in the generated set, and $p(a)$ the empirical frequency of amino acid $a \in \mathcal{A}$. The token entropy is

$$H = -\sum_{a \in \mathcal{A}} p(a) \log p(a),$$

with higher values indicating richer amino acid usage.

- **Sequence diversity**: quantifies variability across full sequences. For a batch $\{\mathbf{x}^{(1)}, \ldots, \mathbf{x}^{(B)}\}$ of equal length $L$, define pairwise identity as

$$\text{Id}(\mathbf{x}^{(m)}, \mathbf{x}^{(n)}) = \frac{1}{L} \sum_{i=1}^{L} \mathbf{1}\left[x_i^{(m)} = x_i^{(n)}\right].$$

The sequence diversity is then

$$\text{Diversity} = 1 - \frac{2}{B(B-1)} \sum_{1 \leq m < n \leq B} \text{Id}(\mathbf{x}^{(m)}, \mathbf{x}^{(n)}),$$

which ranges from 0 (identical sequences) to 1 (completely dissimilar).

**Training Details for the 150M DLM**  We train a 150M-parameter masked diffusion model on protein data. Training follows the open-source DPLM implementation[1], using the same transformer backbone as DPLM-150M and ESM2-150M. The model is trained from scratch for 500k steps with an effective batch size of 320k tokens per iteration, achieved via multi-GPU, multi-node training with gradient accumulation on L40, H100, and A100. The dataset is UniRef50, which contains roughly 40M protein sequences clustered at 50% sequence identity, ensuring non-redundant coverage. UniRef50 is a widely adopted resource for protein language modeling.

---

[1]https://github.com/bytedance/dplm

## B.2 TEXT GENERATION

### B.2.1 SETUP

**Dataset.** We use the OPENWEBTEXT (OWT) corpus[2], a large-scale collection of English web pages curated to match the distribution of OpenAI's WebText. The dataset is preprocessed with the GPT-2 byte-pair tokenizer and sequences are wrapped or truncated to a maximum length of $L = 1024$ tokens. For evaluation, a held-out split is reserved to compute distributional metrics.

**Baselines.** We compare against a wide range of autoregressive and diffusion-based language models:

- **AR (GPT-style):** standard autoregressive language model trained on OWT.
- **MDLM (Sahoo et al., 2024):** masked diffusion language model with uniform random masking.
- **MDLM + FB / DFM (Campbell et al., 2024):** MDLM augmented with forward–backward or discrete flow matching correctors.

All checkpoints are reused from prior work for comparability.

**Training.** We follow the same training configurations as in MDLM (Sahoo et al., 2024). Unless otherwise noted, models are initialized with the same GPT-2 tokenizer and architecture. Training details are:

- Optimizer: AdamW, with learning rate $3 \times 10^{-4}$ and linear warmup of 2.5k steps.
- Batch size: 32 sequences per GPU, 16 H100 GPUs in total.
- Gradient clipping: 0.1.
- Training steps: $228k$ steps, with checkpoints saved every $19k$ steps.

**Sampling and Decoding.** We use P2 self-planning (Peng et al., 2025a) in sampling. Unless otherwise specified, we use:

- 64-bit floating-point, as prior work showed 32-bit sampling led to reduced diversity.
- nucleus sampling with $p = 0.9$.
- 5,000 sequences per model–sampler pair.

**Evaluation.** We evaluate generated text against the held-out OWT distribution. Metrics include:

- **MAUVE (Pillutla et al., 2021):** MAUVE directly compares the generated distribution from a text generation model to a distribution of human-written text using divergence frontiers. Generated samples and a ground-truth corpus of data are embedded using an external language model. The two distributions are compared in the embedding space using an area under the divergence curve to summarize both Type I and Ttype II errors.
- **Generative perplexity (Gen PPL.):** cross-entropy of generated samples under a pretrained language model. This measures the concordance of the model under consideration and a strong pretrained language model on the generated text. This can be problematic as it only considers generated text. Some distributions of text make it much simpler (and thereby better Gen PPL) even when not satisfactory to a human reader.
- **Entropy:** average per-token entropy of the generated distribution. This measures the token diversity of the generated text. Model collapse can be evaluated if entropy decreases substantially.

MAUVE is a most robust indicator, while Gen PPL. can be gamed by overconfident sampling schedules. For all evaluation metrics we match the settings of Wang et al. (2025a).

---

[2]http://Skylion007.github.io/OpenWebTextCorpus

### B.2.2 ABLATION OF SAMPLING METHODS

Table 5 compares inference-time planners while holding the denoiser fixed. Across all step budgets, P2-Self sampling consistently yields the best quality–diversity tradeoff: it achieves the highest MAUVE and the lowest generative perplexity, with only a mild reduction in entropy relative to Greedy and Probability Margin. The gap is most pronounced in the fast-sampling regime. At $T = 64$, Greedy decoding as in MaskGIT (Chang et al., 2022) substantially underperforms, suggesting that purely myopic confidence selection can lock the trajectory into suboptimal local choices when the model is imperfect. Probability Margin (Kim et al., 2025) improves over Greedy at intermediate and large $T$, but remains below P2-Self and does not scale monotonically with more steps, indicating that margin-based heuristics are less robust to sampling budget. Overall, these results reinforce that planner choice remains a crucial degree of freedom at inference, and that the P2 framework (Peng et al., 2025a) provides a more reliable path selection rule for converting denoiser confidence into high-quality generations.

Table 5: Comparison of Greedy, Probability Margin, and P2-Self Sampling across metrics and number of sampling steps.

| Metric | Method | $T = 32$ | $T = 64$ | $T = 128$ |
|---|---|---|---|---|
| MAUVE | Greedy | 0.011 | 0.021 | 0.056 |
| | Probability Margin | 0.011 | 0.039 | 0.051 |
| | P2-Self | 0.013 | 0.046 | 0.067 |
| Gen PPL | Greedy | 44.34 | 34.18 | 29.38 |
| | Probability Margin | 44.27 | 34.37 | 29.39 |
| | P2-Self | 40.19 | 29.98 | 24.33 |
| Entropy | Greedy | 5.35 | 5.30 | 5.25 |
| | Probability Margin | 5.35 | 5.30 | 5.25 |
| | P2-Self | 5.32 | 5.24 | 5.16 |

## B.3 CODE GENERATION

### B.3.1 SETUP

We evaluate PAPL on code generation. Training follows the Open-dLLM framework (Peng et al., 2025b), where we initialize from Qwen2.5-Coder and adapt it to the diffusion setting with bidirectional attention. The model is trained on the FineCode corpus, which integrates the *opc-annealing-corpus* and *Ling-Coder-SyntheticQA*, providing both algorithmic and synthetic QA data. Following the Open-dLLM recipe, each training sequence is randomly masked with a ratio uniformly sampled from $[0, 1]$, and the model is optimized with a cross-entropy loss over masked positions, weighted by the inverse of the mask ratio. This ensures compatibility with the released training pipeline, data recipes, and evaluation protocols.

Evaluation is conducted on the following benchmarks:

- HUMANEVAL (Chen et al., 2021): 164 hand-written Python programming problems designed to test functional correctness.

- MBPP (Austin et al., 2021b): 974 crowd-sourced Python problems of varying difficulty.

- HUMANEVAL+ and MBPP+: augmented variants that extend the original datasets with paraphrased prompts and additional test cases.

For code infilling, we use HUMANEVAL-INFILL (Bavarian et al., 2022) and the Python subset of SANTACODER-FIM (Sagtani et al., 2025). We report pass@1 and pass@10 for HUMANEVAL and MBPP, and exact match for SANTACODER-FIM, following the official evaluation protocols. To examine length control, we additionally test under different initial mask spans (4, 8, 16, 32, 64) and report their averaged results.

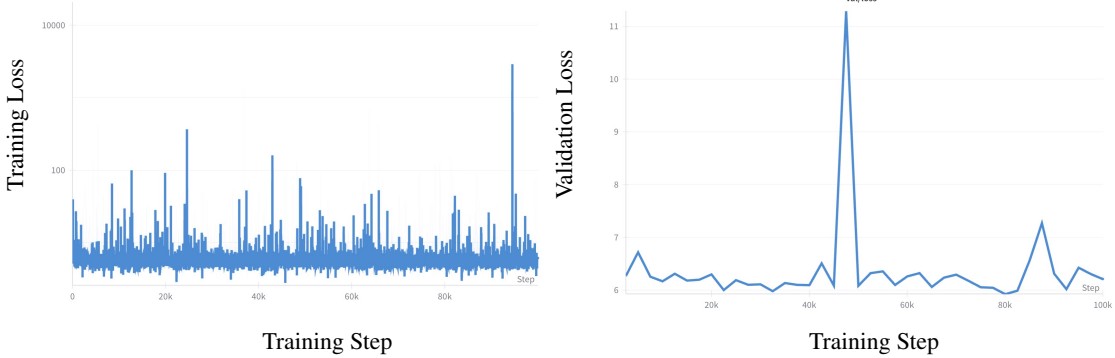

Figure 5: Training with pure PAPL loss ($\tau = 1$) leads to unstable behavior, with large fluctuations in training (left) and poor convergence on validation (right).

### B.3.2 ABLATION OF SAMPLING METHODS

We ablate the role of the sampling strategy while keeping the denoiser and training setup fixed. Table 6 compares P2-self against several commonly used alternatives: (i) vanilla ancestral sampling with uniform unmasking, (ii) greedy ancestral sampling that always selects the highest-confidence positions, (iii) entropy-based confidence ordering, and (iv) the TopK-margin heuristic. Across all six code benchmarks, P2-self is consistently the strongest method. The improvements are most pronounced in Pass@1, where P2-self exceeds the best baseline by large margins (e.g., 20.8 vs. 12.6 on HumanEval and 16.7 vs. 9.2 on MBPP), and it also yields steady gains in Pass@10 and in the infilling and SantaCoder evaluations. Among baselines, confidence-aware orderings (greedy ancestral and entropy-based confidence) substantially outperform vanilla ancestral sampling, indicating that exploiting denoiser uncertainty during decoding is critical for code generation. However, these heuristics remain myopic: they make locally optimal unmasking decisions without explicitly reasoning about longer-range dependencies or future refinement steps, which limits their ability to traverse the high-probability decoding trajectories that matter at inference. TopK-margin is weaker than other confidence-based methods, suggesting that hard-thresholding margins can be overly conservative and discard useful intermediate updates. Overall, this ablation shows that simply reordering unmasking by instantaneous confidence is insufficient; self-planning with path-level lookahead is necessary to fully close the training–inference gap and achieve robust gains.

| Method | HumanEval P@1 | P@10 | HumanEval+ P@1 | P@10 | MBPP P@1 | P@10 | MBPP+ P@1 | P@10 | HumanEval Infill P@1 | SantaCoder P@1 |
|---|---|---|---|---|---|---|---|---|---|---|
| **P2-self** | **20.8** | **38.4** | **17.6** | **35.2** | **16.7** | **38.4** | **23.9** | **53.6** | **77.4** | **56.4** |
| Vanilla Ancestral | 3.3 | 18.3 | 3.2 | 15.2 | 1.8 | 13.2 | 2.9 | 21.8 | 72.7 | 53.8 |
| Greedy Ancestral | 9.3 | 31.1 | 8.1 | 28.7 | 5.3 | 29.0 | 8.7 | 41.5 | 75.1 | 53.7 |
| Entropy-based Confidence | 12.6 | 35.4 | 10.9 | 29.9 | 9.2 | 36.8 | 15.2 | 50.7 | 75.1 | 53.2 |
| TopK-Margin | 7.6 | 27.4 | 6.5 | 26.2 | 3.9 | 24.0 | 6.2 | 33.5 | 75.0 | 54.4 |

Table 6: Performance comparison across coding benchmarks for sampling different methods.

## C ADDITIONAL RESULTS

### C.1 UNSTABLE TRAINING WITH PURE PAPL LOSS

We investigate the effect of training solely with the PAPL loss under the default temperature setting ($\tau = 1$). As shown in Figure 5, training exhibits large fluctuations and fails to achieve stable convergence on the validation loss. We hypothesize that this instability arises from the denoiser becoming overly confident: the path weights bias the model toward a narrow set of generation paths, which reduces diversity and leads to overfitting.

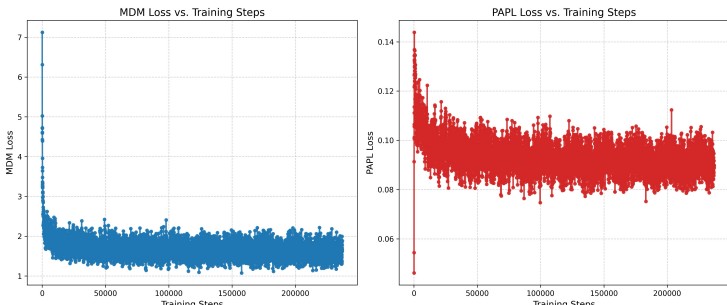

Figure 6: **Vanilla MDM vs PAPL Training Curves.** Early in training, the PAPL loss remains small because the denoiser has not yet formed meaningful beliefs about the correct token positions, causing the planner-dependent weights $w_i$ to be close to zero. As the model begins to identify correct positions with higher certainty, these weights increase, leading to a temporary rise in the PAPL loss. Once the denoiser becomes sufficiently confident, the loss decreases and eventually mirrors the behavior of the standard MDM loss. These dynamics provide additional evidence that PAPL naturally adapts its emphasis as the model's confidence improves, stabilizing precisely when confidence becomes a reliable signal.

## C.2 COMPARISON OF TRAINING CURVES WITH VANILLA MDM LOSS

In Figure 6 we observe that the PAPL loss exhibits distinct training curves compared to the vanilla MDM loss. In particular, initially the PAPL loss is small, because the denoiser has not established any confidence about the correct token positions, and hence the weights $w^i$ are near-zero. As the denoiser gains confidence about the correct token positions during the initial training stages, the weights and hence loss increase, before decreasing with a curve similar to that of the vanilla MDM loss.

## C.3 EMPIRICAL ESTIMATION OF THE EFFECT OF THE APPROXIMATION STEPS

| Loss | Greedy | E1-softmax | E2-softmax | Vanilla-softmax | PAPL-softmax | E1-rand | E2-rand | PAPL | Vanilla |
|---|---|---|---|---|---|---|---|---|---|
| Value | 23294 | .002 | 3.364 | 17.897 | 17.897 | .002 | 3.365 | 18.362 | 18.362 |

Table 7: Ablation of the approximation steps used to obtain the PAPL loss.

Here we provide Table 7, where we empirically estimate the effect of the series of approximations used to move from the true ELBO of Proposition 3.2 specialized to Greedy Ancestral (see Corollary A.9) the PAPL loss equation 7. Here, rather than sampling a random timestep $k$, we sum along the entire generation trajectory. Greedy is the full greedy loss from Corollary A.9, calculated along the true greedy path from Corollary A.9. We remark that this term is extremely large due to the crude lower bound on $\mathcal{E}_2^{\theta,\phi}$ used to arrive at A.9 from Proposition 3.2. $E1$-softmax is the contribution of the term $\mathcal{E}_1^{\theta,\phi,\tau}$ from specialization of Proposition 3.2 to the softmax planner from Corollary A.10 to the true softmax loss, and $E2$-softmax is the contribution of $\mathcal{E}_2^{\theta,\phi,\tau}$. PAPL-softmax is the PAPL loss equation 7 but with $\mathbf{x}_k$ sampled from the softmax path from Corollary A.10, and Vanilla-softmax is the vanilla loss from equation 1 but with $\mathbf{x}_k$ sampled from the softmax path. Finally, $E1$-rand and $E2$-rand are the contribution of $\mathcal{E}_1^{\theta,\phi,\tau}, \mathcal{E}_2^{\theta,\phi,\tau}$ Corollary A.10 but taken along a random path, and PAPL and Vanilla are the actual PAPL and Vanilla losses, which are computed along the random path by definition.

The losses are estimated via accumulating along the entire trajectories for a batch-size of 1024, using the PAPL-fine-tuned protein MDM with $\alpha = 5$ and $\tau = 1$.

We remark that the computations of the $E2$ terms are highly unstable. However, we can observe still in Table 7 the following: As predicted by Proposition 3.1, Greedy Loss is much larger than Vanilla Loss - Vanilla Loss does not provide an upper bound for greedy ancestral sampling, so we would expect the true upper bound to be larger. Moreover, as predicted by Proposition A.11, when computed

along the same path, $E_1$ is dominated by the vanilla loss. Finally, we observe that indeed the path taken in terms of unmasking order does have effect on the losses - the losses increase while taking a random path. This is expected, as it is known that taking a random path, as is trained for in vanilla MDMs, yields significantly worse sample quality than a greedy or soft-greedy path.

## C.4 HUMANEVAL PERFORMANCE ANALYSIS

We evaluated the model on 40 HumanEval tasks. While it handles straightforward problems well, performance declines sharply for tasks requiring careful constraint handling, multi-step logic, or less familiar algorithms.

**Strengths** The model performs best when tasks align with standard Python idioms or textbook solutions. In `HumanEval/12: longest`, for example, it produced the compact and idiomatic implementation in Listing 1, which is more direct than the canonical reference.

Listing 1: Model's solution for `HumanEval/12: longest`.

```python
def longest(strings: list) -> str
    | None:
    if not strings:
        return None
    return max(strings, key=len)
```

Listing 2: Canonical solution.

```python
def longest(strings: list) -> str
    | None:
    if not strings:
        return None
    maxlen = max(len(x) for x in
        strings)
    for s in strings:
        if len(s) == maxlen:
            return s
```

The model also demonstrates competence in basic algorithmic tasks. For example, `HumanEval/25: factorize` was solved with a standard trial division approach (Listing 3), and string prefix generation (`HumanEval/14`) and set-based deduplication (`HumanEval/34`) were handled correctly.

```python
def factorize(n: int) -> list[int]:
    factors = []
    while n % 2 == 0:
        factors.append(2)
        n //= 2
    i = 3
    while i * i <= n:
        while n % i == 0:
            factors.append(i)
            n //= i
        i += 2
    if n > 2:
        factors.append(n)
    return factors
```

Listing 3: Model's correct solution to `HumanEval/25: factorize`.

**Weaknesses** The most common failures stem from flawed algorithmic reasoning. In `HumanEval/9: rolling_max`, the model produced a redundant nested loop instead of the correct single-pass running maximum (Listing 4).

Another recurring issue is misinterpretation of constraints. In `HumanEval/3: below_zero`, the model ignored the requirement to detect negative balances *at any point*, checking only the final state instead (Listing 5).

Finally, there are occasional catastrophic failures, where the generated code bears no relation to the task. In `HumanEval/2: truncate_number`, for instance, the model produced irrelevant

```python
# Model (incorrect)
def rolling_max(numbers: list) -> list[int]:
    result = []
    max_val = numbers[0]
    i = 0
    while i < len(numbers):
        max_val = max(max_val, numbers[i])
        while i < len(numbers):  # Redundant nested loop
            max_val = max(max_val, numbers[i])
            i += 1
        result.append(max_val)
    return result

# Canonical
def rolling_max(numbers: list) -> list[int]:
    running_max = None
    result = []
    for n in numbers:
        running_max = n if running_max is None else max(running_max, n)
        result.append(running_max)
    return result
```

Listing 4: Incorrect vs. canonical solutions for HumanEval/9: rolling_max.

```python
# Model (incorrect)
def below_zero(operations: list[int]) -> bool:
    balance = 0
    for op in operations:
        # Missing balance update logic
        if balance < 0:
            return False
    return balance < 0

# Canonical
def below_zero(operations: list[int]) -> bool:
    balance = 0
    for op in operations:
        balance += op
        if balance < 0:
            return True
    return False
```

Listing 5: Misinterpretation of temporal constraint in HumanEval/3: below_zero.

variable assignments and returned the input unchanged, instead of applying a simple modulo operation (Listing 6).

```python
# Model (incorrect)
def truncate_number(number: float) -> float:
    a, b, c = 1, 2, 3  # Irrelevant assignments
    return number

# Canonical
def truncate_number(number: float) -> float:
    return number % 1.0
```

Listing 6: Catastrophic failure on HumanEval/2: truncate_number.

**Discussion** Overall, the model succeeds when tasks resemble common idioms or well-documented examples, but struggles as soon as prompts introduce layered constraints or require multi-step reasoning. The lack of self-checking mechanisms is evident in tautological or irrelevant code that would be immediately rejected by a human programmer. While the system can accelerate routine coding tasks, it cannot yet be relied upon for problems that demand algorithmic novelty or strict logical consistency. Improving decomposition of complex prompts and incorporating verification steps remain key directions for future work.

## D    PRACTITIONER'S GUIDE

There are two main parameters for PAPL, temperature $\tau$ and loss weight $\alpha$. For practitioners, we recommend initial settings of $\tau = 1$ and $\alpha = 1$. All experiments in this work leave $\tau = 1$. We experimented with $\alpha$ values in the range $1 \ldots 10$. We found that higher values of $\alpha$ can be more effective in the range of $\approx 5$, especially for protein models, but not beyond that. If hyperparameter tuning with PAPL, we therefore recommend starting with $\tau = 1$ and $\alpha = 1$ and doubling $\alpha$ to efficiently search the space.

We recommend this because there is no huge issue with setting $\alpha$ as low as $0$, the model will still achieve good performance, as a reminder $\alpha = 0$ recovers standard DLM training. However, you may not get the benefit of PAPL training, and slow convergence on practical inference paths.

With $\alpha$ too high, the training may become unstable, and may not be trained well on all paths. We hypothesis this may hinder performance on more out of distribution tasks, where the planner is not as confident, or may hinder performance by learning sub-optimal paths for generation by latching on to a specific path too quickly.

