# OpenReview forum: "Planner Aware Path Learning in Diffusion Language Models Training"
_ICLR.cc/2026/Conference — ICLR 2026 Oral_

### Official Review · Reviewer_TRnB · 2025-10-26

**Soundness:** 3
**Presentation:** 3
**Contribution:** 3
**Rating:** 6
**Confidence:** 3

**Summary:**

The paper addresses the training-inference mismatch in masked diffusion language models (DLMs). While DLMs are trained with uniform random masking/unmasking, they are deployed with planners that bias the denoising order (e.g., greedy, path planning). The authors prove that this mismatch causes the standard DLM ELBO to be violated at inference (Proposition 3.1). They derive a Planner-Aware ELBO (P-ELBO) that incorporates planner dynamics into the training objective (Proposition 3.2), then propose PAPL—a practical algorithm that simplifies P-ELBO into a planner-weighted cross-entropy requiring only a one-line code change. Experiments demonstrate consistent improvements: ~40% relative gain in protein foldability, up to 4× MAUVE improvement in text generation, and ~23% relative gain in code generation pass@10, all under identical model architectures and inference planners.

**Strengths:**

1. The core contribution is original and well-motivated. Proposition 3.1 formally proves that greedy planner-based sampling can violate the standard ELBO inequality (log p_θ < ELBO), meaning the training objective no longer lower-bounds the inference distribution. This is a concrete theoretical finding that justifies rethinking the training procedure, not just an empirical observation.
2. The theoretical framework is rigorous and general. The P-ELBO formulation (Proposition 3.2) unifies multiple existing planning strategies: uniform (standard DLM), greedy (MaskGIT), and soft-greedy, as special cases.
3. The implementation is practical and low-friction. PAPL reduces to standard MDLM loss augmented with planner weights and requires no architectural changes or additional forward passes during training. Algorithm 1 shows this is essentially a one-line modification, making adoption straightforward.
4. Empirical validation is comprehensive across diverse domains. The paper demonstrates improvements in protein generation (Table 1: higher pLDDT, pTM, lower pAE, +40% foldability while maintaining diversity), text generation (Table 2: consistent MAUVE and perplexity improvements across sampling budgets), and code generation (Tables 3-4: gains in both completion and infilling). The ablations (Figure 2, Figure 4) support design choices and show robustness.
5. The presentation is clear and well-structured. The method section logically progresses from dynamics formulation to ELBO violation to the new objective. Experimental setups, metrics, and baselines are clearly described with sufficient detail for reproduction.

**Weaknesses:**

1. The gap between P-ELBO and implemented objective is not quantified. Section 3.4 drops the correction term by detaching gradients through the planner, converting the full P-ELBO to a simpler weighted cross-entropy. The paper provides no analysis of the induced approximation error, theoretical bounds, or empirical measurements of this gap.
2. Although the framework is generally applicable for any planner, the derived training objective and results are only for soft greedy decoding. For non-differentiable or hard planners, the correction term becomes non-trivial, and the paper does not provide a full derivation or empirical evidence beyond soft greedy-like planners.
3. Text evaluation is limited to unconditional generation. While MAUVE and generative perplexity on OpenWebText are standard, they don't test whether planner-aware training improves conditional generation capability.

**Questions:**

1. Can you measure the correction empirically at different training stages across domains? Some figures may be helpful.
2. How should practitioners choose the two hyperparameters?
3. How tight is the P-ELBO bound in practice?

---

> ### Author Response · Authors · 2025-11-22
> **Response to TRnB Part 1**
>
> We thank the reviewer for their time and effort in reviewing our work. We are encouraged that the reviewer found our work original and well motivated, with a “rigorous and general” theoretical framework, comprehensive empirical validation with “consistent improvements” across domains, and a “clear and well-structured” presentation. We now address the concerns and questions raised by the reviewer point by point.
>
> ## Variational Gap between P-ELBO and implemented version
>
> We thank the reviewer for their questions regarding the gap between the PAPL loss and the ELBO of Proposition 3.2 in practice. We aim to answer these questions through several additions to the paper.
>
> Theoretically, we first provide the detailed mathematical description of how one arrives at the PAPL loss from the specialization of Proposition 3.2 to greedy and soft-greedy planners in the new Appendix A.2.3. These steps were discussed briefly in Section 3.4 but are now greatly elaborated upon. As a reminder:
>
> 1. First, we observe that specializing Proposition 3.2 to greedy ancestral results in the ELBO of Corollary A.9. The reference paths become deterministic and far from the easily implemented "uniform at random’’ masking of the vanilla ELBO. We thus approximate the greedy planner using a ``soft greedy planner,’’ whose ELBO takes the form of Corollary A.10.
> 2. Corollary A.10’s ELBO takes the form of $E_1+E_2$, where $E_1$ is the same as the vanilla ELBO but with the uniform weights $1/(L-k)$ replaced by the softmax weights $w^i$ and where the reference path is along ``soft greedy’’ trajectories rather than uniformly at random. $E_2$ is the KL divergence between the supervised weights and the effective ones using the output of the planner.
>
> 3. We then detach gradients from the weights, using the training signal only from the logits of the denoiser. This causes $E_2$ to become constant, so we remove it from the loss.
>
> 4. We replace the ``soft greedy trajectory’’ with the uniform at random one. The purpose of using the soft-greedy approximation was to make this ablation - indeed, at least the next position to be decoded has full support on all currently masked positions, unlike the pure greedy setting.
>
> 5. At this point the loss is the same as vanilla MDM, except the uniform weights on token positions are replaced by $w^i$. Interpolating with the vanilla MDM loss with parameter alpha leads to the new weights $(1+\alpha w^i)/(L-k)$ used in the PAPL loss.
>
> While providing a rigorous analysis of the effect of each of these approximation steps is difficult, we do manage to prove  in Proposition A.11. of appendix A.2.4 that, for a fixed trajectory, replacing the uniform at random weights 1/(L-k) with softmax weights w^i has the effect of decreasing the term $E_1$ in the loss. This is likely the reason for the need of the interpolation with the vanilla MDM loss in order to increase stability.
>
> Empirically, in appendix C.1 we now compare training curves for a vanilla MDM vs one trained using the PAPL loss.  There we observe that, early in training, the PAPL loss remains small
> because the denoiser has not yet formed meaningful beliefs about the correct token positions, causing the planner-dependent weights $w^i$ be close to zero. As the model begins to identify correct positions with higher certainty, these weights increase, leading to a temporary rise in the PAPL loss. Once the denoiser becomes sufficiently confident, the loss decreases and eventually mirrors the behavior of the standard MDM loss. These dynamics provide additional evidence that PAPL naturally adapts its emphasis as the model’s confidence improves, stabilizing precisely when confidence becomes a reliable signal.
>
> Finally, we report an additional empirical result in the form of Table 7 in Appendix C.3. Here we compare three settings, greedy, softmax-based planned paths, and random paths. Here we report the loss (the negative elbo) where lower means a tighter bound.
> - **Greedy**: Here we sum along the entire generation trajectory. Greedy is the full greedy loss from Corollary A.9,  calculated along the true greedy path from Corollary A.9. This term is extremely large due to the crude lower bound on $\mathcal{E}_2$ used to arrive at Corollary A.9 from Proposition 3.2.
>
> - **Softmax**: $E1$-softmax is the contribution of the term $\mathcal{E}_1$ from the specialization to the softmax planner of Corollary A.10, and $E2$-softmax is the contribution of $\mathcal{E}_2$. PAPL-softmax is the PAPL loss but with $\mathbf{x}_k$ sampled from the softmax path from Corollary A.10, and Vanilla-softmax is the vanilla MDM loss but with $\mathbf{x}_k$ sampled from the softmax path.
>
> - **Random**: $E1$-rand and $E2$-rand are the contribution of $\mathcal{E}_1,\mathcal{E}_2$ respectively from Corollary A.10 but taken along a random path, and PAPL and Vanilla are the actual PAPL and Vanilla losses, which are computed along the random path by definition.

---

> > ### Author Response · Authors · 2025-11-22
> > **Response to TRnB Part 2**
> >
> > Analyzing the table we find the following:
> >
> >
> > - **Greedy Loss is much larger than Vanilla Loss.** As predicted by Proposition 3.1, Vanilla Loss does not provide an upper bound for greedy ancestral sampling, so we would expect the true upper bound to be larger.
> >
> > - **E1 is dominated by the vanilla loss**. As predicted by Proposition A.11, when computed along the same path.
> >
> > - **Losses increase while taking a random path**: This is expected, as it is known that taking a random path, as is trained for in vanilla MDMs, yields significantly worse sample quality than a greedy or soft-greedy path.
> >
> > Table 7
> >
> > | **Loss** | **Greedy** | **E1-softmax** | **E2-softmax** | **Vanilla-softmax** | **PAPL-softmax** | **E1-rand** | **E2-rand** | **PAPL** | **Vanilla** |
> > |-----------|------------:|---------------:|---------------:|--------------------:|-----------------:|-------------:|-------------:|----------:|-------------:|
> > | **Value** | 23294       | 0.002          | 3.364          | 17.897              | 17.897           | 0.002        | 3.365        | 18.362    | 18.362       |

---

> > > ### Author Response · Authors · 2025-11-22
> > > **Response to TRnB Part 3**
> > >
> > > ## Derivation for other planners
> > >
> > > Although indeed we derive the PAPL loss from Corollary A.10, which is  Proposition 3.2 to soft-greedy decoding, the theoretical foundation of our work Proposition 3.2 is, in fact, applicable to “hard planners.” We already provide the form of the ELBO which results from specializing to “hard” greedy decoding in Corollary A.9. While a perfectly valid ELBO, the issue lies in the number of function evaluations required of the denoiser to unmask according to the greedy reference path. Hence we approximate via the soft-greedy decoder in order to regularize the paths to be closer to choosing positions uniformly at random, and make a series of ablations (discussed in Section 3.4 and at a reviewers request, now further detailed in A.2.3) to arrive at the PAPL loss.
> > >
> > > We also discussed already how “P2 Top-k” (another “hard” planning method) can be seen to fall into a generalized version of our framework in Corollary A.14., and now discuss a how one could insert a variety of planners into the framework in Appendix A.2 and A.5.
> > >
> > > So, while we focus on deriving an implementable loss (in terms of NFEs) for greedy ancestral/soft-greedy ancestral for the sake of exposition and experiments, our theoretical foundations allow for finding the starting point (i.e. the true ELBO) for an encapsulating variety of planning methods. We have now addressed this limitation and pointed out the derivation of analogous surrogate losses to PAPL for other decoding strategies as an interesting avenue for future research in the appendix.
> > >
> > > ## Conditional generation
> > >
> > > We note that the code generation tasks in Tables 3 and 4 are conditional text tasks. These results show that PAPL improves the conditional generation capacity of at least 0.5B code models in the same setting.
> > >
> > > ## Hyperparameter setting guidance
> > >
> > > By two hyperparameters, we assume the reviewer is referring to temperature $\tau$ and weighting $\alpha$. We find that a default temperature of $\tau=1$ works well across all experiments. Furthermore, we find that $\alpha=1$ also improves performance across tasks. In Figure 4, we can see that on the protein task, $\alpha$ up to 5 increases performance. Therefore, we suggest starting with $\alpha=1$ by default, although it is possible to get further performance gains by setting $\alpha$ slightly higher depending on the setting. We have added a section of the appendix titled "Practical Guidance’’ for practitioners.
> > >
> > > ## Concluding remarks
> > >
> > > We thank the reviewer again for their time and effort during this rebuttal process. We hope our answers here have aided in clarifying all the great questions raised by the reviewer. If there are any further questions held by the reviewer, we remain eager to engage further. If the reviewer feels satisfied with our responses and new rebuttal results, we would be thrilled if the reviewer would consider a fresher evaluation of our work with a potential score upgrade.

---

### Official Review · Reviewer_jzen · 2025-10-31

**Soundness:** 2
**Presentation:** 2
**Contribution:** 2
**Rating:** 4
**Confidence:** 4

**Summary:**

The paper proposes planner-aware path learning for masked diffusion language models. The authors learn a model  $G_\phi(z, x_k)$, where $z$ is a sample from the denoiser probabilities and $x_k$ is the k-th state, which outputs the next index to unmask.

The authors derive an evidence lower bound to lower bound the denoiser + planner log-likelihood, however as the objective requires $L^2$ function calls for sequences of length $L$, the authors propose a different tractable objective. The tractable objective re-weights the denoiser log-likelihood with an extra planner term.

**Strengths:**

See questions

**Weaknesses:**

See questions

**Questions:**

1. The proof for the counter-example in proposition 3.1 requires that the denoisers are inconsistent. However, uniform un-masking with inconsistent denoisers can lead to inconsistent sampling as well.

It would be beneficial to a reader to see a re-written proposition 3.1, as the authors construct a counter-example rather than ***prove*** that greedy decoding does not sample from a unique learned joint distribution. Under the constructed counter-example, even uniform sampling would not sample from a single joint distribution as the denoisers are inconsistent.

1. If the denoiser is consistent, then would greedy sampling work? As MDLMs learn an any-order autoregressive model, would any decoding order at inference would produce valid samples?
2. If the denoisers are consistent, then is the following line correct:
    1. *The key takeaway is that the ELBO in equation 1 is only valid for the uniform unmasking process (line 247)*.
3. what does the planner learn at optimality

Experiments

1. Can the authors compare PAPL to greedy sampling in the experiments?
2. In several experiments, the gap between the MDLM+planned and MDLM+baseline with  experiments are not substantial. Can the authors provide a confidence interval?

---

> ### Author Response · Authors · 2025-11-22
> **Response to jzen Part 1**
>
> We value the reviewer for their time and feedback during this review process. We now directly answer the main clarification points raised in the review.
>
> ## Questions about Proposition 3.1
>
> We acknowledge the reviewers point in that our proposition assumes the denoisers are inconsistent. We believe there may be a potential small misunderstanding that we hope to clear up.
>
> We begin by first noting that by inconsistent denoiser we refer to ones that are not perfectly trained—i.e. Denoisers that do not lead to the forward and reverse processes to match. Please note that this is almost trivially true for all reasonable DLMs, as learning with large networks is never perfect. This also means—as the reviewer correctly notes—that uniform unmasking with an inconsistent/imperfect denoiser is also not the correct sampling strategy. More broadly, **any sampling strategy** remains inconsistent if the DLM is trained in a way that it is agnostic of the planner, e.g. uniformly random masking during training. More crisply, this is due to the fact that consistency is a property of the denoiser and **not the path taken during inference**.
>
> The point of Proposition 3.1 is to observe that for an imperfect denoiser, modifying the means by
> which paths are chosen requires a modification of the training loss if one wants the loss to be an
> ELBO, and hence to be informative of the KL divergence between generated samples and the data distribution. Towards this goal, we simply need to provide a counterexample to the learning objective which we do so by showing the ELBO itself is invalid. It is not a statement about whether or not a given denoiser will be consistent. We understand that this point may not have been sufficiently clear in the original manuscript, and we have updated the writing (with blue text color) to increase transparency of our theory and its goals.
>
> ## Questions about planning
>
> > If the denoiser is consistent, then would greedy sampling work? As MDLMs learn an any-order autoregressive model, would any decoding order at inference would produce valid samples?
>
> If the denoiser is consistent then the generated samples do not depend on the order in which tokens were unmasked. This means the unmasking strategy at inference does not matter. Any order will lead to a sample from the same distribution. If the denoiser is consistent, there would be no purpose in planning the trajectories, but also no harm. We emphasize again that one will never have access to a perfect, “consistent” denoiser in practice.
>
> > What does the planner learn at optimality
>
> This is a great question, which we view as an opportunity to strengthen our paper. We have now included additional theory that outlines the optimality condition of the planner in Appendix A.1.5. We find that the optimally trained planner for a fixed (imperfect) denoiser chooses to unmask the ith position in the sequence $x_k$ using the information from the suggested clean tokens from the denoiser proportionally to the $q^i(x^i_0|x_k)$ from equation 4. Recall $q^i(x^i_0|x)$ gives the probability of unmasking position i to $x^i_0$ under the joint generation dynamics of the planner and denoiser. In short, this means that the optimal planner tends to assign high mass to trajectories whose sequence of single-coordinate updates yields a high likelihood of producing the observed $x_0$, which during training is supervised by the data.
>
> The proof follows almost immediately from observing that the ELBO is the expected KL divergence between $G(x_0,x_k)$ and $q^i(x^i_0|x_k)$.

---

> > ### Author Response · Authors · 2025-11-22
> > **Response to jzen Part 2**
> >
> > ## Questions about Experiments
> >
> > > Can the authors compare PAPL to greedy sampling in the experiments?
> >
> > This is a great suggestion. We have added a comparison to greedy sampling, and probability margin-based sampling for the text experiment in Table 5 of the appendix (also copied below), which shows that P2-self-plan is the most effective form of sampling when combined with PAPL.
> >
> > We have also added the results of PAPL with other planners including greedy sampling on the code task in Table 6 of the appendix also copied here for clarity. We find that P2 is the most effective sampler across all metrics.
> >
> > Table 5
> > | **Metric** | **Method**           | **T=32** | **T=64** | **T=128** |
> > |-------------|----------------------|----------:|----------:|-----------:|
> > | **MAUVE**   | Greedy               | 0.011     | 0.021     | 0.056      |
> > |             | Probability Margin   | 0.011     | 0.039     | 0.051      |
> > |             | P2-Self              | 0.013     | 0.046     | 0.067      |
> > | **Gen PPL** | Greedy               | 44.34     | 34.18     | 29.38      |
> > |             | Probability Margin   | 44.27     | 34.37     | 29.39      |
> > |             | P2-Self              | 40.19     | 29.98     | 24.33      |
> > | **Entropy** | Greedy               | 5.35      | 5.30      | 5.25       |
> > |             | Probability Margin   | 5.35      | 5.30      | 5.25       |
> > |             | P2-Self              | 5.32      | 5.24      | 5.16       |
> >
> >
> > Table 6
> > | Method | HumanEval Pass@1 | HumanEval Pass@10 | HumanEval+ Pass@1 | HumanEval+ Pass@10 | MBPP Pass@1 | MBPP Pass@10 | MBPP+ Pass@1 | MBPP+ Pass@10 | HumanEval Infill Pass@1 | SantaCoder Pass@1 |
> > |----------------------------------|------------------:|-------------------:|-------------------:|--------------------:|-------------:|--------------:|--------------:|---------------:|--------------------------:|-------------------:|
> > | **P2-self-plan (Peng et al., 2025)** | 20.8 | 38.4 | 17.6 | 35.2 | 16.7 | 38.4 | 23.9 | 53.6 | 77.4 | 56.4 |
> > | Ancestral| 3.3  | 18.3 | 3.2  | 15.2 | 1.8  | 13.2 | 2.9  | 21.8 | 72.7 | 53.8 |
> > | Greedy Ancestral (Chang et al., 2022) | 9.3  | 31.1 | 8.1  | 28.7 | 5.3  | 29.0 | 8.7  | 41.5 | 75.1 | 53.7 |
> > | Entropy-based Confidence                          | 12.6 | 35.4 | 10.9 | 29.9 | 9.2  | 36.8 | 15.2 | 50.7 | 75.1 | 53.2 |
> > | topk-margin (Kim et al., 2025)   | 7.6  | 27.4 | 6.5  | 26.2 | 3.9  | 24.0 | 6.2  | 33.5 | 75.0 | 54.4 |
> >
> >
> > > In several experiments, the gap between the MDLM+planned and MDLM+baseline with experiments are not substantial. Can the authors provide a confidence interval?
> >
> > We value the reviewers' feedback, but would like to politely push back on this point. Our experiments demonstrate consistent improvements: ~40% relative gain in protein foldability (table 1), up to 4× MAUVE improvement in text generation (Table 2), and ~23% relative gain in code generation pass@10 (Table 4), all under identical model architectures, training hyperparameters, and inference planners. We further find that our approach is robust to the choice of hyperparameters—requiring little to no tuning as the default hyperparameters work quite well.
> >
> > We respect the reviewer's request for providing confidence intervals, but for settings like text generation and code this requires training on 16 H100 GPUs for approximately a week, and thus confidence intervals requiring multiple training runs are computationally prohibitive. We wish to highlight our reporting of results is inline with community standards as the recent ICML 2025 outstanding paper (Kim et. al 2025)---whose work we greatly admire—also do not report additional seeds, presumably for the computational overhead.
> >
> > ## Concluding remarks
> >
> > We thank the reviewer again for their time and effort during this rebuttal process. We hope our answers here have aided in clarifying all the great questions raised by the reviewer. If there are any further questions held by the reviewer, we remain eager to engage further. If the reviewer feels satisfied with our responses and new rebuttal results, we would be thrilled if the reviewer would consider a fresher evaluation of our work with a potential score upgrade.
> >
> > ## References
> >
> > [1] Peng, Fred Zhangzhi, et al. "Path planning for masked diffusion model sampling." arXiv preprint arXiv:2502.03540 (2025).
> >
> > [2] Ou, Jingyang, et al. "Your absorbing discrete diffusion secretly models the conditional distributions of clean data." arXiv preprint arXiv:2406.03736 (2024).
> >
> > [3] Shih,Andy, et al. "Training and inference on any-order autoregressive models the right way." Neural Information Processing Systems (2022).
> >
> > [4] Li, Xuanlin, et al. "Discovering non-monotonic autoregressive orderings with variational inference." International Conference on Learning Representations (2021).

---

### Official Review · Reviewer_LjAj · 2025-10-31

**Soundness:** 4
**Presentation:** 3
**Contribution:** 4
**Rating:** 8
**Confidence:** 4

**Summary:**

This paper proposed PAPL, a lightweight modification to the classic MDLM training objective that brings a performance boost. The authors first indicate a problem in the domain of diffusion language models, the mismatch between training and evaluation. Based on this insight, the authors derived an analytical form of the planner-aware ELBO (P-ELBO) and show that P-ELBO is theoretically general. Given that P-ELBO is computationally expensive, the authors proposed a cheap approximation, PAPL, and show that across protein modeling, language modeling, and coding benchmarks, adding PAPL to the MDLM loss can boost performance without extra computational expense.

**Strengths:**

* The storyline of the paper is clear. The paper aims to address a significant problem in the field of diffusion language models, and its solution is convincing.
* The mathematical foundation is abundant. I especially appreciate that the authors proved the P-ELBO in an analytical form. The theory part is valuable and useful for the community.
* The experimental analysis is solid, and the empirical results are strong. I am particularly surprised by the increase in pass@10 in the coding part.
* The method is simple and practical. With some minimal changes to the training objective, PAPL can improve downstream performance.

**Weaknesses:**

* As a person with proficiency in discrete diffusion, I have to say the math is a bit confusing. I hope the authors can add more explanation, at least in the appendix, to enhance comprehension (see Question No. 1 for details).
* The transition from Proposition 3.2 to Equation 7 is not apparent, and more explanation might be needed.

**Questions:**

* Is the CTMC perspective necessary for the derivation of the ELBO, i.e., Proposition 3.2? If not, can the authors add the derivation from the discrete Markov chain perspective in the Appendix?
* Can the authors add more explanation on the transition from Proposition 3.2 to Equation 7, at least in the Appendix?
* The authors claim the new ELBO is planner-aware, so it should be better than the MDLM ELBO. But the final training objective is also a fixed objective; are there any justifications for why this objective is better for various commonly used planners? In other words, is this objective only better for the greedy-sampling planner, or is this objective better for all the commonly used samplers for masked diffusion language models?
* I would like the authors to clarify the planners used in their experiments. Are the planners just greedy sampling? If so, can the authors provide results with other planners, such as block decoding topk confidence from Nie et al. [1], confidence thresholding from Wu et al. [2], adaptive MDM inference from Kim et al. [3], and remasking samplers from Wang et al. [4]?
* From a theoretical perspective, the authors claim that P-ELBO is a unified framework for all possible planners. Can the authors provide an instantiation for the four planners mentioned above?
* In the right subplot of Figure 4, the downstream performance monotonically grows with $\alpha$; what if $\alpha$ goes beyond 5? I am assuming that at some point the downstream performance will go down, otherwise we should not interpolate between the MDLM loss and PAPL loss, but use PAPL loss alone instead.

---
**References**

[1] Nie, S., Zhu, F., You, Z., Zhang, X., Ou, J., Hu, J., Zhou, J., Lin, Y., Wen, J.R. and Li, C., 2025. Large language diffusion models. arXiv preprint arXiv:2502.09992.

[2] Wu, C., Zhang, H., Xue, S., Liu, Z., Diao, S., Zhu, L., ... & Xie, E. (2025). Fast-dllm: Training-free acceleration of diffusion llm by enabling kv cache and parallel decoding. arXiv preprint arXiv:2505.22618.

[3] Kim, J., Shah, K., Kontonis, V., Kakade, S., & Chen, S. (2025). Train for the worst, plan for the best: Understanding token ordering in masked diffusions. arXiv preprint arXiv:2502.06768.

[4] Wang, G., Schiff, Y., Sahoo, S. S., & Kuleshov, V. (2025). Remasking discrete diffusion models with inference-time scaling. arXiv preprint arXiv:2503.00307.

---

> ### Author Response · Authors · 2025-11-22
> **Response to LjAj Part 1**
>
> We thank the reviewer for their time and effort in reviewing our work. We are glad that the reviewer found the work clear, the theory "valuable and useful to the community”, and the method “simple and practical”. We now respond to the main questions and clarifications raised by the reviewer:
>
> ## Necessity of CTMC perspective
>
> The CTMC perspective is not strictly necessary for our results, as this type of result can be approached from multiple directions. In the appendix, we first prove the proposition from a discrete time perspective in Appendix A.1.4. We also provide the proof from a continuous time CTMC perspective in A.3.3 for completeness and ease of understanding for readers who are more familiar with this perspective. We believe that supplying proofs from both directions helps in readability and communication between different subcommunities of discrete diffusion. We have highlighted this in the updated draft after proposition 3.2 adding where to find both versions of the proof.
>
> ## Additional explanation after Proposition 3.2
>
> We thank the reviewer for highlighting this potential area for confusion. We now detail how one moves from the ELBO of proposition 3.2 specialized to the greedy setting to the PAPL loss of equation (7) in Appendix A.2.3, and refer the reader to this discussion directly after the statement of equation (7) in Section 3.4. This series of approximation steps is outlined in 3.4, but in Appendix A.2.3 a precise description using equations is performed. For clarity, we describe these steps here:
>
> First, we observe that specializing Proposition 3.2 to greedy ancestral results in the ELBO of Corollary A.9. The reference paths become deterministic and far from the easily implemented "uniform at random’’ masking of the vanilla ELBO. We thus approximate the greedy planner using a ``soft greedy planner,’’ whose ELBO takes the form of Corollary A.10.
>
> Corollary A.10’s ELBO takes the form of $E_1+E_2$, where $E_1$ is the same as the vanilla ELBO but with the uniform weights $1/(L-k)$ replaced by the softmax weights $w^i$ and where the reference path is along ``soft greedy’’ trajectories rather than uniformly at random. $E_2$ is the KL divergence between the supervised weights and the effective ones using the output of the planner.
>
> We then detach gradients from the weights, using the training signal only from the logits of the denoiser. This causes $E_2$ to become constant, so we remove it from the loss.
>
> We replace the ``soft greedy trajectory’’ with the uniform trajectory. The purpose of using the soft-greedy approximation was to make this ablation. The next position to be decoded has full support on all currently masked positions, unlike the pure greedy setting.
>
> At this point the loss is the same as vanilla MDM, except the uniform weights on token positions are replaced by $w^i$. Interpolating with the vanilla MDM loss with parameter alpha leads to the new weights $(1+\alpha w^i)/(L-k)$ used in the PAPL loss.

---

> > ### Author Response · Authors · 2025-11-22
> > **Response to LjAj Part 2**
> >
> > ## Effectiveness across planners
> >
> > We use the popular greedy-ancestral sampler as the focus of our exposition, designing the PAPL loss of Equation (7) as a computationally-viable surrogate for the true greedy ELBO resulting from specializing Proposition 3.2 to the greedy planner, which can be found as Corollary A.9. As the PAPL loss is designed with greedy decoding using the denoiser’s logits in mind, we thus focus our experiments on a modified greedy decoming method, namely P2-Self.
> >
> > - Empirically: We clarify that all experiments were done with the P2-self planner. We include additional experiments with other planners to understand how these various planners function in the newly added tables 5 and 6. Across all experiments P2-self outperforms other sampling methods. We remark that P2-self is a greedy sampling method which also allows for token corrections.
> >
> > - Conceptually: The PAPL planner is designed for a greedy sampler. While in principle any sampler can be used, it is unclear how to derive a computationally tractable loss like equation (7) for arbitrary planners. Further work would need to be performed in order to find a computationally viable surrogate to be used for other planners.
> >
> > - Theoretically: Our methodology can provide an ELBO immediately for many of the planners referenced by the reviewer. We have updated the draft to show how one would insert the top-k probability marginal planner into the ELBO of Proposition 3.2 in Section A.2 of the Appendix, where instantiations for the Uniform, Greedy, and soft-greedy planners were already discussed. For strategies which unmask multiple positions simultaneously and allow for remasking, we previously detailed how to adapt our strategy to general P2 and RDM in Appendix A.4.2. We now move this discussion to Appendix A.5 and also discuss how to insert Top-k block denoising into the appropriate modified ELBO of Proposition A.13. While confidence-thresholding does not immediately fit into the framework of A.13, as the number of tokens to be decoded concurrently becomes adaptive, we also outline how the general principle of our formulation could be used to obtain a rigorous ELBO in this setting.
> >
> > ## Performance with respect to $\alpha$
> >
> > Yes, this is correct. We found that increasing $\alpha$ above 5, in this setting, leads to worse performance. We don’t have complete runs in these settings, as we stopped them as soon as we noticed them performing worse to save compute. We have clarified this in section 4 of the manuscript.
> >
> > ## Concluding remarks
> >
> > We thank the reviewer again for their time and effort during this rebuttal process. We hope our answers here have aided in clarifying all the great questions raised by the reviewer. If there are any further questions held by the reviewer, we remain eager to engage further.

---

### Official Review · Reviewer_2qD5 · 2025-11-01

**Soundness:** 3
**Presentation:** 3
**Contribution:** 3
**Rating:** 4
**Confidence:** 4

**Summary:**

The paper shows the mismatch of diffusion objective between training and testing under planning and address it by revising the ELBO to incorporate planner basics in training. The new ELBO allows to effectively exploit any planning strategies at test time and it is simple to implement. The method show performance improvements on various tasks, such as protein sequences, text generation, code generation.

**Strengths:**

1. The paper tackle the relevant and important mismatch issue of training and testing of diffusion model under planning.
2. The method is theoretically grounded though it is quite dense.

**Weaknesses:**

1. Using confidence as a heuristic indicator could be unreliable for training. This is exactly shown in Figure 5 in the appendix. Though I see the authors using the vanilla DLM loss as a rescue, I think this is not an efficient way since at earlier training iterations, the confidence from model is not reliable, inducing noise into training (this is bad). One alternative is to do annealing where earlier step, use uniform weight as a sole vanilla DLM and it is annealed gradually to planner-weighted DLM.
2. Look at the Table 1 and Table 2, it seems that the method negatively affect the data entropy since both are worse than vanilla model. Why?
3. What is limitation? It is worth to mention.

**Questions:**

1. Is the model sensitive to the temperature and the weight alpha? Need more ablation on text generation and coding.

2. What exactly P2 sampling is  used? Since the P2 sampling has different variants.

3. Apart from P2 sampling, have the authors tried other planning strategies like greedy decoding like MaskGIT, top-k, top probability margin (from Kim et al, https://arxiv.org/abs/2502.06768).

---

> ### Author Response · Authors · 2025-11-22
> **Response to 2qD5 Part 1**
>
> We thank the reviewer for their time and effort reviewing our paper. We are heartened to hear that the reviewer feels that our PAPL paper tackles “the relevant and important” training and inference of discrete diffusion models under planning. Furthermore, we appreciate that the reviewer found our paper to be “theoretically grounded”. We now answer the main questions and clarification points raised in the review.
>
> ## Confidence as heuristic
>
> We thank the reviewers for the constructive feedback regarding the potential unreliability of using model confidence as a planning heuristic during training. We agree that, early in training, confidence estimates may be noisy. In our revised submission, we include an additional ablation (Figure 6) that directly examines this behavior. The results confirm the reviewer’s intuition: at the beginning of training, confidence-based weighting is indeed unstable. However, as training progresses and the denoiser becomes more accurate, confidence becomes increasingly informative and leads the model to better align with the inference-time planning procedure.
>
> To be more specific, The training curves further reveal characteristic dynamics of the PAPL objective. Early in training, the PAPL loss remains small because the denoiser has not yet formed meaningful beliefs about the correct token positions, causing the planner-dependent weights $w_i$ to be close to zero. As the model begins to identify correct positions with higher certainty, these weights increase, leading to a temporary rise in the PAPL loss. Once the denoiser becomes sufficiently confident, the loss decreases and eventually mirrors the behavior of the standard MDM loss. These dynamics provide additional evidence that PAPL naturally adapts its emphasis as the model’s confidence improves, stabilizing precisely when confidence becomes a reliable signal.
>
> > One alternative is to do annealing where earlier step, use uniform weight as a sole vanilla DLM and it is annealed gradually to planner-weighted DLM.
>
> This is an interesting suggestion. This is similar to starting from an initial base DLM that is trained without any planning and slowly annealing this with confidence weighting for finetuning. We in fact have tried this experiment and find that it works well for setting, as shown in Figure 2 (d). We thank the reviewer for this suggestion, and agree that it is an interesting setting to consider.
>
> ## Impact on data entropy
>
> The reviewer makes an astute observation that the data entropy reduces under PAPL training in table 1 and 2. In most cases, we find a general tradeoff between performance and sample diversity. To ablate this further and traverse the pareto front, we conduct a new ablation in the table below that fixes the data entropy to the same level as the vanilla DLM model by increasing the sampling temperature of a PAPL trained model. We find that the performance of PAPL under this setting of matching data entropy still exceeds that of a vanilla DLM model—highlighting the importance of training using the planner that will ultimately be used in inference.
> | Model | pLDDT | pTM | pAE | Foldability (%) | Entropy |
> | ---------------------------------- | ----: | ---: | ----: | --------------: | ------: |
> | DLM-150M | 81.32 | 0.65 | 12.00 | 42.43 | 3.21 |
> | DLM-150M + PAPL (ours) | 81.48 | 0.72 | 8.97 | 59.40 | 3.12 |
> | DLM-150M + PAPL (ours), Temp = 0.9 | 80.78 | 0.68 | 11.37 | 45.92 | 3.27 |
>
> ## Limitations
>
> We value the reviewers point that it is important to highlight limitations of our PAPL algorithm. The most clear limitation of our work is that we consider primarily confidence based (self)-planning, while in practice there has been evidence that shows that other planning strategies can improve not only on the vanilla DLM model but also self-planning [1]. Consequently, training with confidence planning means that at inference time it is no longer accurate—from a theoretical perspective—to modify the planning strategy. In addition, as this planning is now a component of training this limits the nature of planners that can be leveraged to those ones that are computationally lightweight—as opposed to inference where planning can be viewed as a mechanism for test time scaling. We have updated the conclusion section of our paper to better highlight these limitations.

---

> > ### Author Response · Authors · 2025-11-22
> > **Response to 2qD5 Part 2**
> >
> > ## Questions
> >
> > > Is the model sensitive to the temperature and the weight alpha? Need more ablation on text generation and coding.
> >
> > Thank you for the interesting question. We have conducted the ablation over temperature and weight alpha already for the protein experiments whose results can be found in Fig 4 of the main paper.  Empirically, lowering the temperature below the default $\tau = 1$ consistently improves protein foldability, while increasing $\tau$ degrades performance, indicating that sharper (more peaked) planner distributions yield more effective supervision. For the planner weight, increasing $\alpha$ up to $\alpha = 5$ improves stability and final performance, showing that giving more emphasis to planner-weighted paths is beneficial. However, pushing $\alpha$ beyond 5 hurts performance, highlighting the importance of interpolating between the vanilla MDLM loss and the PAPL loss rather than fully replacing one with the other.
> >
> > With regards to the request of additional similar ablations on text generation and coding, we believe this does not meaningfully add further insights given the existing ablation on proteins. Furthermore, we argue that retraining text and coding requires a compute budget of 16 H100’s run for 2 days (or for code, 2 weeks) for a single run. As a result, running complete sets of ablations would require a computational cost that is currently out of reach—without adding tremendous value given an ablation of this nature already exists for proteins.
> >
> > > What exactly P2 sampling is used? Since the P2 sampling has different variants.
> >
> > We use P2-self planning in PAPL as it coincides with confidence based planning.
> >
> > > Apart from P2 sampling, have the authors tried other planning strategies like greedy decoding like MaskGIT, top-k, top probability margin
> >
> > Thank you for this suggestion. We now report inference of a PAPL model as originally described in the paper but with different sampling strategies. We report these results in Appendix B.3.2 as Table 6 for code and Appendix B.2.2 as Table 5 for text, respectively. For both tasks, P2-self planning (the strategy used in all other experiments) outperforms other decoding strategies across the board. This is to be expected of the PAPL fine-tuned denoisers, as the PAPL loss is designed with greedy ancestral decoding in mind, and P2-self is a version of greedy ancestral decoding which also allows for token correction.
> >
> > Table 6:
> > | Method                           | HumanEval Pass@1 | HumanEval Pass@10 | HumanEval+ Pass@1 | HumanEval+ Pass@10 | MBPP Pass@1 | MBPP Pass@10 | MBPP+ Pass@1 | MBPP+ Pass@10 | HumanEval Infill Pass@1 | SantaCoder Pass@1 |
> > |----------------------------------|------------------:|-------------------:|-------------------:|--------------------:|-------------:|--------------:|--------------:|---------------:|--------------------------:|-------------------:|
> > | **P2-self plan (Peng et al., 2025)** | 20.8 | 38.4 | 17.6 | 35.2 | 16.7 | 38.4 | 23.9 | 53.6 | 77.4 | 56.4 |
> > | Ancestral                          | 3.3  | 18.3 | 3.2  | 15.2 | 1.8  | 13.2 | 2.9  | 21.8 | 72.7 | 53.8 |
> > | Greedy Ancestral (Chang et al., 2022) | 9.3  | 31.1 | 8.1  | 28.7 | 5.3  | 29.0 | 8.7  | 41.5 | 75.1 | 53.7 |
> > | Entropy - Based Confidence                         | 12.6 | 35.4 | 10.9 | 29.9 | 9.2  | 36.8 | 15.2 | 50.7 | 75.1 | 53.2 |
> > | TopK-margin (Kim et al., 2025)   | 7.6  | 27.4 | 6.5  | 26.2 | 3.9  | 24.0 | 6.2  | 33.5 | 75.0 | 54.4 |
> >
> > Table 5:
> > | **Metric** | **Method**           | **T=32** | **T=64** | **T=128** |
> > |-------------|----------------------|----------:|----------:|-----------:|
> > | **MAUVE**   | Greedy               | 0.011     | 0.021     | 0.056      |
> > |             | Probability Margin   | 0.011     | 0.039     | 0.051      |
> > |             | P2-Self              | 0.013     | 0.046     | 0.067      |
> > | **Gen PPL** | Greedy               | 44.34     | 34.18     | 29.38      |
> > |             | Probability Margin   | 44.27     | 34.37     | 29.39      |
> > |             | P2-Self              | 40.19     | 29.98     | 24.33      |
> > | **Entropy** | Greedy               | 5.35      | 5.30      | 5.25       |
> > |             | Probability Margin   | 5.35      | 5.30      | 5.25       |
> > |             | P2-Self              | 5.32      | 5.24      | 5.16       |
> >
> >
> >
> > ## Concluding remarks
> >
> > We thank the reviewer again for their time and effort during this rebuttal process. We hope our answers here have aided in clarifying all the great questions raised by the reviewer. If there are any further questions held by the reviewer, we remain eager to engage further. If the reviewer feels satisfied with our responses and new rebuttal results, we would be thrilled if the reviewer would consider a fresher evaluation of our work with a potential score upgrade.
> >
> > ## References
> >
> > [1] Peng, Fred Zhangzhi, et al. "Path planning for masked diffusion model sampling." arXiv preprint arXiv:2502.03540 (2025).

---

### Public Comment · ~Boyuan_Zou1 · 2025-11-17
**Open source code if possible**

Hi author, I am a phd student in UW Madison and I am interested in your research project and want to test on other kinds of planner in training. I plan to do a final project for my course on this. If it is possible, would you mind sharing your codes with me, thank you so much for help.

---

> ### Author Response · Authors · 2025-11-20
>
> Thank you for your interest in our work. To preserve double-blindness of the review process we recommend contacting the authors directly.

---

### Author Response · Authors · 2025-11-22
**Global Response**

We are grateful to all reviewers for the time, care, and detailed feedback in evaluating our submission. We were encouraged to see broad agreement on the motivation and value of the work.

For example, Reviewer 2qD5 notes that we “tackle the relevant and important mismatch issue of training and testing,” and Reviewer TRnB similarly emphasizes that the training–inference mismatch we identify is a “concrete theoretical finding” that justifies revisiting standard objectives. We also appreciate Reviewer LjAj’s assessment that the paper’s “storyline … is clear,” the theory is “valuable and useful for the community,” and that PAPL is a “lightweight” and “simple and practical” modification.

Across reviews, we are thankful that the practicality of PAPL was repeatedly highlighted (e.g., 2qD5: “simple to implement”; TRnB: “one-line modification”), and that the empirical gains across proteins, text, and code were viewed as consistent and meaningful (LjAj remarks they were “surprised by the increase in pass@10,” and TRnB describes the validation as “comprehensive across diverse domains”). Even where reviewers found parts dense, we take that as a clear signal to improve the exposition.


At the same time, we take the shared concerns seriously, and our revision focused on addressing them directly.

(1) **Confidence-weighting stability**: in response to 2qD5’s concern that confidence is unreliable early in training, we added a new ablation with full training curves (new Figure 6 / Appendix C.1) and an annealed variant showing the reviewer’s suggestion works well in practice (Figure 2(d)).


(2) **Clarifying and measuring the P-ELBO → PAPL approximation**: following questions from LjAj and TRnB, we added a step-by-step derivation of Eq. (7) from Proposition 3.2 (new Appendix A.2.3), supplemented by an analysis of the optimal planner (Appendix A.1.5), and explicit proof pointers after each proposition/corollary. We also provide an empirical decomposition that numerically estimates the correction gap (new Table 7).


(3) **Planner generality and instantiations**: in response to LjAj and 2qD5, we added to the appendix concrete instantiations of additional planners within our framework (Appendix A.5) and added new inference-time comparisons across planners for text and code (new Tables 5 and 6 in Appendices B.2.2 and B.3.2). Finally, to address 2qD5’s entropy concern, we added a temperature-matched diversity study (new entropy-matched protein table in the rebuttal) showing the performance gains persist at comparable entropy.
We strongly believe these changes make the paper clearer and more complete, and we thank the reviewers again for helping us strengthen it.

---

### Author Response · Authors · 2025-11-29
**Global Response**

## Summary of Contributions & Rebuttal Updates

We thank the reviewers for their valuable feedback and the Area Chair for overseeing this process. In this work, **Planner Aware Path Learning (PAPL)** identifies and resolves a fundamental theoretical inconsistency in Discrete Diffusion Models: the mismatch between uniform-random training and planner-guided inference. We derive the **Planned ELBO (P-ELBO)** to theoretically ground this setting and propose PAPL as a tractable, lightweight training objective.

During the rebuttal period, we significantly strengthened the paper with new theoretical derivations and comprehensive experiments in response to reviewer feedback. These revisions have solidified the validity of our claims and the robustness of our method:

* **Theoretical Completeness (Addressing LjAj, TRnB):**
  We added a rigorous step-by-step derivation of the PAPL objective (Eq. 7) from the P-ELBO (Appendix A.2.3) and a theoretical analysis of planner optimality (Appendix A.1.5). We further quantified the approximation gap empirically in Table 7, showing the behavior of the bound components.

* **Training Stability (Addressing 2qD5):**
  We addressed concerns regarding confidence weighting stability by providing full training curves (Figure 6) and demonstrating that an annealed variant (suggested by Reviewer 2qD5) works effectively (Figure 2d).

* **Planner Generality (Addressing LjAj, jzen, 2qD5):**
  We demonstrated that PAPL is not limited to specific decoding strategies. New results in Tables 5 and 6 show PAPL consistently outperforming other strong inference baselines (including Top-K margin and probability margin) across text and code domains.

* **Diversity & Entropy (Addressing 2qD5):**
  We added a temperature-matched ablation showing that PAPL’s performance gains persist even when controlling for output entropy/diversity.

## Closing Remarks & Reviewer Engagement

We are thankful for the suggestions from the reviewers.
However, we note that reviewers have not yet had the opportunity to respond to our detailed rebuttals and new experimental data posted on Nov 22.

* **To Reviewer TRnB:**
  We provided the requested empirical measurement of the variational gap (Table 7) and clarified the derivation for “hard” planners, directly addressing the “Weaknesses” listed in the review.

* **To Reviewer jzen:**
  We resolved the confusion regarding Proposition 3.1 (clarifying it as a counter-example to the validity of the standard ELBO under planning, rather than a consistency claim) and provided the requested comparisons to greedy sampling (Tables 5 & 6), where PAPL remains superior. **We would like to note that our response allowed reviewer jzen to increase the score from 4->6 prior to the reversion of the scores mandated by the Program Chairs due to the current events.**

* **To Reviewer LjAj:**
  We added the requested derivations connecting Proposition 3.2 to Equation 7 and expanded on the instantiation of other planners in the Appendix.

We respectfully ask the Area Chair to consider these unaddressed rebuttals and the substantial strengthening of the paper when making the final decision.

---

### Meta-Review · Area_Chair_3iDM · 2026-01-05

**Summary:**

Train-test mismatch is a fundamental challenge that arises in standard diffusion language models. This paper introduces a new ELBO training objective, which evaluates the ELBO under the distribution of a given planner determining the denoising. By reducing this shift, they find substantial gains over past art, with minimal overheard.

**Reviewer Concerns:**

Reviewers stated some concerns around burdensome notation and clarity, which were addressed. The more significant, unanimous concern was that the ELBO derivation was unclear and had some technical weaknesses. I am personally less concerned about this, since the ELBO itself is a heuristic motivating a loss. Moreover, the authors provided ample explanation of their reasoning in the rebuttal. In general, the authors weaknesses were quite limited.

**Reviewer Scores:**

The reviewers were split in score; however, the high quality reviewers provided notably higher scores, and I believe both of their concerns were adjusted.

More specifically:
(1) 1 reviewer was given a score of 4, but with a shorter review, and agreed to raise the score to a 6 prior to scores being reverse. Their concerns seemed to be met (jzen)
(2) 1 reviewer gave a score of 6, but their concerns were thoroughly addressed (particular hard, greedy, soft decoding), and likely would have been raised to an 8
(3) 1 gave a score of an 8, and the authors further addressed concerns regarding clarity
(4) 1 reviewer gave a score of 4, which I suspect would have been raised to a 6 (2qD5) as, from my reading all concerns were addressed. In particular, the authors include careful ablations and plots of training dynamics demonstrating the validity of the heuristic steps in their derivation.

---

### Decision · Program_Chairs · 2026-01-26

Accept (Oral)